# Lower-level Duality Based Penalty Methods for Bilevel Hyperparameter Optimization

## Abstract

Hyperparameter optimization (HO) is a critical task in machine learning and can be formulated as a bilevel optimization problem. However, many existing algorithms for addressing nonsmooth lower-level problems involve solving sequential subproblems, which are computationally expensive. To address this challenge, we propose penalty methods for solving HO, leveraging strong duality between the lower-level problem and its dual. We show that the penalized problem closely approximates the optimal solutions of the original HO under certain conditions. Moreover, we develop first-order single-loop algorithms to solve the penalized problems efficiently. Theoretically, we establish the convergence of the proposed algorithms. Numerical experiments demonstrate the efficiency and superiority of our method.

## 1 Introduction

Hyperparameter optimization (HO) arises in many diverse fields, neural architecture search [16, 29, 57], feature learning [35], ensemble models [25], semi-supervised learning [42] and sample-weighting schemes [34, 77, 74, 82]. The hyperparameters control model complexity, training stability and convergence. Unlike model parameters, they need to be chosen externally. A poor choice can cripple performance, whereas good hyperparameters greatly enhance accuracy, robustness and generalization.

Regularization is a common way to guide hyperparameter tuning, especially in regression and classification [32]. By adding a penalty term to the empirical risk, one trades off data fitting against model complexity to curb overfitting. The general framework can be formulated as

$$\min_{\mathbf{x}} \ l(\mathbf{x}) + \sum_{i=1}^{M+1} \lambda_i R_i(\mathbf{x}), \tag{1}$$

where $l(\mathbf{x})$ represents the loss function and $\boldsymbol{\lambda} = (\lambda_1, \lambda_2, ..., \lambda_{M+1})$ encompasses hyperparameters. Meanwhile, $R_i(\mathbf{x}), i = 1, 2, ..., M + 1$ denotes the regularizers related to norms, which can be categorized as follows:

$$R_i(\mathbf{x}) = \|\mathbf{x}\|_{(i)}, \ i = 1, 2, ..., M, \ R_{M+1}(\mathbf{x}) = \frac{1}{2}\|\mathbf{x}\|_2^2. \tag{2}$$

For each $i$, $\|\cdot\|_{(i)}$ represents a specific norm, such as the $\ell_1$, $\ell_2$, $\ell_\infty$, $\ell_{1,2}$ norm for vectors, the spectre or nuclear norm for matrices, or other commonly used norms. Note that these two types of regularizers may appear simultaneously or individually.

Based on the formulation (1), training/validation approach is involved as a sophisticated method. This method optimizes parameters in the form (1) on the training set and observes the corresponding error on the validation set. The approach can be summarized as bilevel optimization framework [57, 9] and

Submitted to 39th Conference on Neural Information Processing Systems (NeurIPS 2025). Do not distribute.

has demonstrated outstanding performance in practical applications [66, 31, 35, 15]. In essence, the process can be outlined in the following bilevel optimization (BLO) [72, 28]:

$$\min_{\mathbf{x}\in\mathbb{R}^n,\boldsymbol{\lambda}\in\mathbb{R}_+^{M+1}} L(\mathbf{x}) \quad \text{s.t. } \mathbf{x} \in \arg\min_{\hat{\mathbf{x}}} \left\{ l(\hat{\mathbf{x}}) + \sum_{i=1}^{M+1} \lambda_i R_i(\hat{\mathbf{x}}) \right\}, \tag{3}$$

where $L, l, R_i : \mathbb{R}^n \to \mathbb{R} \cup \{+\infty\}$ are proper, closed functions, $\mathbf{x}$ is the parameter to learn, and $\boldsymbol{\lambda}$ is hyperparameter. In BLO (3), the lower-level (LL) problem serves as a base learner, aiming to determine the optimal hypothesis on the training set for a given hyperparameter configuration. In contrast, the upper-level (UL) problem aims to identify the hyperparameter and corresponding hypothesis that minimizes the given criteria on the validation set. We explain the mathematical forms of the component functions in problem (3) using several illustrative examples listed in Table 1, including elastic net [100], sparse group Lasso [83], logistic regression [68, 46], low-rank matrix completion [20] and smoothed support vector machine [78, 65].

Table 1: Examples of bilevel hyperparameter optimization [48, 31, 46] in the form (3).

| Machine learning algorithm | Upper Criteria | Base Learner |
|---|---|---|
| Elastic net | $\frac{1}{2}\sum_{i\in I_{val}} \|b_i - \mathbf{x}^T\mathbf{a}_i\|^2$ | $\frac{1}{2}\sum_{i\in I_{tr}} \|b_i - \mathbf{x}^T\mathbf{a}_i\|^2 + \lambda_1\|\mathbf{x}\|_1 + \frac{\lambda_2}{2}\|\mathbf{x}\|_2^2$ |
| Sparse group Lasso | $\frac{1}{2}\sum_{i\in I_{val}} \|b_i - \mathbf{x}^T\mathbf{a}_i\|^2$ | $\frac{1}{2}\sum_{i\in I_{tr}} \|b_i - \mathbf{x}^T\mathbf{a}_i\|^2 + \sum_{m=1}^M \lambda_m\|\mathbf{x}^{(m)}\|_2 + \lambda_{M+1}\|\mathbf{x}\|_1$ |
| Smoothed support vector machine | $\sum_{i\in I_{val}} l_h(b_i\mathbf{w}^T\mathbf{a}_i)$ | $\sum_{i\in I_{tr}} l_h(b_i\mathbf{w}^T\mathbf{a}_i) + \frac{\lambda}{2}\|\mathbf{w}\|^2$ (with constraint $-\bar{\mathbf{w}} \leq \mathbf{w} \leq \bar{\mathbf{w}}$.) |
| Low-rank matrix completion | $\sum_{(i,j)\in\Omega_{val}} \frac{1}{2}\|M_{ij} - \mathbf{x}_i\theta - \mathbf{z}_j\beta - \Gamma_{ij}\|^2$ | $\sum_{(i,j)\in\Omega_{tr}} \frac{1}{2}\|M_{ij} - \mathbf{x}_i\theta - \mathbf{z}_j\beta - \Gamma_{ij}\|^2 + \lambda_0\|\Gamma\|_* + \sum_{g=1}^G \lambda_g\|\theta^{(g)}\|_2 + \sum_{g=1}^G \lambda_{g+G}\|\beta^{(g)}\|_2$ |
| Logistic regression | $\sum_{j\in I_{val}} \log(1 + e^{-b_j\mathbf{x}^T\mathbf{a}_j})$ | $\sum_{j\in I_{tr}} \log(1 + e^{-b_j\mathbf{x}^T\mathbf{a}_j}) + \frac{\lambda}{2}\|\mathbf{x}\|^2$ |

$l_h$ denotes the smoothed hinge loss given by $l_h(x) = \frac{1}{2} - x$ if $x \leq 0$, $\frac{1}{2}(1-x)^2$ if $0 \leq x \leq 1$ and 0 else.

## 1.1 Related Work

**Hyperparameter Optimization.** A variety of approaches have been developed for hyperparameter optimization (HO) [44]. The simplest model-free techniques include grid search [45] and random search [12]. More advanced methods such as Bayesian optimization [11, 84] iteratively select evaluation points based on prior observations. However, these approaches often struggle with scalability when faced with high-dimensional parameter spaces.

**Bilevel Optimization.** Bilevel optimization (BLO) underpins many machine learning tasks, including meta-learning [33], adversarial learning [19, 86, 87], reinforcement learning [80, 85, 93, 89], model selection [47, 39], generative adversarial networks [38, 40], and game theory [55]. Early methods primarily relied on gradient-based algorithms, which can be broadly classified into two categories: Iterative Differentiation (ITD) and Approximate Implicit Differentiation (AID). ITD methods unroll the lower-level problem and compute hypergradients via backpropagation [34, 35, 41, 61, 5, 77], while AID methods derive gradients from the lower-level optimality conditions [72, 73, 63, 92, 91].

Recent advances include fully first-order methods that avoid Hessian and implicit gradient computations [23, 54, 24]. To address the challenge of multiple lower-level minima, [59] introduce a value-function-based reformulation, leading to penalization-based algorithms [60]. This line of work has grown into a prominent direction, with various penalty-based single-level reformulations proposed in [79, 64, 50, 49, 56]. Another promising direction leverages the Moreau envelope to smooth the bilevel structure, yielding single-loop, Hessian-free algorithms capable of converging to well-defined KKT points [37, 95, 94].

For BLO with nonsmooth lower-level problems, [14] propose an implicit differentiation framework based on block coordinate descent, which is later extended to general nonsmooth settings [15]. Other approaches include DC methods [96, 97] and penalized DC formulations [36], both requiring the computation of the lower-level value function. Smoothing-based strategies have also been explored to handle nonsmoothness [3, 2, 71]. Additionally, [23] present a gradient-free method with inexact subproblem solutions, while [22] reformulate BLO via duality, avoiding the value function entirely and solving the problem through cone programming. [62] further extend the Moreau envelope approach to nonsmooth lower-level problems, offering efficient single-loop algorithms.

## 1.2 Motivations and Contributions

In this work, we focus on solving the bilevel optimization (3). We extend the reformulation initially proposed by [22] and incorporate penalty strategy. We demonstrate that our framework is applicable to commonly used hyperparameter optimization problems schemed in (3). Moreover, we propose

the **L**ower-level **D**uality Based **P**enalty **M**ethods (LDPM), which are first-order algorithms specifically designed for the penalized problem. The algorithms efficiently handle the nonsmooth norm components with epigraphic projections. Notably, our algorithms are single-loop and Hessian-free, relying solely on the first-order information of the functions in (3). Theoretically, we establish the convergence results of the algorithms under mild conditions. We summarize our contributions as follows.

- We propose a penalty method based on lower-level duality for hyperparameter optimization (3), which is in the form of BLO with nonsmooth LL problem.
- We introduce two first-order *single-loop* algorithms to solve the penalized problem and provide theoretical proof of the convergence.
- We evaluate the efficiency of our algorithms with numerical experiments on synthetic and real-world data. Experimental results validate superiority of our algorithm in practical scenarios.

## 2    Penalty-based Approach

In this section, we propose our penalization framework for the original problem (3). Prior to this, we observe that the loss functions of base learners in Table 1 share a unified structure of the form $\varphi(A\mathbf{x} - \mathbf{b})$, where $A\mathbf{x} - \mathbf{b}$ abstracts the data-sample relationship. Accordingly, we denote that

$$l(\mathbf{x}) = \varphi(A_t\mathbf{x} - \mathbf{b}_t),  \tag{4}$$

where $l(\mathbf{x})$ corresponds to the loss on validation and training sets as described in (3). We now provide a detailed discussion of the mathematical forms of the function $\varphi$ for problems in Table 1, along with the expressions $A_t, \mathbf{b}_t$:

**Least squares loss**: $\varphi(t) = \frac{1}{2}t^2$, with $A_t\mathbf{x} - \mathbf{b}_t = A_{tr}\mathbf{x} - \mathbf{b}_{tr}$.

**Smoothed hinge loss**: $\varphi(t) = l_h(t)$ with $A_t\mathbf{w} - \mathbf{b}_t = (\mathbf{b}_{tr}A_{tr})\mathbf{w}$.

**Logistic loss**: $\varphi(t) = \log(1 + e^{-t})$, with $A_t\mathbf{x} - \mathbf{b}_t = (\mathbf{b}_{tr}A_{tr})\mathbf{x}$.

Building on the inner structure of $\varphi$ and $R_i$, our approach is grounded in a reformulation based on the duality of LL problem. We embrace the idea initially proposed by [22] and summarize the following lemma, which is a modification and extension of [22, Theorem 2.1].

**Lemma 2.1.** *Given the convex lower semi-continuous functions $l$ and $R_i$, if $\mathrm{ri}(\mathrm{dom}\ l \cap (\cap_{i=1}^{M+1}\mathrm{dom}\ R_i)) \neq \emptyset$[1], then problem (3) has the following equivalent form:*

$$\min_{\mathbf{x},\lambda,\rho,\xi} L(\mathbf{x}) \quad s.t. \quad \begin{cases} l(\mathbf{x}) + \sum_{i=1}^{M+1} \lambda_i R_i(\mathbf{x}) + \varphi^*(\boldsymbol{\xi}) + \sum_{i=1}^{M+1} \lambda_i R_i^*\left(\frac{\rho_i}{\lambda_i}\right) + \boldsymbol{\xi}^\top \mathbf{b}_t \ \leq\ 0, \\ A_t\,\boldsymbol{\xi} + \sum_{i=1}^{M+1} \boldsymbol{\rho}_i \ =\ \mathbf{0}. \end{cases}  \tag{5}$$

*where $\boldsymbol{\rho} = (\boldsymbol{\rho}_1, ..., \boldsymbol{\rho}_{M+1})$ and $A_t, \mathbf{b}_t, \varphi$ are consistent with those in (4) and $\varphi^*$ and $R_i^*$ are the conjugate functions of $\varphi$ and $R_i$ for $i = 1, 2, ..., M + 1$, respectively.[2]*

*Remark* 2.2. Slater's condition is broadly satisfied by all examples in Table 1, ensuring strong duality for the LL problem in (3) without requiring strong convexity. For instance, the least squares loss is not strongly convex, yet strong duality still holds under this condition.

We present a detailed proof of Lemma 2.1 in the Appendix A.1. Notably, each problem listed in Table 1 can be reformulated into the structure of (5). For clarity, we calculate the **closed-form** expressions of the conjugate functions in Appendix A.3. We remark that our reformulation utilizes the structure of $l(\mathbf{x})$ in (4), which is different from the one in [22, Theorem 2.1]. Notably, each problem listed in Table 1 can be reformulated into the structure of (5).

To elaborate, we discuss the terms $R_i^*$ in (5) as follows. For $i = 1, 2, ..., M$, $R_i$ denotes a norm, i.e., $R_i(\mathbf{x}) = \|\mathbf{x}\|_{(i)}$. In this case, we know that $R_i^*(\mathbf{y})$ is the indicator function of the set $\{\|\mathbf{y}\|_{*(i)} \leq 1\}$

---

[1]This condition is commonly known as Slater's condition. $\mathrm{ri}(\cdot)$ denotes the relative interior of the set.

[2]We define the conjugate $h^*(\mathbf{y}) = \sup_{\mathbf{x}}\{\mathbf{y}^T\mathbf{x} - h(\mathbf{x})\}$ for a function $h$.

where $\| \cdot \|_{*(i)}$ denoted the dual norm of $\| \cdot \|_{(i)}$ [18, Example 3.26]. The term $R_{M+1}$ denotes the squared $\ell_2$-norm, i.e., $R_{M+1}(\mathbf{x}) = \frac{1}{2}\|\mathbf{x}\|_2^2$. In this case, we can compute that $\lambda_{M+1} R_{M+1}^*(\frac{\boldsymbol{\rho}_{M+1}}{\lambda_{M+1}}) = \frac{\|\boldsymbol{\rho}_{M+1}\|_2^2}{2\lambda_{M+1}}$ [18, Example 3.27]. To refine the intricate constraints of (5), we introduce auxiliary variables $r_i$ and $s$ satisfying $R_i(\mathbf{x}) \le r_i$ and $\frac{\|\boldsymbol{\rho}_{M+1}\|_2^2}{2\lambda_{M+1}} \le s$. This results in a further reformulation based on Lemma 2.1.

**Proposition 2.3.** *The original problem (3) can be reformulated as*

$$
\begin{aligned}
\min_{\mathbf{x},\boldsymbol{\lambda},\boldsymbol{\rho},\mathbf{r},\boldsymbol{\xi},s} \quad & L(\mathbf{x}) \\
\text{s.t.} \quad & l(\mathbf{x}) + \sum_{i=1}^{M+1} \lambda_i r_i + \varphi^*(\boldsymbol{\xi}) + \boldsymbol{\xi}^T \mathbf{b}_t + s \le 0, \ A_t \boldsymbol{\xi} + \sum_{i=1}^{M+1} \boldsymbol{\rho}_i = \mathbf{0}, \\
& \|\mathbf{x}\|_{(i)} \le r_i, \|\boldsymbol{\rho}_i\|_{*(i)} \le \lambda_i, \ i = 1, 2, ..., M, \\
& \tfrac{1}{2}\|\mathbf{x}\|_2^2 \le r_{M+1}, \|\boldsymbol{\rho}_{M+1}\|_2^2 \le 2\lambda_{M+1} s.
\end{aligned}
\tag{6}
$$

For simplicity, we rewrite the left-hand of the first inequality constraint in (6) as:

$$
p(\mathbf{x}, \boldsymbol{\lambda}, \mathbf{r}, \boldsymbol{\xi}, s) = l(\mathbf{x}) + \sum_{i=1}^{M+1} \lambda_i r_i + \varphi^*(\boldsymbol{\xi}) + \boldsymbol{\xi}^T \mathbf{b}_t + s.
\tag{7}
$$

Now we consider the penalization of problem (6) as follows,

$$
\min_{\mathbf{z}} \ F_k(\mathbf{z}) \quad \text{s.t.} \quad
\begin{cases}
\|\mathbf{x}\|_{(i)} \le r_i, & \|\boldsymbol{\rho}_i\|_{*(i)} \le \lambda_i, \quad i = 1, 2, \ldots, M, \\
\tfrac{1}{2}\|\mathbf{x}\|_2^2 \le r_{M+1}, & \|\boldsymbol{\rho}_{M+1}\|_2^2 \le 2\lambda_{M+1} s.
\end{cases}
\tag{8}
$$

where $F_k(\mathbf{z}) := L(\mathbf{x}) + \beta_k p(\mathbf{x}, \boldsymbol{\lambda}, \mathbf{r}, \boldsymbol{\xi}, s) + \frac{\beta_k}{2}\|A_t \boldsymbol{\xi} + \sum_{i=1}^{M+1} \boldsymbol{\rho}_i\|^2$ with $\mathbf{z} := (\mathbf{x}, \boldsymbol{\lambda}, \boldsymbol{\rho}, \mathbf{r}, \boldsymbol{\xi}, s)$ for convenience, and $\beta_k$ serves as the penalty parameter. This penalty strategy is commonly employed in bilevel optimization [79, 62, 95, 94, 59]. Inspired by [70, Theorem 17.1], the following theorem reveals the relationship between the optimal solutions of penalization and reformulation (6).

**Theorem 2.4.** *Assume $L, l$ and $R_i$ are lower semi-continuous, with the loss function $l$ and the regularization term $R_i$ in LL objective being convex. Suppose the penalty parameter satisfying $\beta_k \to \infty$. If $\mathbf{z}^{k+1}$ is the minimizer of penalized problem (8) with $\beta_k$, then every limit point $\mathbf{z}^*$ of the sequence $\{\mathbf{z}^k\}$ is a solution to the reformulation (6).*

The proof of Theorem 2.4 is provided in Appendix A.2. From the equivalence between (5) and (6), it follows that if $\mathbf{z}$ is the solution of (6), then $(\mathbf{x}, \boldsymbol{\lambda}, \boldsymbol{\rho}, \boldsymbol{\xi})$ is the corresponding solution of (5). Thus, Theorem 2.4 also reveals the connection between (5) and (8).

# 3 Epigraphical Projection-based First-order Algorithms

We develop our algorithms based on the penalized formulation (8), beginning with general assumptions on the original problem (3) to support analysis and algorithm design.

**Assumption 3.1.** The UL objective $L$ is $\alpha_L$-smooth with respect to LL variable $x$. Additionally, as a loss function, $L$ is non-negative, i.e., $L(\mathbf{x}) \ge 0$ for all $\mathbf{x}$.

**Assumption 3.2.** The function $\varphi$ is convex. Moreover, the function $\varphi$ and its conjugate $\varphi^*$ is $\alpha_p$- and $\alpha_d$-smooth, respectively.

*Remark* 3.3. Assumptions 3.1 and 3.2 are satisfied by commonly used loss functions. Specifically, the problems listed in Table 1 adhere to Assumptions 3.1 and 3.2. We remark the UL objective $L$ *can be nonconvex*, which remains compatible with our framework.

*Remark* 3.4. The smoothness properties of $l$ are naturally inherited by $\varphi$. Therefore, Assumption 3.2 implies that $l$ is convex and Lipschitz smooth. Combined with the definition of $R_i$ in (2), Assumption 3.2 ensures that the LL problem in (3) is convex. Importantly, our framework relies only on the first-order differentiability and does not require the LL objective in (3) to exhibit strong convexity.

However, the primary challenges of solving (8) stem from the nonsmooth nature of the constraints, particularly when different norms are involved. We define corresponding sets for the constraints in

problem (8) in the form of cones as follows:

$$\mathcal{K}_i := \{(\mathbf{x}, \mathbf{r}) \mid \|\mathbf{x}\|_{(i)} \leq r_i\}, \ \mathcal{K}_i^d := \{(\boldsymbol{\rho}_i, \lambda_i) \mid \|\boldsymbol{\rho}_i\|_{*(i)} \leq \lambda_i\}, \quad i = 1, 2, ..., M,$$
$$\mathcal{K}_{M+1} := \{(\mathbf{x}, \mathbf{r}) \mid \|\mathbf{x}\|_2^2 \leq 2r_{M+1}\}, \ \mathcal{K}_{M+1}^d := \{(\boldsymbol{\rho}_{M+1}, \lambda_{M+1}, s) \mid \|\boldsymbol{\rho}_{M+1}\|_2^2 \leq 2\lambda_{M+1}s\}. \tag{9}$$

Furthermore, each set in (9) is projection-friendly, which facilitates efficient epigraphic projection of corresponding norms. The details of the projection operations are discussed in Appendix B.

Given these insights, a natural approach to manage the constraints in (8) is through projections onto $\mathcal{K}_i$ and $\mathcal{K}_i^d$. To address problems with different regularizers, we discuss the proposed algorithms in various scenarios. Section 3.1 focuses on problem (3) with single-round global regularization applied to the entire vector $\mathbf{x}$. Section 3.2 extends this to problems with multiple interacting regularizers. This division provides a structured approach to handling varying constraints and regularization terms.

## 3.1 Separable Regularizers

In this subsection, we explore the algorithm for (3) when the LL problem incorporates separate regularizers, structured as a single group of component-wise terms. Specifically, the LL problem in (3) can be expressed as

$$\mathbf{x} \in \arg\min_{\hat{\mathbf{x}}} \left\{ l(\hat{\mathbf{x}}) + \sum_{i=1}^{M} \lambda_i \|\hat{\mathbf{x}}^{(i)}\|_{(t)} \right\},$$

where $\mathbf{x}^{(i)}$ represents the $i$-th subvector of $\mathbf{x}$ with $\mathbf{x} = (\mathbf{x}^{(1)}, ..., \mathbf{x}^{(M)})$ and $\|\cdot\|_{(t)}$ represents a prescribed norm applied to each group.

**When $M = 1$,** the LL problem of (3) involves a single regularizer $R_1(\mathbf{x})$, corresponding to simpler models such as toy Lasso or logistic regression. In this case, the constraints of (8) simplify as follows. If $R_1(\mathbf{x}) = \|\mathbf{x}\|_{(t)}$ and $\|\cdot\|_{(t)}$ is a norm, the constraints of (8) reduce to:

$$\|\mathbf{x}\|_{(t)} \leq r_1, \ \|\boldsymbol{\rho}\|_{*(t)} \leq \lambda_1. \tag{10}$$

If $R_1(\mathbf{x}) = \frac{1}{2}\|\mathbf{x}\|_2^2$, the constraints of (8) simplify to:

$$\frac{1}{2}\|\mathbf{x}\|_2^2 \leq r_1, \ \frac{1}{2}\|\boldsymbol{\rho}\|_2^2 \leq \lambda_1 s. \tag{11}$$

The constraints (10)-(11) are consistent with the structure in (9) and can be compactly expressed as

$$\mathbf{z} \in \mathcal{K} := \mathcal{K}_1 \times \mathcal{K}_1^d. \tag{12}$$

**When $M > 1$,** the LL problem of (3) incorporates group regularization, where group-wise $\ell_2$-regularization is the most common choice. This setting is widely adopted in practice, as illustrated by examples such as group Lasso in Table 1. Although the problem may appear to involve multiple regularization terms and hyperparameters, it essentially amounts to applying a single-round regularization process over the entire variable $\mathbf{x}$. Under this structure, the constrains of (8) simplifies to:

$$\|\mathbf{x}^{(i)}\|_{(t)} \leq r_i, \ \|\boldsymbol{\rho}^{(i)}\|_{(t)} \leq \lambda_i, \ i = 1, \dots, M, \tag{13}$$

where $\boldsymbol{\rho}^{(i)}$ is the $i$-th subvector of $\boldsymbol{\rho}$ with $\boldsymbol{\rho} = (\boldsymbol{\rho}^{(1)}, ..., \boldsymbol{\rho}^{(M)})$. Since constraints of (13) are independent for each $i$, they can be equivalently expressed as:

$$(\mathbf{x}, \mathbf{r}) \in \mathcal{K}_1 \times \cdots \times \mathcal{K}_M, \ (\boldsymbol{\rho}, \boldsymbol{\lambda}) \in \mathcal{K}_1^d \times \cdots \times \mathcal{K}_M^d,$$

which implies that

$$\mathbf{z} \in \mathcal{K} := (\mathcal{K}_1 \times \cdots \times \mathcal{K}_M) \times (\mathcal{K}_1^d \times \cdots \times \mathcal{K}_M^d). \tag{14}$$

Importantly, we observe that $\mathcal{K}$ in (12) and (14) remains projection-friendly, facilitating efficient implementation. Accordingly, we adopt a gradient projection method to solve the penalized problem (8), as outlined in Algorithm 1. In each iteration, we update $\mathbf{z}$ as

$$\mathbf{z}^{k+1} = \text{proj}_{\mathcal{K}}(\mathbf{z}^k - e_k \nabla_{\mathbf{z}} F_k(\mathbf{z}^k)), \tag{15}$$

where $e_k > 0$ is the step size, and $\text{proj}_{\mathcal{K}}(\mathbf{x})$ is the projection of $\mathbf{x}$ onto $\mathcal{K}$.

---
**Algorithm 1** First-order Projection-based Method
---
1: Input $\boldsymbol{\lambda}^0, \boldsymbol{\xi}^0$, sequences $\{\beta_k\}, \{e_k\}$. Initialize $\mathbf{x}^0, \mathbf{r}^0, \boldsymbol{\rho}^0, s^0$.
2: **for** $k = 0, 1, 2, ...$ **do**
3:    Update $\mathbf{z}^{k+1}$ with projection gradient descent as (15).
4: **end for**
---

In Algorithm 1, we choose the penalty parameter as $\beta_k = \underline{\beta}(1 + k)^p$ with a constant $\underline{\beta} > 0$ and $0 < p < 1/2$, which corresponds to Theorem 2.4. Such a selection strategy is common in penalty method and augmented Lagrangian methods [70, 69, 27, 62, 95, 94]. The initialization of Algorithm 1 is detailed in Appendix C.1. We remark that Algorithm 1 is a single loop algorithm that does not require solving any subproblem.

Next, we proceed to the convergence analysis of Algorithm 1, specifically investigating the non-asymptotic convergence properties of the sequence $\{\mathbf{z}^k\}$ generated by Algorithm 1. By leveraging the reformulation in Lemma 2.1 and the definition of $p$, it follows that $p(\mathbf{x}, \boldsymbol{\lambda}, \mathbf{r}, \boldsymbol{\xi}, s) \geq 0$ and no interior point exists for the feasible set [98, 59, 22]. In this case, the classical KKT condition for nonsmooth constrained optimization [76] are unsuitable for our analysis. Instead, we adopt the approximation KKT conditions introduced in [4]. We denote merit functions below,

$$\phi_{res}^k(\mathbf{z}) := \text{dist}\left(0, \nabla_{\mathbf{z}} F_k(\mathbf{z}) + \mathcal{N}_{\mathcal{K}}(\mathbf{z})\right), \tag{16}$$

$$\phi_{fea}(\mathbf{z}) := \max\{p(\mathbf{x}, \boldsymbol{\lambda}, \mathbf{r}, \boldsymbol{\xi}, s), \|A_t \boldsymbol{\xi} + \boldsymbol{\rho}\|^2\}. \tag{17}$$

The residual function $\phi_{res}^k(\mathbf{z})$ quantifies the stationarity for (8), because $\phi_{res}^k(\mathbf{z}) = 0$ if and only if $\mathbf{z}$ is a stationary point of (8). Meanwhile, the function $\phi_{fea}(\mathbf{z})$ is interpreted as a feasibility measure for the penalized constraints of problem (6) [67]. Indeed, the merit functions in (16) and (17) are associated with the reformulation (5). Combined with the structure of BLO, $\phi_{fea}(\mathbf{z})$ regulates optimality conditions of LL problem of (3). We clarify corresponding conclusions in Proposition C.1.

**Theorem 3.5.** *Suppose Assumptions 3.1 and 3.2 hold. If the step size $\{e_k\}$ in Algorithm 1 satisfies $0 < e_k \leq \min\{\frac{1}{\alpha_L + \beta_k \|A_t\|_2^2 \alpha_p}, \frac{1}{\beta_k}, \frac{1}{\beta_k(\alpha_d + \|A_t\|_2^2)}\}$, the sequence $\{\mathbf{z}^k\}$ generated by Algorithm 1 satisfies*

$$\min_{0 \leq k \leq K} \phi_{res}^k(\mathbf{z}^{k+1}) = \mathcal{O}\left(\frac{1}{K^{\frac{1}{2}-p}}\right).$$

*Furthermore, if the sequence $\{F_k(\mathbf{z}^k)\}$ is bounded, then it holds that*

$$0 \leq \min_{0 \leq k \leq K} \phi_{fea}(\mathbf{z}^k) = \mathcal{O}\left(\frac{1}{K^p}\right).$$

We remark that boundedness assumptions on $\{F_k(\mathbf{z}^k)\}$ are widely adopted in relevant literature [95, 94, 62]. We provide explanations, proofs and more details in Appendix C.3.

## 3.2 Nonseparable Regularizers

In this subsection, we focus on developing a first-order algorithm for solving (3) in scenarios involving multiple interacting regularizers. These cases arise when the LL problem of (3) incorporates multiple regularization terms applied to the entire vector $\mathbf{x}$, such as elastic net or sparse group Lasso. Our discussion centers on addressing the penalized formulation in this setting, leveraging the reformulation (8). Using the definitions of $\mathcal{K}_i$ and $\mathcal{K}_i^d$ from (9), the constraints of (8) can be written as

$$(\mathbf{x}, \mathbf{r}) \in \mathcal{K}_i, (\boldsymbol{\rho}_i, \lambda_i) \in \mathcal{K}_i^d, \ i = 1, 2, ..., M,$$
$$(\mathbf{x}, \mathbf{r}) \in \mathcal{K}_{M+1}, (\boldsymbol{\rho}_{M+1}, \lambda_{M+1}, s) \in \mathcal{K}_{M+1}^d,$$

which can be further expressed as

$$(\mathbf{x}, \mathbf{r}) \in \mathcal{K}_1 \cap \cdots \cap \mathcal{K}_{M+1}, \ (\boldsymbol{\rho}, \boldsymbol{\lambda}, s) \in \mathcal{K}_1^d \times \cdots \times \mathcal{K}_{M+1}^d. \tag{18}$$

We denote $\mathcal{K}_*^d := \mathcal{K}_1^d \times \cdots \times \mathcal{K}_{M+1}^d$. (18) can be equivalently expressed as

$$\mathbf{z} \in (\mathcal{K}_1 \cap \cdots \cap \mathcal{K}_{M+1}) \times \mathcal{K}_1^d \times \cdots \mathcal{K}_{M+1}^d = (\mathcal{K}_1 \cap \cdots \cap \mathcal{K}_{M+1}) \times \mathcal{K}_*^d.$$

Since each $\mathcal{K}_i^d$ is projection-friendly, the product set $\mathcal{K}_*^d$ inherits this property. In contrast, the intersection $\cap_{i=1}^{M+1} \mathcal{K}_i$ defined over the shared variable $(\mathbf{x}, \mathbf{r})$ may not be projection-friendly. Although

projection onto such intersections has been studied [6, 58], the required iterations are often complex. To address this, we reformulate the constraint to avoid direct projection onto the intersection:

$$\mathbf{z} \in \mathcal{K}_i \times \mathcal{K}_*^d, \ i = 1, 2, ..., M + 1. \tag{19}$$

For each $i$, since both $\mathcal{K}_*^d$ and $\mathcal{K}_i$ are projection-friendly, the product set $\mathcal{K}_i \times \mathcal{K}_*^d$ is also projection-friendly. Consequently, we introduce auxiliary variables $\mathbf{u}_i$ for constraints (19), leading to the following reformulation of (8):

$$\min_{\mathbf{z}, \mathbf{u}} \quad \frac{1}{\beta_k} F_k(\mathbf{z}) \quad \text{s.t.} \quad \mathbf{z} = \mathbf{u}_i, \ \mathbf{u}_i \in \mathcal{K}_i \times \mathcal{K}_*^d, \ i = 1, ..., M + 1, \tag{20}$$

where $\mathbf{u} = (\mathbf{u}_1, ..., \mathbf{u}_{M+1})$. We define the indicator function as $g_i(\mathbf{z}) = I_{\mathcal{K}_i \times \mathcal{K}_*^d}(\mathbf{z}), i = 1, 2, ..., M + 1$. The augmented Lagrangian function of problem (20) is given by:

$$\mathcal{L}_\gamma^k(\mathbf{z}, \mathbf{u}, \boldsymbol{\mu}) = \frac{1}{\beta_k} F_k(\mathbf{z}) + \sum_{i=1}^{M+1} g_i(\mathbf{u}_i) + \sum_{i=1}^{M+1} \langle \boldsymbol{\mu}_i, \mathbf{u}_i - \mathbf{z} \rangle + \frac{\gamma}{2} \sum_{i=1}^{M+1} \|\mathbf{u}_i - \mathbf{z}\|^2,$$

where $\boldsymbol{\mu} := (\boldsymbol{\mu}_1, ..., \boldsymbol{\mu}_{M+1})$ denotes the Lagrangian multiplier associated with constraint $\mathbf{z} = \mathbf{u}_i$. Based on $\mathcal{L}_\gamma^k(\mathbf{z}, \mathbf{u}, \boldsymbol{\mu})$, we adopt an alternative approach to solve (20) inspired by the core idea of the Alternating Direction Method of Multipliers (ADMM). This method alternates between updating primal variables $\mathbf{z}$ and $\mathbf{u}$ in separate subproblems, followed by a dual ascent step to update $\boldsymbol{\mu}$. At the $k$-th iteration, we update $\mathbf{z}$ by performing a gradient step with given $\mathbf{z}^k$:

$$\mathbf{z}^{k+1} = \mathbf{z}^k - e_k \mathbf{d}_{\mathbf{z}}^k, \tag{21}$$

where the update direction $\mathbf{d}_{\mathbf{z}}^k$ corresponds to the gradient of $\mathcal{L}_\gamma^k$ with respect to $\mathbf{z}$ evaluated at $(\mathbf{z}^k, \mathbf{u}^k, \boldsymbol{\mu}^k)$ and $e_k$ is the step size of $k$-th iteration. This is equivalent to minimize the proximal subproblem of $\mathcal{L}_\gamma^k$:

$$\mathbf{z}^{k+1} = \arg\min_{\mathbf{z}} \left\{ \mathcal{L}_\gamma^k(\mathbf{z}^k, \mathbf{u}^k, \boldsymbol{\mu}^k) + \langle \nabla_z \mathcal{L}_\gamma^k(\mathbf{z}^k, \mathbf{u}^k, \boldsymbol{\mu}^k), \mathbf{z} - \mathbf{z}^k \rangle + \frac{1}{2e_k} \|\mathbf{z} - \mathbf{z}^k\|^2 \right\}.$$

Next, for the $\mathbf{u}$-subproblem, we update $\mathbf{u}_i$ by minimizing $\mathcal{L}_\gamma^k$ with respect to $\mathbf{u}_i$ as

$$\mathbf{u}_i^{k+1} = \arg\min_{\mathbf{u}_i} \left\{ g_i(\mathbf{u}_i) + \frac{\gamma}{2} \|\mathbf{u}_i - \mathbf{z}^{k+1} + \frac{\boldsymbol{\mu}_i^k}{\gamma}\|^2 \right\}, \tag{22}$$

which is equivalent to performing the direct projection onto $\mathcal{K}_i \times \mathcal{K}_*^d$, yielding:

$$\mathbf{u}_i^{k+1} = \text{proj}_{\mathcal{K}_i \times \mathcal{K}_*^d}(\frac{\boldsymbol{\mu}_i^k}{\gamma} - \mathbf{z}^{k+1}), \ i = 1, ..., M + 1. \tag{23}$$

Finally, for the dual multipliers $\boldsymbol{\mu}_i$, we update them as

$$\boldsymbol{\mu}_i^{k+1} = \boldsymbol{\mu}_i^k + \gamma(\mathbf{u}_i^{k+1} - \mathbf{z}^{k+1}), \ i = 1, ..., M + 1. \tag{24}$$

---

**Algorithm 2** Alternating approaches for (20)

---

1: Input $\boldsymbol{\lambda}^0, \boldsymbol{\xi}^0$, sequences $\{\beta_k\}, \{e_k\}$, a constant $\gamma$. Initialize $\mathbf{x}^0, \mathbf{r}^0, \boldsymbol{\rho}_i^0, s^0$, set $\mathbf{u}_i^0 = \mathbf{z}^0$.
2: **for** $k = 0, 1, 2, ...$ **do**
3:     Update $\mathbf{z}^{k+1}$ with (21).
4:     Update $\mathbf{u}^{k+1}$ with (23).
5:     Update $\boldsymbol{\mu}^{k+1}$ with (24).
6: **end for**

---

The penalty parameter is updated as $\beta_k = \beta(1 + k)^p$, where $\beta > 0$ is a constant and $0 < p < \frac{1}{2}$. The initialization of Algorithm 2 is also detailed in Appendix C.1. We remark that Algorithm 2 differs from standard ADMM or DRS in two key aspects: (i) the augmented Lagrangian $\mathcal{L}_\gamma^k$ varies with the iteration-dependent parameter $\beta_k$. (ii) instead of exactly minimizing $\mathcal{L}_\gamma^k$ in the $\mathbf{z}$-subproblem, we adopt its first-order approximation at $\mathbf{z}^k$. The strategy is commonly employed in gradient-based alternating minimization approaches [1, 17].

In the following, we discuss the convergence property of Algorithm 2. Similar to the analysis for Algorithm 1, we utilize the stationarity and feasibility measure commonly used in penalty methods [67, 95, 94]. We define the following merit functions in the same arguments as (16) and (17):

$$\phi_{res}^k(\mathbf{z}) := \text{dist}\left(0, \nabla F_k(\mathbf{z}) + \mathcal{N}_{\mathcal{K}}(\mathbf{z})\right), \tag{25}$$

$$\phi_{fea}(\mathbf{z}) := \max\{p(\mathbf{x}, \boldsymbol{\lambda}, \mathbf{r}, \boldsymbol{\xi}, s), \|A_t\boldsymbol{\xi} + \sum_{i=1}^{M+1} \boldsymbol{\rho}_i\|^2\}, \tag{26}$$

where $\mathcal{K} := (\mathcal{K}_1 \cap \cdots \cap \mathcal{K}_{M+1}) \times \mathcal{K}_1^d \times \cdots \mathcal{K}_{M+1}^d$. Based on the above functions, we establish the convergence results for Algorithm 2 in Theorem 3.7. In pursuit of this, we make the following assumption, which is popularly employed in ADMM approaches [90, 8, 81, 26].

**Assumption 3.6.** The sequence $\{\boldsymbol{\mu}^k\}$ is bounded and satisfies $\sum\limits_{k=1}^{\infty} \|\boldsymbol{\mu}^{k+1} - \boldsymbol{\mu}^k\|^2 \leq \infty$.

**Theorem 3.7.** *Suppose Assumptions 3.1, 3.2 and 3.6 hold. If the step sizes in Algorithm 2 satisfy* $0 < \underline{e} \leq e_k < \min\{\frac{\beta_k}{\alpha_L + \beta_k\|A_t\|_2^2\alpha_p}, \frac{1}{\alpha_d + \|A_t\|_2^2}, 1\}$, *the sequence* $\{\mathbf{z}^k\}$ *generated by Algorithm 2 satisfies* $\lim_{k\to\infty} \phi_{res}^k(\mathbf{z}^{k+1}) = 0$. *Furthermore, if the sequence* $\{F_k(\mathbf{z}^k)\}$ *is bounded, then it holds that* $\lim_{k\to\infty} \phi_{fea}(\mathbf{z}^k) = 0$.

Note that the lower bound $\underline{e}$ for step sizes $e_k$ is commonly utilized in single-loop Hessian-free algorithms for BLO [94, 95, 62]. We provide the detailed proof for Theorem 3.7 in Appendix C.4.

# 4 Numerical Experiments

In this section, we evaluate the numerical performance of our proposed LDPM through experiments on both synthetic and real datasets. Specifically, we compare LDPM with several existing hyperparameter optimization algorithms under the BLO framework (3), including search methods, TPE [13], IGJO [31], IFDM [14, 15], VF-iDCA [36], LDMMA [22], BiC-GAFFA [94], as detailed in Appedix D.1.

We consider all hyperparameter optimization problems listed in Table 1. Performance is evaluated using validation and test errors based on the obtained LL minimizers, as well as the total running time. These metrics are standard in the evaluation of bilevel hyperparameter optimization algorithms [36, 31]. For each problem, we perform experiments across various data settings or datasets with 10 repetitions, and report the aggregated statistical results. Depending on the regularization structure of each problem, we apply either Algorithm 1 or 2, as detailed in Section 3.

## 4.1 Experiments on synthetic data

We focus on two prototypical tasks built from simple synthetic data: least squares regression with various Lasso-type regularizers and low-rank matrix completion, as listed in Table 1. The synthetic data consists of observation matrices sampled from specific distributions and response vectors generated with controlled noise. The detailed data generation process is provided in Appendix D.2.

**Lasso-type Regression.** We consider three regularizers: elastic net [100], group Lasso [99], and sparse group Lasso [83]. These formulations all promote sparsity while balancing model complexity and predictive accuracy. Table 2 presents the statistical results for the sparse group Lasso problem, including validation error, test error, and running time. Results for the elastic net and group Lasso problems are reported in Tables 3 and 4, respectively. Detailed experimental settings for each method are provided in the corresponding subsections of Appendix D.2. Overall, LDPM demonstrates superior performance on synthetic data, consistently achieving the lowest test errors while requiring the least computational time compared to baseline methods.

**Low-rank matrix completion.** For this problem, we conduct the numerical experiments on $60 \times 60$ matrices [36, 31]. The data generation process, detailed statistical results, and corresponding analysis are presented in Appendix D.2.4.

**Sensitivity of parameters.** We conduct sensitivity experiments on both Algorithm 1 and Algorithm 2. The results summarized in Table 6 show that both algorithms exhibit stable convergence across various parameter settings.

Table 2: Sparse group Lasso problems on synthetic data, where $p$ represents the number of features.

| Settings | $p = 600$ | | | $p = 1200$ | | |
|---|---|---|---|---|---|---|
| | Time(s) | Val. Err. | Test Err. | Time(s) | Val. Err. | Test Err. |
| Grid | $6.36 \pm 1.88$ | $84.73 \pm 5.29$ | $87.34 \pm 15.91$ | $13.68 \pm 2.49$ | $84.68 \pm 4.31$ | $86.00 \pm 18.43$ |
| Random | $6.02 \pm 2.01$ | $135.17 \pm 5.95$ | $147.43 \pm 25.54$ | $12.64 \pm 2.84$ | $137.87 \pm 14.21$ | $146.25 \pm 15.52$ |
| IGJO | $1.58 \pm 0.28$ | $101.93 \pm 4.07$ | $96.36 \pm 13.72$ | $7.35 \pm 1.46$ | $130.56 \pm 14.02$ | $106.70 \pm 4.01$ |
| VF-iDCA | $0.56 \pm 0.15$ | $56.96 \pm 5.58$ | $76.84 \pm 11.33$ | $8.63 \pm 2.91$ | $86.38 \pm 6.40$ | $87.58 \pm 8.90$ |
| LDMMA | $0.57 \pm 0.13$ | $82.70 \pm 5.03$ | $72.44 \pm 14.72$ | $4.72 \pm 2.15$ | $83.93 \pm 7.32$ | $84.03 \pm 9.08$ |
| BiC-GAFFA | $0.39 \pm 0.02$ | $67.42 \pm 6.28$ | $71.45 \pm 10.74$ | $2.52 \pm 0.29$ | $82.21 \pm 5.03$ | $79.81 \pm 7.66$ |
| LDPM | $\mathbf{0.35 \pm 0.03}$ | $65.11 \pm 6.62$ | $\mathbf{69.48 \pm 9.40}$ | $\mathbf{2.15 \pm 0.14}$ | $81.39 \pm 6.51$ | $\mathbf{78.11 \pm 6.35}$ |

| Settings | $p = 2400$ | | | $p = 4800$ | | |
|---|---|---|---|---|---|---|
| | Time(s) | Val. Err. | Test Err. | Time(s) | Val. Err. | Test Err. |
| Grid | $24.23 \pm 4.05$ | $95.63 \pm 14.13$ | $84.86 \pm 15.09$ | $47.09 \pm 6.34$ | $128.94 \pm 24.11$ | $115.41 \pm 17.62$ |
| Random | $22.17 \pm 6.85$ | $120.04 \pm 15.36$ | $146.77 \pm 16.70$ | $46.3 \pm 5.57$ | $99.41 \pm 16.55$ | $122.49 \pm 19.46$ |
| IGJO | $11.14 \pm 7.44$ | $91.59 \pm 14.97$ | $115.98 \pm 14.94$ | $29.76 \pm 9.44$ | $99.75 \pm 15.14$ | $106.49 \pm 7.48$ |
| VF-iDCA | $14.31 \pm 1.45$ | $63.21 \pm 5.36$ | $81.92 \pm 10.54$ | $45.12 \pm 3.10$ | $73.66 \pm 10.53$ | $96.09 \pm 9.14$ |
| LDMMA | $7.50 \pm 0.21$ | $66.23 \pm 7.47$ | $79.09 \pm 13.75$ | $36.14 \pm 3.65$ | $78.61 \pm 12.32$ | $95.81 \pm 9.43$ |
| BiC-GAFFA | $5.11 \pm 0.10$ | $86.83 \pm 13.53$ | $76.38 \pm 8.60$ | $5.03 \pm 0.63$ | $94.34 \pm 8.19$ | $92.05 \pm 7.13$ |
| LDPM | $\mathbf{4.87 \pm 0.05}$ | $92.32 \pm 6.62$ | $\mathbf{74.14 \pm 2.79}$ | $\mathbf{4.58 \pm 0.17}$ | $91.35 \pm 6.04$ | $\mathbf{90.21 \pm 5.74}$ |

## 4.2 Experiments on real-world data

To assess the robustness of our algorithm in practical settings, we conduct experiments on real-world datasets that are larger and exhibit more complex sampling distributions. Specifically, we consider experiments on elastic net, smoothing support vector machine and sparse logistic regression, as listed in Table 1. All datasets are drawn from the LIBSVM repository[3] [21]. For each repetition, we randomly shuffle and split the data into training, validation and test sets.

**Elastic Net.** In this part, we conduct experiments on datasets gisette [43] and sensit [30]. We summarize the comparative experimental results in Table 7 and show the validation and test error curves over time for each algorithm in Figure 1. Even in these high-dimensional settings, LDPM delivers competitive accuracy while maintaining fast convergence. Additional experimental details are provided in Appendix D.4.1.

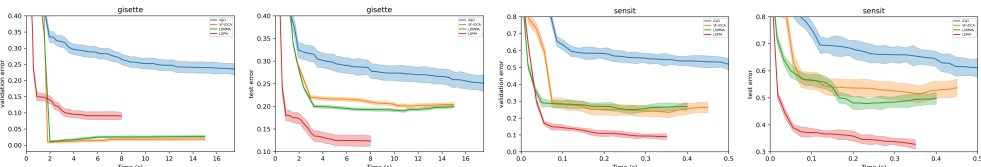

Figure 1: Comparison of the algorithms on Elastic Net problem for real-world datasets.

**Smoothed Support Vector Machine.** In this part, we perform 6-fold cross-validation using medical statistics datasets, including diabetes, sonar, a1a [7]. Details of the datasets and experimental setup are given in Appendix D.4.2. We plots the validation and test errors of each algorithm over time in Figure 2, which clearly shows that LDPM converges more rapidly and achieves lower error levels than the competing methods.

**Sparse Logistic Regression.** In this part, we conduct experiments on three large-scale document classification datasets, news20.binary, rcv1.binary and real-sim. Dataset characteristics and experimental details are provided in Appendix D.4.3. In this experiment, we compare LDPM with search methods, IFDM and BiC-GAFFA. We plot the validation and test error curves over time in Figure 3 and report the corresponding final validation and test accuracies in Table 9 for comparison. LDPM consistently converges faster and achieves the lowest validation and test errors.

## 5 Conclusion

In this paper, we introduce a penalty framework based on lower-level duality for bilevel hyperparameter optimization. Notably, we solve the penalized problem using single-loop first-order algorithms. Theoretically, we establish convergence guarantees for the proposed algorithms. Empirically, through numerical experiments on both synthetic and real-world datasets, our methods exhibit superior performance compared to existing approaches, particularly among the illustrated HO examples.

---

[3]https://www.csie.ntu.edu.tw/ cjlin/libsvmtools/datasets/

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

## A Proofs for Section 2

In this subsection, we provide the proofs of the results concerning the penalty framework in Section 2.

### A.1 Proof of Lemma 2.1

The following proof follows [22].

*Proof.* We prove the conclusion based on the formulation (3). First we introduce augmented variables $\mathbf{z}$ and $\mathbf{z}_i, i = 1, 2, ..., M + 1$ and deduce the equivalent form of LL problem of (3),

$$\min_{\mathbf{x}, \mathbf{z}_i} \varphi(\mathbf{z}) + \sum_{i=1}^{M+1} \lambda_i R_i(\mathbf{z}_i) \quad \text{s.t. } \mathbf{z} = A_t \mathbf{x} - \mathbf{b}_t, \ \mathbf{x} = \mathbf{z}_i, \ i = 1, 2, ..., M + 1 \tag{27}$$

Since $l, R_i$ are convex and the constraints are affine, strong duality holds under Slater's condition. If $\text{ri}(\text{dom } l \cap (\cap_{i=1}^{M+1} \text{dom } R_i)) \neq \emptyset$, then (27) is equivalent to its Lagrangian dual problem:

$$\max_{\boldsymbol{\xi}, \boldsymbol{\rho}} \min_{\mathbf{x}, \mathbf{z}, \mathbf{z}_i} \varphi(\mathbf{z}) + \sum_{i=1}^{M+1} \lambda_i R_i(\mathbf{z}_i) - \boldsymbol{\xi}^T (A_t \mathbf{x} - \mathbf{b}_t - \mathbf{z}) + \sum_{i=1}^{M+1} \boldsymbol{\rho_i}^T (\mathbf{x} - \mathbf{z}_i),$$

where $\boldsymbol{\xi}$ is Lagrangian multiplier of constraint $A_t \mathbf{x} - \mathbf{b}_t = \mathbf{z}$, while $\boldsymbol{\rho}_i$ are those associated with constraints $\mathbf{x} = \mathbf{z}_i$. By adding the negative signs, we obtain

$$\max_{\boldsymbol{\xi}, \boldsymbol{\rho}} - \max_{\mathbf{x}, \mathbf{z}, \mathbf{z}_i} -\varphi(\mathbf{z}) - \sum_{i=1}^{M+1} \lambda_i R_i(\mathbf{z}_i) + \boldsymbol{\xi}^T (A_t \mathbf{x} - \mathbf{b}_t - \mathbf{z}) - \sum_{i=1}^{M+1} \boldsymbol{\rho_i}^T (\mathbf{x} - \mathbf{z}_i).$$

The above problem can be further simplified as,

$$\max_{\boldsymbol{\xi}, \boldsymbol{\rho}} \quad -\varphi^*(\boldsymbol{\xi}) - \sum_{i=1}^{M+1} \lambda_i R_i^*(\tfrac{\boldsymbol{\rho}_i}{\lambda_i}) - \boldsymbol{\xi}^T \mathbf{b}_t.$$
$$\text{s.t.} \quad A_t \boldsymbol{\xi} + \sum_{i=1}^{M+1} \boldsymbol{\rho}_i = \mathbf{0}. \tag{28}$$

Meanwhile, leveraging the value function of the lower-level problem, the constraint of (3) is equivalent to

$$l(\mathbf{x}) + \sum_{i=1}^{M+1} \lambda_i R_i(\mathbf{x}) \leq \min_{\mathbf{x}} \{l(\mathbf{x}) + \sum_{i=1}^{M+1} \lambda_i R_i(\mathbf{x})\}. \tag{29}$$

From the equivalence of (27) and (28), (29) is further equivalent to

$$l(\mathbf{x}) + \sum_{i=1}^{M+1} \lambda_i R_i(\mathbf{x}) \leq \max_{\boldsymbol{\xi}, \boldsymbol{\rho}} \{-\varphi^*(\boldsymbol{\xi}) - \sum_{i=1}^{M+1} \lambda_i R_i^*(\tfrac{\boldsymbol{\rho}_i}{\lambda_i}) - \boldsymbol{\xi}^T \mathbf{b}_t \mid A_t \boldsymbol{\xi} + \sum_{i=1}^{M+1} \boldsymbol{\rho}_i = \mathbf{0}\}. \tag{30}$$

Because the inequality in (30) holds if and only if there exists a feasible pair $(\boldsymbol{\xi}, \boldsymbol{\rho})$ satisfying (30), dropping the max operator, we obtain that the constraint in (3) is equivalent to

$$l(\mathbf{x}) + \sum_{i=1}^{M+1} \lambda_i R_i(\mathbf{x}) + \varphi^*(\boldsymbol{\xi}) + \sum_{i=1}^{M+1} \lambda_i R_i^*(\frac{\boldsymbol{\rho_i}}{\lambda_i}) + \boldsymbol{\xi}^T \mathbf{b}_t \leq 0,$$

$$A_t \boldsymbol{\xi} + \sum_{i=1}^{M+1} \boldsymbol{\rho}_i = \mathbf{0}.$$

We complete the proof. $\qquad\qquad\square$

## A.2  Proof of Theorem 2.4

*Proof.* We adopt the convention $A(\mathbf{z}) = \frac{1}{2}\|A_t\boldsymbol{\xi} + \sum_{i=1}^{M+1} \boldsymbol{\rho}_i\|^2$. It is straightforward that $A(\mathbf{z}) \geq 0$. Let $\bar{\mathbf{z}}$ be any limit point of the sequence $\{\mathbf{z}^k\}$ and $\{\mathbf{z}^{j_k}\} \subset \{\mathbf{z}^k\}$ be the subsequence such that $\mathbf{z}^{j_k} \to \bar{\mathbf{z}}$.

Assume that $\mathbf{z}^*$ is a solution of the reformulation (6). Then it holds that $L(\mathbf{x}^*) \leq L(\mathbf{x})$ for all $\mathbf{z} = (\mathbf{x}, \boldsymbol{\lambda}, \boldsymbol{\rho}, \mathbf{r}, \boldsymbol{\xi}, s)$ feasible to (6). Note that any point $\mathbf{z}$ feasible to (6) is also feasible to (8).

Since $\mathbf{z}^{k+1}$ is the minimizer of the problem (8) with $\beta_k$, it follows that

$$L(\mathbf{x}^{k+1}) + \beta_k(p(\mathbf{z}^{k+1}) + A(\mathbf{z}^{k+1})) \overset{(a)}{\leq} L(\mathbf{x}^*) + \beta_k(p(\mathbf{z}^*) + A(\mathbf{z}^*)) \overset{(b)}{\leq} L(\mathbf{x}^*), \qquad (31)$$

where $(a)$ follows from the feasibility of $\mathbf{z}^{k+1}$ and $\mathbf{z}^*$ for the penalized problem (8) and the optimality of $\mathbf{z}^{k+1}$, $(b)$ holds because $\mathbf{z}^*$ is feasible to (6). From (31), we deduce

$$p(\mathbf{z}^{k+1}) + A(\mathbf{z}^{k+1}) \leq \frac{1}{\beta_k}(L(\mathbf{x}^*) - L(\mathbf{x}^{k+1})).$$

Since the functions $L, p$ and $A$ are lower semi-continuous in $\mathbf{z}$, letting $k = k_j$ and taking the limit $j \to \infty$ for the above inequality, we have $p(\bar{\mathbf{z}}) + A(\bar{\mathbf{z}}) \leq 0$ with $\beta_k \to \infty$.

Since the assumptions of Theorem 2.4 are consistent with those of Lemma 2.1, we obtain the following relation from the formulation of $p$ in (7)

$$p(\mathbf{z}) = l(\mathbf{x}) + \sum_{i=1}^{M+1} \lambda_i R_i(\mathbf{x}) - \min_{\mathbf{x}}\{l(\mathbf{x}) + \sum_{i=1}^{M+1} \lambda_i R_i(\mathbf{x})\},$$

which directly implies that $p(\mathbf{z}) \geq 0$. Combined with $A(\mathbf{z}) \geq 0$ for all $\mathbf{z}$, we further deduce that $p(\bar{\mathbf{z}}) = 0$ and $A(\bar{\mathbf{z}}) = 0$. Therefore, $\bar{\mathbf{z}}$ is feasible for (6). Since $\mathbf{z}^*$ is optimal for (6), it holds that $L(\mathbf{x}^*) \leq L(\bar{\mathbf{x}})$.

Letting $k = k_j$ and taking the limit $j \to \infty$ for (31), we have $L(\bar{\mathbf{x}}) \leq L(\mathbf{x}^*)$. Hence, we deduce that $L(\bar{\mathbf{x}}) = L(\mathbf{x}^*)$ and $\bar{\mathbf{z}}$ is also an optimal solution of (6). This completes the proof. $\qquad\square$

## A.3  Conjugate functions for problems listed in Table 1

we calculate the closed-form expression of the conjugate functions of $\varphi$ in problems as follows:

For **least squares loss**, $\varphi^*(v) = \frac{1}{2}v^2$.

For **smoothed hinge loss**, $\varphi^*(v) = \frac{1}{2}v^2 + v$ if $-1 < v < 0$ and $\varphi^*(v) = \infty$ otherwise.

For **logistic loss**, $\varphi^*(v) = -v\log(v) - (1-v)\log(1-v)$ if $0 < v < 1$ and $\varphi^*(v) = \infty$ otherwise.

# B  Epigraphical Projections

In this section, we discuss the projection onto the cones in Algorithms 1 and 2. According to different cases detailed in Section 3.1 and 3.2, we discuss the projections when involving different norm regularizers.

## B.1  Projections Involving Vector Norms

The most commonly used norms in hyperparameter optimization include the $\ell_1$-, $\ell_2$- and $\ell_\infty$-norm, each serving distinct purposes depending on the specific application. When $R_i$ represents a single norm, the explicit forms of $\mathcal{K}_i$ and $\mathcal{K}_i^d$ defined in (9) are expressed as follows.

- $R_i(x) = \|\mathbf{x}\|_1$: $\mathcal{K}_i = \{(\mathbf{x}, r_i) \mid \|\mathbf{x}\|_1 \leq r_i\}$, $\mathcal{K}_i^d = \{(\boldsymbol{\rho}_i, \lambda_i) \mid \|\boldsymbol{\rho}_i\|_\infty \leq \lambda_i\}$.
- $R_i(x) = \|\mathbf{x}\|_2$: $\mathcal{K}_i = \{(\mathbf{x}, r_i) \mid \|\mathbf{x}\|_2 \leq r_i\}$, $\mathcal{K}_i^d = \{(\boldsymbol{\rho}_i, \lambda_i) \mid \|\boldsymbol{\rho}_i\|_2 \leq \lambda_i\}$.
- $R_i(x) = \|\mathbf{x}\|_\infty$: $\mathcal{K}_i = \{(\mathbf{x}, r_i) \mid \|\mathbf{x}\|_\infty \leq r_i\}$, $\mathcal{K}_i^d = \{(\boldsymbol{\rho}_i, \lambda_i) \mid \|\boldsymbol{\rho}_i\|_1 \leq \lambda_i\}$.

Thus, we investigate the projection onto the epigraph $\{(\mathbf{x}, t) \mid \|\mathbf{x}\|_q \leq t\}$ of the $\ell_q$-norm ($q = 1, 2, \infty$). The projection for the $\ell_2$-norm epigraph has a well-known closed-form solution, as detailed below:

**Proposition B.1.** *[10, Example 6.37] Let $L_2^n = \{(\mathbf{x}, t) \mid \|\mathbf{x}\|_2 \leq t\}$, for any $(\mathbf{x}, t) \in \mathbb{R}^n \times \mathbb{R}$, we have*

$$\mathrm{proj}_{L_2^n}((\mathbf{x}, t)) = \begin{cases} (\frac{\|\mathbf{x}\|_2 + t}{2\|\mathbf{x}\|_2}\mathbf{x}, \frac{\|\mathbf{x}\|_2 + t}{2}), & \|\mathbf{x}\|_2 \geq |t|, \\ (\mathbf{0}, 0), & t < \|\mathbf{x}\|_2 < -t, \\ (\mathbf{x}, t), & \|\mathbf{x}\|_2 \leq t. \end{cases}$$

Next, we discuss the $\ell_1$ norm epigraphic projection. We first provide the following theorem on the projection onto epigraphs of convex functions.

**Theorem B.2.** *[10, Theorem 6.36] Let $C = \mathrm{epi}(g) = \{(\mathbf{x}, t) \mid g(\mathbf{x}) \leq t\}$ where $g$ is convex. Then for any $(\mathbf{x}, t) \in \mathbb{R}^n \times \mathbb{R}$, it holds that*

$$\mathrm{proj}_C((\mathbf{x}, t)) = \begin{cases} (\mathbf{x}, t), & g(\mathbf{x}) \leq t, \\ (\mathrm{prox}_{\lambda^* g}(\mathbf{x}), t + \lambda^*), & g(\mathbf{x}) > t, \end{cases}$$

*where $\lambda^*$ is any positive root of the function*

$$\psi(\lambda) = g(\mathrm{prox}_{\lambda g}(\mathbf{x}) - \lambda - t).$$

*In addition, $\psi$ is nonincreasing.*

**Proposition B.3.** *[10, Example 6.38] Let $L_1^n = \{(\mathbf{x}, t) \mid \|\mathbf{x}\|_1 \leq t\}$, for any $(\mathbf{x}, t) \in \mathbb{R}^n \times \mathbb{R}$, we have*

$$\mathrm{proj}_{L_1^n}((\mathbf{x}, t)) = \begin{cases} (\mathbf{x}, t), & \|\mathbf{x}\|_1 \leq t, \\ (\mathcal{T}_{\lambda^*}(\mathbf{x}), t + \lambda^*), & \|\mathbf{x}\|_1 > t, \end{cases}$$

where $\mathcal{T}_\lambda = \mathrm{prox}_{\lambda\|\cdot\|_1}$ denotes the proximal of $\ell_1$-norm, defined as

$$\mathcal{T}_\lambda(y) = [|y| - \lambda]_+ \mathrm{sgn}(y) = \begin{cases} y - \lambda, & y \geq \lambda \\ 0, & |y| < \lambda, \\ y + \lambda, & y \leq -\lambda. \end{cases}$$

Here, $\lambda^*$ is any positive root of the nonincreasing function $\psi(\lambda) = \|\mathcal{T}_\lambda(\mathbf{x})\|_1 - \lambda - s$. In practice, the $\ell_1$ norm epigraphical projection can be computed in linear time using the quick-select algorithm proposed by [88].

Finally, the projection for the $\ell_\infty$ norm epigraph can be computed directly via the Moreau decomposition. Let $L_\infty^n = \{(\mathbf{x}, t) \mid \|\mathbf{x}\|_\infty \leq t\}$, then the projection is given by

$$\mathrm{proj}_{L_\infty^n}(\mathbf{x}, t) = (\mathbf{x}, t) - \mathrm{proj}_{L_1^n}(\mathbf{x}, t).$$

When $R_i$ represents the squared $\ell_2$ norm, the corresponding rotated second-order cones are defined as $\mathcal{K}_{M+1}$ and $\mathcal{K}_{M+1}^d$ in (9). According to Theorem B.2, for any $(\mathbf{x}, t) \in \mathbb{R}^n \times \mathbb{R}$, we have

$$\mathrm{proj}_{\mathcal{K}_{M+1}}(\mathbf{x}, t) = \begin{cases} (\mathbf{x}, t), & \|\mathbf{x}\|_2^2 \leq 2t, \\ (\frac{\mathbf{x}}{1 + \lambda^*}, t + \lambda^*), & \|\mathbf{x}\|_2^2 > 2t, \end{cases}$$

where $\lambda^*$ is any positive root of the nonincreasing function $\psi(\lambda) = (\frac{1}{2}\lambda + t)(1 + 2\lambda^2) - \|x\|_2^2$. Similar to $\ell_1$-norm epigraphic projection, it can also be effectively solved in linear time with quick-select algorithm proposed by [88].

For the rotated second-order cone $\mathcal{K}_{M+1}^d = \{(\boldsymbol{\rho}, \lambda, s) \mid \|\boldsymbol{\rho}\|_2^2 \leq 2\lambda s\}$ where $\boldsymbol{\rho} \in \mathbb{R}^n$, an equivalent representation is given by $\{(\boldsymbol{\rho}, \lambda, s) \mid \|(\boldsymbol{\rho}, \lambda, s)\|_2 \leq \lambda + s\}$. We introduce auxiliary variables $\mathbf{w} = (\boldsymbol{\rho}, \lambda, s) \in \mathbb{R}^{n+2}$ and $t = \lambda + s \in \mathbb{R}$. In this way, the projection onto $\mathcal{K}_{M+1}^d$ for given $(\bar{\boldsymbol{\rho}}, \bar{\lambda}, \bar{s})$ is equivalent to the following optimization problem with $(\bar{\mathbf{w}}, \bar{t})$:

$$\min_{\mathbf{w}, t} \frac{1}{2}\|\mathbf{w} - \bar{\mathbf{w}}\|^2 + \frac{1}{2}(t - \bar{t})^2 \text{ s.t. } \|\mathbf{w}\|_2 \leq t, \mathbf{w}^T \mathbf{c}_0 = t,$$

where $\mathbf{c}_0 = (0, ..., 0, 1, 1) \in \mathbb{R}^{n+2}$. The problem can be solved directly using the analytic solution provided in [53, Proposition 6.4].

When the regularization involves a group component-wise regularizers, i.e., $R_i(\mathbf{x}) = \|\mathbf{x}^{(i)}\|_{(t)}$, where represents the $i$-th subvector of $\mathbf{x}$ with $\mathbf{x} = (\mathbf{x}^{(1)}, ..., \mathbf{x}^{(M)})$, as described in Section 3.1. In this case, we observe that projection onto the set $\mathcal{K}_i$ and $\mathcal{K}_i^d$ corresponds to the $\ell_1$, $\ell_2$ or $\ell_\infty$-norm. The same projection applies to the vector $\boldsymbol{\rho} = (\boldsymbol{\rho}^{(1)}, ..., \boldsymbol{\rho}^{(M)})$.

## B.2 Projections Involving Matrix Norms

Now we study the projection onto the epigraphs of nuclear norm $\|\cdot\|_*$ and spectral norm $\|\cdot\|_{op}$. Since our reformulation relies on conjugate functions and the conjugate of a norm is its dual norm, we need to take both into consideration.

For a matrix $X \in \mathbb{R}^{m \times n}$, the nuclear norm is defined as $\|X\|_* = \sum_{i=1}^{\min\{m,n\}} \sigma_i(X)$ and the spectral norm is defined as $\|X\|_{op} = \max_i \sigma_i(X)$, where $\sigma_i(X)$ is singular values for $X$.

Given a matrix $A \in \mathbb{R}^{m \times n}$ and a scalar $t$, the projection onto the epigraph of the nuclear norm $\{X \in \mathbb{R}^{m \times n}, \tau \geq 0 \mid \|X\|_* \leq \tau\}$ involves solving the following optimization problem

$$\min_{X, \tau \geq 0} \frac{1}{2}\|X - A\|_F^2 + \frac{1}{2}\|t - \tau\|^2 \text{ s.t. } \|X\|_* \leq \tau,$$

where $\|\cdot\|_F$ denotes Frobenius norm of a matrix.

- If $\|A\|_* \leq t$, the point $(A, t)$ already lies in the epigraph and the projection is simply $(X, \tau) = (A, t)$.
- If $\|A\|_* > t$, we first compute the singular value decomposition of $A$ as $A = U\Sigma V$, where $\Sigma = \text{diag}\{\sigma_1, \sigma_2, ..., \sigma_r\}$ is the single value matrix of $A$ and $U \in \mathbb{R}^{m \times r}$, $V \in \mathbb{R}^{n \times r}$. According to [1,Theorem 6.36], the projected matrix is obtained by soft-thresholding the singular values:

$$\bar{\sigma}_i = \max(\sigma_i - \lambda, 0), i = 1, 2, ..., r,$$

where $\lambda$ is determined by the equation $\sum_{i=1}^{r} \max(\sigma_i - \lambda, 0) = t + \lambda$. This equation is typically solved efficiently via a bisection search. Subsequently, we obtain the solution $\tau^* = t + \lambda$ and reconstruct the projected matrix as $X^* = U\bar{\Sigma}V^T$ where $\bar{\Sigma} = \text{diag}\{\bar{\sigma}_1, \bar{\sigma}_2, ..., \bar{\sigma}_r\}$. The projected pair $(X^*, \tau^*)$ is the closest point to $(A, t)$ in the epigraph of the nuclear norm.

Given a matrix $A \in \mathbb{R}^{m \times n}$ and a scalar $t$, now we consider projection onto the epigraph of the nuclear norm $\{X \in \mathbb{R}^{m \times n}, \tau \geq 0 \mid \|X\|_{op} \leq \tau\}$

- If $\|A\|_{op} \leq t$, the point $(A, t)$ already lies in the epigraph and the projection is simply $(X, \tau) = (A, t)$.
- If $\|A\|_{op} > t$, we first compute the singular value decomposition of $A$ as $A = U\Sigma V$, where $\Sigma = \text{diag}\{\sigma_1, \sigma_2, ..., \sigma_r\}$ is the single value matrix of $A$ and $U \in \mathbb{R}^{m \times r}$, $V \in \mathbb{R}^{n \times r}$.

  Since the epigraph of the spectral norm is defined by the constraint $\|X\|_{op} = \max_i \sigma_i(X) \leq \tau$, we need to adjust the singular values so that the largest does not exceed the new scalar $\tau^*$ as

$$\tilde{\sigma}_i = \min\{\sigma_i, \tau^*\} \quad \text{for } i = 1, 2, \ldots, r.$$

  To determine $\tau^*$, we solve the one-dimensional optimization problem

$$\min_{\tau \geq 0} \frac{1}{2} \sum_{i: \sigma_i > \tau} (\sigma_i - \tau)^2 + \frac{1}{2}(\tau - t)^2.$$

  In practice, the optimal $\tau^*$ can be efficiently computed using a bisection search.

  Subsequently, we reconstruct the projected matrix as $X^* = U\tilde{\Sigma}V^T$ where $\tilde{\Sigma} = \text{diag}\{\tilde{\sigma}_1, \tilde{\sigma}_2, \ldots, \tilde{\sigma}_r\}$.

  The projected pair $(X^*, \tau^*)$ is the closest point to $(A, t)$ in the epigraph of the spectral norm.

682 From the above discussions, it is evident that the projections can be computed efficiently.

# C Explanations and proofs for Section 3

684 In this section, we provide additional explanations and the proofs for the convergence results of our
685 proposed algorithms in Section 3.

## C.1 Initialization of Algorithm 1 and 2

687 We initialize the starting point by following the algorithms for BLO proposed in [36, 22, 95]. For
688 Algorithm 1, given the input $\boldsymbol{\lambda}^0, \boldsymbol{\xi}^0$, we initialize $\mathbf{x}^0$ by solving the LL problem of (3). The remaining
689 initial variables are set as $r_i^0 = R_i(\mathbf{x})$, $\boldsymbol{\rho}^0 = -\nabla l(\mathbf{x}^0)$ and $s^0 = \|\boldsymbol{\rho}^0\|^2/2\lambda_1^0$. For Algorithm 2, given
690 the input $\boldsymbol{\lambda}^0, \boldsymbol{\xi}^0$, we also initialize $\mathbf{x}^0$ with solving the LL problem of (3). The other initial variables
691 are set as $r_i^0 = R_i(\mathbf{x}^0)$, $\boldsymbol{\rho}_i^0 = -\frac{1}{M+1}A_t\boldsymbol{\xi}^0$ and $s^0 = \|\boldsymbol{\rho}_{M+1}^0\|^2/2\lambda_{M+1}^0$.

692 This initialization strategy ensures a feasible starting point for the corresponding reformulation of
693 original BLO, thereby facilitating convergence and enhancing the overall efficiency of the optimization
694 process.

## C.2 Explanations for Merit Functions

696 To initiate the proof of the convergence results, we establish the rationale for selecting $\phi_{res}^k$ and $\phi_{fea}$
697 as the merit measures. Note that $\phi_{res}^k$ and $\phi_{fea}$ in Section 3.1 and 3.2 are both defined based on the
698 penalized formulation (8) within a unified framework as follows:

$$\phi_{res}^k(\mathbf{z}) := \text{dist}\left(0, \nabla_{\mathbf{z}} F_k(\mathbf{z}) + \mathcal{N}_{\mathcal{K}}(\mathbf{z})\right), \tag{32}$$

699

$$\phi_{fea}(\mathbf{z}) := \max\{p(\mathbf{x}, \boldsymbol{\lambda}, \mathbf{r}, \boldsymbol{\xi}, s), \|A_t\boldsymbol{\xi} + \sum_{i=1}^{M+1} \boldsymbol{\rho}_i\|\}, \tag{33}$$

700 where $\mathcal{K} = (\mathcal{K}_1 \cap \cdots \cap \mathcal{K}_{M+1}) \times \mathcal{K}_1^d \times \cdots \mathcal{K}_{M+1}^d$. For the case of single-round global regularization
701 discussed in Section 3.1, the set $\mathcal{K}$ reduces to $\mathcal{K} = \mathcal{K}_1 \times \mathcal{K}_1^d$ and $(\boldsymbol{\rho}_1, ..., \boldsymbol{\rho}_{M+1})$ is replaced by a
702 single $\boldsymbol{\rho}$.

703 From Lemma 2.1, we know that (5) is a direct reformulation of (3). For convenience, we simplify the
704 left hand of the first constraint as:

$$F(\mathbf{x}, \boldsymbol{\lambda}, \boldsymbol{\rho}, \boldsymbol{\xi}) = l(\mathbf{x}) + \sum_{i=1}^{M+1} \lambda_i R_i(\mathbf{x}) + \varphi^*(\boldsymbol{\xi}) + \sum_{i=1}^{M+1} \lambda_i R_i^*(\frac{\boldsymbol{\rho_i}}{\lambda_i}) + \boldsymbol{\xi}^T\mathbf{b}_t.$$

705 Similar to (8), we construct the penalized formulation for (5) as follows,

$$\min_{\mathbf{z}} \ L(\mathbf{x}) + \beta_k F(\mathbf{x}, \boldsymbol{\lambda}, \boldsymbol{\rho}, \boldsymbol{\xi}) + \frac{\beta_k}{2}\|A_t\boldsymbol{\xi} + \sum_{i=1}^{M+1} \boldsymbol{\rho}_i\|^2, \tag{34}$$

706 where $\beta_k$ serves as the penalty parameter.

707 **Proposition C.1.** *If $\phi_{fea}(\mathbf{z}) = 0$, then $(\mathbf{x}, \boldsymbol{\lambda}, \boldsymbol{\rho}, \boldsymbol{\xi})$ is a feasible point to (5). Moreover, if $\phi_{fea}(\mathbf{z}) = 0$*
708 *and $\phi_{res}(\mathbf{z}) = 0$ both hold, then $(\mathbf{x}, \boldsymbol{\lambda}, \boldsymbol{\rho}, \boldsymbol{\xi})$ is a stationary point of (34).*

709 *Proof.* **(a) When $\phi_{fea} = 0$ holds:**

710 From the non-negativity of the function $p$ and $\|\cdot\|^2$, if $\phi_{fea}(\mathbf{z}) = 0$, it holds that $p(\mathbf{x}, \boldsymbol{\lambda}, \mathbf{r}, \boldsymbol{\xi}, s) = 0$
711 and $A_t\boldsymbol{\xi} + \sum_{i=1}^{M+1} \boldsymbol{\rho}_i = \mathbf{0}$

712 According to the constraints of (8), we know that

$$R_i(\mathbf{x}) \leq r_i, \ i = 1, ..., M + 1,$$
$$R_i^*(\frac{\boldsymbol{\rho}_i}{\lambda_i}) = 0, \ i = 1, ..., M.$$

Additionally, we restore $\lambda_{M+1} R^*_{M+1}(\frac{\boldsymbol{\rho}_{M+1}}{\lambda_{M+1}})$ with the inequality $\frac{\|\boldsymbol{\rho}_{M+1}\|_2^2}{2\lambda_{M+1}} \le s$. Consequently, we observe that

$$
\begin{aligned}
F(\mathbf{x}, \boldsymbol{\lambda}, \boldsymbol{\rho}, \boldsymbol{\xi}) &= l(\mathbf{x}) + \sum_{i=1}^{M+1} \lambda_i R_i(\mathbf{x}) + \varphi^*(\boldsymbol{\xi}) + \sum_{i=1}^{M+1} \lambda_i R_i^*(\frac{\boldsymbol{\rho}_i}{\lambda_i}) + \boldsymbol{\xi}^T \mathbf{b}_t \\
&= l(\mathbf{x}) + \sum_{i=1}^{M+1} \lambda_i R_i(\mathbf{x}) + \varphi^*(\boldsymbol{\xi}) + \lambda_{M+1} R_{M+1}^*(\frac{\boldsymbol{\rho}_{M+1}}{\lambda_{M+1}}) + \boldsymbol{\xi}^T \mathbf{b}_t \\
&\le l(\mathbf{x}) + \sum_{i=1}^{M+1} \lambda_i r_i + \varphi^*(\boldsymbol{\xi}) + \boldsymbol{\xi}^T \mathbf{b}_t + s \\
&= p(\mathbf{x}, \boldsymbol{\lambda}, \mathbf{r}, \boldsymbol{\xi}, s) = 0,
\end{aligned}
$$

which implies that $(\mathbf{x}, \boldsymbol{\lambda}, \boldsymbol{\rho}, \boldsymbol{\xi})$ is feasible to (5).

**(b) When $\phi_{res}^k(\mathbf{z}) = 0$ and $\phi_{fea}(\mathbf{z}) = 0$ both hold:**

In this part, we use Moreau-Rockafellar theorem [75, Theorem 23.8] to calculate the sum rule of subdifferentials. If $f_1$ and $f_2$ are convex and lower continuous at $x$ and $f_2$ is differentiable at $x \in \mathrm{int}(\mathrm{dom}(f_1)) \cap \mathrm{int}(\mathrm{dom}(f_2))$, then it holds that

$$
\partial(f_1 + f_2)(x) \subset \partial f_1(x) + \partial f_2(x).
$$

We analyze $\phi_{res}^k(\mathbf{z}) = 0$ for each component of $\mathbf{z}$.

- For $\mathbf{x}$ and $\mathbf{r}$, we have

$$
-(\nabla L(\mathbf{x}) + \beta_k \nabla l(\mathbf{x}), \beta_k \boldsymbol{\lambda}) \in \mathcal{N}_{\mathcal{K}_1 \cap \cdots \cap \mathcal{K}_{M+1}}(\mathbf{x}, \mathbf{r}), \tag{35}
$$

  where $\mathcal{K}_i = \{(\mathbf{x}, \mathbf{r}) \mid R_i(\mathbf{x}) \le r_i\}$. Let $\partial R_i$ denote the limiting subdifferential of the function $R_i$ [76]. According to the definition of the normal cone of inequality constraints [51, 52] and the definition of $\mathcal{K}_i$ in (9), we know that

$$
\begin{aligned}
\mathcal{N}_{\mathcal{K}_1 \cap \cdots \cap \mathcal{K}_{M+1}}(\mathbf{x}, \mathbf{r}) &= \mathrm{cone}\{(\partial R_i(\mathbf{x}), -1), i = 1, ..., M+1\} \\
&= \{\sum_{i=1}^{M+1} t_i(\partial R_i(\mathbf{x}), -1) \mid t_i \ge 0\},
\end{aligned}
$$

  where $\mathrm{cone}$ denotes the conic hull of a set. Combining with (35), we obtain

$$
0 \in \nabla L(\mathbf{x}) + \beta_k \nabla l(\mathbf{x}) + \beta_k \sum_{i=1}^{M+1} \lambda_i \partial R_i(\mathbf{x}). \tag{36}
$$

- For $\boldsymbol{\xi}$, we have

$$
\nabla \varphi^*(\boldsymbol{\xi}) + \mathbf{b}_t + A_t^T(A_t \boldsymbol{\xi} + \sum_{i=1}^{M+1} \boldsymbol{\rho}_i) = 0. \tag{37}
$$

- For $(\boldsymbol{\rho}_i, \lambda_i), i = 1, ..., M$, we have

$$
-(A_t \boldsymbol{\xi} + \sum_{i=1}^{M+1} \boldsymbol{\rho}_i, r_i) \in \mathcal{N}_{\mathcal{K}_i^d}(\boldsymbol{\rho}_i, \lambda_i), i = 1, ..., M,
$$

  where $\mathcal{K}_i^d = \{(\boldsymbol{\rho}_i, \lambda_i) \mid \|\boldsymbol{\rho}_i\|_{*(i)} \le \lambda_i\}$. From (30) and the definition of $p$, we know that $F(\mathbf{x}, \boldsymbol{\lambda}, \boldsymbol{\rho}, \boldsymbol{\xi}) \ge 0$ for all $(\mathbf{x}, \boldsymbol{\lambda}, \boldsymbol{\rho}, \boldsymbol{\xi})$. If $\phi_{fea}(\mathbf{z}) = 0$, the following chain of inequalities holds:

$$
0 \le F(\mathbf{x}, \boldsymbol{\lambda}, \boldsymbol{\rho}, \boldsymbol{\xi}) \le p(\mathbf{x}, \boldsymbol{\lambda}, \mathbf{r}, \boldsymbol{\xi}, s) \le 0,
$$

  which naturally reduces to equalities. Consequently, we have $F(\mathbf{x}, \boldsymbol{\lambda}, \boldsymbol{\rho}, \boldsymbol{\xi}) = p(\mathbf{x}, \boldsymbol{\lambda}, \mathbf{r}, \boldsymbol{\xi}, s)$, implying that $R_i(\mathbf{x}) = r_i, i = 1, ..., M$. Therefore, we obtain that

$$
-(A_t \boldsymbol{\xi} + \sum_{i=1}^{M+1} \boldsymbol{\rho}_i, R_i(\mathbf{x})) \in \mathcal{N}_{\mathcal{K}_i^d}(\boldsymbol{\rho}_i, \lambda_i), i = 1, ..., M,
$$

Meanwhile, we note that for $i = 1, ..., M$, $R_i^*$ is the indicator function of the set $\{\|\mathbf{y}\|_{*(i)} \leq 1\}$. Combining with the fact that the normal cone is equivalent to the subdifferential of indicator function, for the variables $\boldsymbol{\rho}_i$ and $\lambda_i$, the above formulation implies that

$$-(A_t\boldsymbol{\xi} + \sum_{i=1}^{M+1} \boldsymbol{\rho}_i) \in \partial_{\boldsymbol{\rho}_i} I_{\{\|\boldsymbol{\rho}\|_{*(i)} \leq \lambda_i\}} = \partial_{\boldsymbol{\rho}_i} I_{\{\|\boldsymbol{\rho}\|_{*(i)}/\lambda_i \leq 1\}} \overset{(*)}{=} \partial_{\boldsymbol{\rho}_i}\left[\lambda_i R_i^*(\frac{\boldsymbol{\rho}_i}{\lambda_i})\right]. \quad (38)$$

$$\begin{aligned}
-R_i(\mathbf{x}) &\in \partial_{\lambda_i} I_{\{\|\boldsymbol{\rho}\|_{*(i)} \leq \lambda_i\}} \overset{(a)}{=} \partial_{\lambda_i} I_{\{\|\boldsymbol{\rho}\|_{*(i)} \leq \lambda_i\}} + I_{\{\|\boldsymbol{\rho}\|_{*(i)} \leq \lambda_i\}} \\
&= \partial_{\lambda_i} I_{\{\|\boldsymbol{\rho}\|_{*(i)} \leq \lambda_i\}} + R_i^*(\frac{\boldsymbol{\rho}_i}{\lambda_i}) \overset{(*)}{=} \partial_{\lambda_i}\left[\lambda_i R_i^*(\frac{\boldsymbol{\rho}_i}{\lambda_i})\right],
\end{aligned} \quad (39)$$

where $(a)$ follows the fact $\|\boldsymbol{\rho}\|_{*(i)} \leq \lambda_i$ and $(*)$ holds from the direct calculation of the subdifferential.

- For $(\boldsymbol{\rho}_{M+1}, \lambda_{M+1}, s)$, we have

$$-(A_t\boldsymbol{\xi} + \sum_{i=1}^{M+1} \boldsymbol{\rho}_i, r_{M+1}, 1) \in \mathcal{N}_{\mathcal{K}_{M+1}^d}(\boldsymbol{\rho}_{M+1}, \lambda_{M+1}, s),$$

where $\mathcal{K}_{M+1}^d = \{(\boldsymbol{\rho}_{M+1}, \lambda_{M+1}, s) \mid \|\boldsymbol{\rho}_{M+1}\|_2^2 \leq 2\lambda_{M+1}s\}$. Similar to the deduction for $(\boldsymbol{\rho}_i, \lambda_i)$ in (38) and (39), we can obtain

$$\begin{aligned}
-(A_t\boldsymbol{\xi} + \sum_{i=1}^{M+1} \boldsymbol{\rho}_i) &\in \partial_{\boldsymbol{\rho}_{M+1}}\left[\lambda_{M+1} R_{M+1}^*(\frac{\boldsymbol{\rho}_{M+1}}{\lambda_{M+1}})\right], \\
-R_{M+1}(\mathbf{x}) &\in \partial_{\lambda_{M+1}}\left[\lambda_{M+1} R_{M+1}^*(\frac{\boldsymbol{\rho}_{M+1}}{\lambda_{M+1}})\right].
\end{aligned} \quad (40)$$

In summary, we find that the equations (36), (37), (38), (39) and (40) coincide with the stationary conditions of (34). Therefore, we conclude that $(\mathbf{x}, \boldsymbol{\lambda}, \boldsymbol{\xi}, \boldsymbol{\rho})$ is a stationary point of (34).

$\square$

From deduction (29) and (30), we conclude that $\phi_{fea}(\mathbf{z}) = 0$ implies

$$l(\mathbf{x}) + \sum_{i=1}^{M+1} \lambda_i R_i(\mathbf{x}) = \min_{\mathbf{x}}\{l(\mathbf{x}) + \sum_{i=1}^{M+1} \lambda_i R_i(\mathbf{x})\}.$$

Following the reasoning in Theorem 2.4, we conclude that as $\beta_k \to \infty$, any limit point of the sequence of optimal solutions to (34) with $\beta_k$ is an optimal solution of (5). According to (36), we further obtain that

$$\text{dist}(0, \nabla l(\mathbf{x}) + \sum_{i=1}^{M+1} \lambda_i \partial R_i(\mathbf{x})) \leq \frac{1}{\beta_k}\|\nabla L(\mathbf{x})\| \to 0,$$

as $\beta_k \to \infty$. Thess results demonstrate that $\phi_{res}^k$ and $\phi_{fea}$ can effectively character the optimality condition of the LL problem in (3). In summary, the selection of $\phi_{res}^k$ and $\phi_{fea}$ is reasonable.

We provide the proofs for the convergence results of Algorithm 1 and 2 in the subsequent sections.

### C.3 Proof of Theorem 3.5

We first recall the update for the variables of $\mathbf{z}$ in Algorithm 1 as follows. We calculate the update directions of $\mathbf{z}$ as $\mathbf{d_z}^k = \beta_k(\mathbf{d_x}^k, \mathbf{d_\lambda}^k, \mathbf{d_\rho}^k, \mathbf{d_r}^k, \mathbf{d_\xi}^k, \mathbf{d_s}^k)$, where

$$\begin{aligned}
\mathbf{d_x}^k &= \frac{1}{\beta_k}\nabla L(\mathbf{x}^k) + \nabla l(\mathbf{x}^k), \\
\mathbf{d_\xi}^k &= \nabla\varphi^*(\boldsymbol{\xi}^k) + \mathbf{b}_t + A_t^T(A_t\boldsymbol{\xi}^k + \boldsymbol{\rho}^k), \\
\mathbf{d_\lambda}^k &= \mathbf{r}^k, \ \mathbf{d_r}^k = \boldsymbol{\lambda}^k, \ d_s^k = 1, \\
\mathbf{d_\rho}^k &= A_t\boldsymbol{\xi}^{k+1} + \boldsymbol{\rho}^k.
\end{aligned} \quad (41)$$

With these directions, the gradient descent step is performed as

$$\bar{\mathbf{z}}^{k+1} = \mathbf{z}^k - e_k \mathbf{d}_{\mathbf{z}}^k.$$

For $\bar{\mathbf{z}}^{k+1} = (\bar{\mathbf{x}}^{k+1}, \bar{\boldsymbol{\lambda}}^{k+1}, \bar{\boldsymbol{\rho}}^{k+1}, \bar{\mathbf{r}}^{k+1}, \bar{\boldsymbol{\xi}}^{k+1}, \bar{s}^{k+1})$, we subsequently apply the projection

$$\mathbf{z}^{k+1} = \operatorname{proj}_{\mathcal{K}}(\bar{\mathbf{z}}^{k+1}). \tag{42}$$

Note that the variable $\boldsymbol{\xi}$ is not involved in the projection step and thus it is evolved directly as $\boldsymbol{\xi}^{k+1} = \bar{\boldsymbol{\xi}}^{k+1}$.

Next, we discuss the sufficient decrease property for Algorithm 1.

**Lemma C.2.** *Suppose Assumption 3.2 hold. For $k \in \mathbb{N}$, let $\{\mathbf{z}^k\}$ be generated from Algorithm 1. Define $V_k = \frac{1}{\beta_k} F_k(\mathbf{z}^k)$, then the following inequality holds:*

$$
\begin{aligned}
V_{k+1} - V_k \quad \leq \quad & \left( \frac{\alpha_L + \beta_k \|A_t\|_2^2 \alpha_p}{2\beta_k} - \frac{1}{e_k \beta_k} \right) \|\mathbf{x}^{k+1} - \mathbf{x}^k\|^2 + \left( \frac{1}{2} - \frac{1}{e_k \beta_k} \right) \|\boldsymbol{\rho}^{k+1} - \boldsymbol{\rho}^k\|^2 \\
& - \frac{1}{e_k \beta_k} \|s^{k+1} - s^k\|^2 + \left( \frac{1}{2} - \frac{1}{e_k \beta_k} \right) \left( \|\boldsymbol{\lambda}^{k+1} - \boldsymbol{\lambda}^k\|^2 + \|\mathbf{r}^{k+1} - \mathbf{r}^k\|^2 \right) \\
& + \left( \frac{\alpha_d + \|A_t\|_2^2}{2} - \frac{1}{e_k \beta_k} \right) \|\boldsymbol{\xi}^{k+1} - \boldsymbol{\xi}^k\|^2.
\end{aligned} \tag{43}
$$

*Furthermore, if the step sizes satisfy $0 < e_k \leq \min\{\frac{2}{\alpha_L + \beta_k \|A_t\|_2^2 \alpha_p}, \frac{2}{\beta_k}, \frac{2}{\beta_k(\alpha_d + \|A_t\|_2^2)}\}$, it holds that $V_{k+1} \leq V_k$.*

*Proof.* Given Assumption 3.2 that $\varphi$ is $\alpha_p$-smooth, we know that $l$ is $\|A_t\|_2^2 \alpha_p$-smooth. By applying the sufficient decrease lemma [10, Lemma 5.7], we obtain that

$$
\begin{aligned}
\frac{1}{\beta_k} L(\mathbf{x}^{k+1}) + l(\mathbf{x}^{k+1}) \leq & \frac{1}{\beta_k} L(\mathbf{x}^k) + l(\mathbf{x}^k) + \langle \frac{1}{\beta_k} \nabla L(\mathbf{x}^k) + \nabla l(\mathbf{x}^k), \mathbf{x}^{k+1} - \mathbf{x}^k \rangle \\
& + \frac{1}{2} (\frac{1}{\beta_k} \alpha_L + \|A_t\|_2^2 \alpha_p) \|\mathbf{x}^{k+1} - \mathbf{x}^k\|^2.
\end{aligned}
$$

Based on the convexity of the cones and the second projection theorem [10, Theorem 6.41], we have

$$\langle (\bar{\mathbf{x}}^{k+1}, \bar{\mathbf{r}}^{k+1}) - (\mathbf{x}^{k+1}, \mathbf{r}^{k+1}), (\mathbf{x}^k, \mathbf{r}^k) - (\mathbf{x}^{k+1}, \mathbf{r}^{k+1}) \rangle \leq 0$$

which implies

$$
\begin{aligned}
& \frac{1}{\beta_k} L(\mathbf{x}^{k+1}) + l(\mathbf{x}^{k+1}) + \langle \boldsymbol{\lambda}^{k+1}, \mathbf{r}^{k+1} - \mathbf{r}^k \rangle \\
\leq \quad & \frac{1}{\beta_k} L(\mathbf{x}^k) + l(\mathbf{x}^k) + \left( \frac{\alpha_L + \beta_k \|A_t\|_2^2 \alpha_p}{2\beta_k} - \frac{1}{e_k \beta_k} \right) \|\mathbf{x}^{k+1} - \mathbf{x}^k\|^2 + \left( \frac{1}{2} - \frac{1}{e_k \beta_k} \right) \|\mathbf{r}^{k+1} - \mathbf{r}^k\|^2.
\end{aligned}
$$

Given $\beta_k = \underline{\beta}(1+k)^p$, we have $\frac{1}{\beta_{k+1}} \leq \frac{1}{\beta_k}$. Combining the above inequalities and the non-negativity of $L$, we derive

$$
\begin{aligned}
& \beta_{k+1} L(\mathbf{x}^{k+1}) + l(\mathbf{x}^{k+1}) + \langle \boldsymbol{\lambda}^{k+1}, \mathbf{r}^{k+1} - \mathbf{r}^k \rangle \leq \frac{1}{\beta_k} L(\mathbf{x}^{k+1}) + l(\mathbf{x}^{k+1}) \langle \boldsymbol{\lambda}^{k+1}, \mathbf{r}^{k+1} - \mathbf{r}^k \rangle \\
\leq \quad & \frac{1}{\beta_k} L(\mathbf{x}^k) + l(\mathbf{x}^k) + \left( \frac{\alpha_L + \beta_k \|A_t\|_2^2 \alpha_p}{2\beta_k} - \frac{1}{e_k \beta_k} \right) \|\mathbf{x}^{k+1} - \mathbf{x}^k\|^2 + \left( \frac{1}{2} - \frac{1}{e_k \beta_k} \right) \|\mathbf{r}^{k+1} - \mathbf{r}^k\|^2.
\end{aligned} \tag{44}
$$

The same derivation process applies to $\boldsymbol{\rho}, \lambda_i, r_i$, leading to the following results:

$$
\begin{aligned}
& \|A_t \boldsymbol{\xi}^{k+1} + \boldsymbol{\rho}^{k+1}\|^2 + \langle \boldsymbol{\lambda}^{k+1} - \boldsymbol{\lambda}^k, \mathbf{r}^k \rangle \\
\leq \quad & \|A_t \boldsymbol{\xi}^{k+1} + \boldsymbol{\rho}^k\|^2 + \left( \frac{1}{2} - \frac{1}{e_k \beta_k} \right) \|\boldsymbol{\rho}^{k+1} - \boldsymbol{\rho}^k\|^2 + \left( \frac{1}{2} - \frac{1}{e_k \beta_k} \right) \|\boldsymbol{\lambda}^{k+1} - \boldsymbol{\lambda}^k\|^2.
\end{aligned} \tag{45}
$$

For the variable $s$, we deduce that $\bar{s}^{k+1} = s^k - e_k$ and $\langle \bar{s}^{k+1} - s^{k+1}, s^k - s^{k+1} \rangle \leq 0$, which implies that

$$s^{k+1} - s^k \leq -\frac{1}{e_k \beta_k} \|s^{k+1} - s^k\|^2. \tag{46}$$

Next, we define $H_k(\boldsymbol{\xi}) = \varphi^*(\boldsymbol{\xi}) + \boldsymbol{\xi}^T \mathbf{b}_t + \frac{1}{2} \|A_t \boldsymbol{\xi} + \boldsymbol{\rho}^k\|^2$, noting that $H_k$ is $(\alpha_d + \|A_t\|_2^2)$-smooth. Then the update of $\boldsymbol{\xi}$ in Algorithm 1 can be expressed as

$$\boldsymbol{\xi}^{k+1} = \bar{\boldsymbol{\xi}}^{k+1} = \boldsymbol{\xi}^k - e_k \beta_k \nabla H_k(\boldsymbol{\xi}^k).$$

Applying the sufficient decrease lemma [10, Lemma 5.7], we obtain

$$H_k(\boldsymbol{\xi}^{k+1}) \leq H_k(\boldsymbol{\xi}^k) + \langle \nabla H_k(\boldsymbol{\xi}^k), \boldsymbol{\xi}^{k+1} - \boldsymbol{\xi}^k \rangle + \frac{\alpha_d + \|A_t\|_2^2}{2} \|\boldsymbol{\xi}^{k+1} - \boldsymbol{\xi}^k\|^2,$$

which simplifies to

$$H_k(\boldsymbol{\xi}^{k+1}) \leq H_k(\boldsymbol{\xi}^k) + \left( \frac{\alpha_d + \|A_t\|_2^2}{2} - \frac{1}{e_k\beta_k} \right) \|\boldsymbol{\xi}^{k+1} - \boldsymbol{\xi}^k\|^2. \tag{47}$$

Summing up the estimates (44)–(47), we arrive at the inequality (43). Furthermore, when the step size satisfies $0 < e_k \leq \min\{\frac{2}{\alpha_L + \beta_k\|A_t\|_2^2 \alpha_p}, \frac{2}{\beta_k}, \frac{2}{\beta_k(\alpha_d + \|A_t\|_2^2)}\}$, the right-hand side of (43) becomes negative, ensuring that $V_{K+1} \leq V_k$. $\qquad\square$

Now we provide the proof for Theorem 3.5.

*Proof.* We compress (43) from $k = 0$ to $K - 1$ and obtain that

$$\begin{aligned}\sum_{k=1}^{K-1} &\left[ \left( \frac{1}{e_k\beta_k} - \frac{\alpha_L + \beta_k\|A_t\|_2^2 \alpha_p}{2\beta_k} \right) \|\mathbf{x}^{k+1} - \mathbf{x}^k\|^2 + \left( \frac{1}{e_k\beta_k} - \frac{\alpha_d + \|A_t\|_2^2}{2} \right) \|\boldsymbol{\xi}^{k+1} - \boldsymbol{\xi}^k\|^2 \right. \\ &\left. + \left( \frac{1}{e_k\beta_k} - \frac{1}{2} \right) \left( \|\boldsymbol{\rho}^{k+1} - \boldsymbol{\rho}^k\|^2 + \|\boldsymbol{\lambda}^{k+1} - \boldsymbol{\lambda}^k\|^2 + \|\mathbf{r}^{k+1} - \mathbf{r}^k\|^2 \right) \right] \leq V_0 - V_K.\end{aligned} \tag{48}$$

From the non-negativity of $L$ and $p$, we know that $V_K \geq 0$ and $V_0 - V_K \leq V_0$. Subsequently, according to the update rule of variables $(\mathbf{x}, \boldsymbol{\lambda}, \boldsymbol{\rho}, \mathbf{r}, \boldsymbol{\xi}, s)$ in Algorithm 1, we have that

$$\begin{aligned}0 &\in e_k(\nabla L(\mathbf{x}^k) + \beta_k \nabla l(\mathbf{x}^k)) + (\mathbf{x}^{k+1} - \mathbf{x}^k) + \mathcal{N}_{\mathcal{K}}(\mathbf{x}^{k+1}), \\ e_k\beta_k&(A_t^T(A_t\boldsymbol{\xi}^k + \boldsymbol{\rho}^k) + \mathbf{b}_t + \nabla\varphi^*(\boldsymbol{\xi}^k)) + (\boldsymbol{\xi}^{k+1} - \boldsymbol{\xi}^k) = 0.\end{aligned}$$

Therefore, it holds that

$$\begin{aligned}&\nabla L(\mathbf{x}^k) + \beta_k \nabla l(\mathbf{x}^k) + e_k(\mathbf{x}^{k+1} - \mathbf{x}^k) \in \mathcal{N}_{\mathcal{K}}(\mathbf{x}^{k+1}), \\ &\nabla_{\boldsymbol{\xi}} F_k(\mathbf{z}^k) + \frac{1}{e_k}(\boldsymbol{\xi}^{k+1} - \boldsymbol{\xi}^k) = 0.\end{aligned} \tag{49}$$

Furthermore, we have similar conclusions for $\boldsymbol{\lambda}, \mathbf{r}, \boldsymbol{\rho}, s$ as follows,

$$\begin{aligned}0 \in \;&(\nabla_{\boldsymbol{\lambda}}, \nabla_{\mathbf{r}}, \nabla_{\boldsymbol{\rho}}, \nabla_s) F_k(\mathbf{z}^k) + \frac{1}{e_k}(\boldsymbol{\lambda}^{k+1} - \boldsymbol{\lambda}^k, \mathbf{r}^{k+1} - \mathbf{r}^k, \boldsymbol{\rho}^{k+1} - \boldsymbol{\rho}^k, s^{k+1} - s^k) \\ &+ \mathcal{N}_{\mathcal{K}}(\boldsymbol{\lambda}^{k+1}, \mathbf{r}^{k+1}, \boldsymbol{\rho}^{k+1}, s^{k+1}).\end{aligned} \tag{50}$$

Now we define

$$M_{\mathbf{z}}^k := \nabla_{\mathbf{z}} F_k(\mathbf{z}^{k+1}) - \mathbf{d}_{\mathbf{z}}^k - \frac{1}{e_k}(\mathbf{z}^{k+1} - \mathbf{z}^k) \overset{(*)}{=} \nabla_{\mathbf{z}} F_k(\mathbf{z}^{k+1}) - \nabla_{\mathbf{z}} F_k(\mathbf{z}^k) - \frac{1}{e_k}(\mathbf{z}^{k+1} - \mathbf{z}^k),$$

where $(*)$ holds from $\mathbf{d}_{\mathbf{z}}^k = \nabla_{\mathbf{z}} F_k(\mathbf{z}^k)$. Using the directions specified in (41) and the relationship given in (49) and (50), we obtain

$$M_{\mathbf{z}}^k \in \nabla F_k(\mathbf{z}^{k+1}) + \mathcal{N}_{\mathcal{K}}(\mathbf{z}^{k+1}), \tag{51}$$

Based on the definition of the residual function $\phi_{res}^k$ in (16) and the relationship (51), we know that

$$\|M_{\mathbf{z}}^k\| \geq \text{dist}\left( 0, \nabla_{\mathbf{z}} F_k(\mathbf{z}^{k+1}) + \mathcal{N}_{\mathcal{K}}(\mathbf{z}^{k+1}) \right) = \phi_{res}^k(\mathbf{z}^{k+1}) \tag{52}$$

Subsequently, we estimate the value $\|M_{\mathbf{z}}^k\|$ with respect to $\mathbf{z}$. By using Assumptions 3.1 and 3.2, we find that $\|\nabla_{\mathbf{z}} F_k(\mathbf{z}^{k+1}) - \nabla_{\mathbf{z}} F_k(\mathbf{z}^k)\| \leq \beta_k L_{\mathbf{z}} \|\mathbf{z}^{k+1} - \mathbf{z}^k\|$ where $L_{\mathbf{z}} = \max\{\frac{\alpha_L + \beta_k\|A_t\|_2^2 \alpha_p}{\beta_k}, \alpha_d + \|A_t\|_2^2, 1\}$. Then we have

$$\|M_{\mathbf{z}}^k\| \leq \beta_k L_{\mathbf{z}} \|\mathbf{z}^{k+1} - \mathbf{z}^k\| + \frac{1}{e_k} \|\mathbf{z}^{k+1} - \mathbf{z}^k\|. \tag{53}$$

By combining (52) and the inequality (53), we deduce that

$$\phi_{res}^k(\mathbf{z}^{k+1}) \leq \beta_k L_{\mathbf{z}} \|\mathbf{z}^{k+1} - \mathbf{z}^k\| + \frac{1}{e_k} \|\mathbf{z}^{k+1} - \mathbf{z}^k\|. \tag{54}$$

When the step sizes are set as $0 < e_k \leq \frac{1}{M_k} \leq \min\{\frac{1}{\alpha_L + \beta_k \|A_t\|_2^2 \alpha_p}, \frac{1}{\beta_k}, \frac{1}{\beta_k(\alpha_d + \|A_t\|_2^2)}\}$, we know that $0 < e_k \leq \frac{1}{\beta_k}$, which implies that $\beta_k \leq \frac{1}{e_k}$. Then we conclude from (54) that there exists a constant $C_{res} > 0$ such that

$$\frac{1}{\beta_k^2} \phi_{res}^k(\mathbf{z}^{k+1})^2 \leq \frac{C_{res}}{e_k \beta_k} \|\mathbf{z}^{k+1} - \mathbf{z}^k\|^2. \tag{55}$$

From (48), we deduce that

$$
\begin{aligned}
&\sum_{k=0}^{\infty} \frac{1}{2 e_k \beta_k} \|\mathbf{z}^{k+1} - \mathbf{z}^k\|^2 \\
\leq\ & \sum_{k=0}^{\infty} \Big[ \Big( \frac{1}{e_k \beta_k} - \frac{(\|A_v\|_2^2 + \beta_k \|A_t\|_2^2)\alpha_p}{2\beta_k} \Big) \|\mathbf{x}^{k+1} - \mathbf{x}^k\|^2 + \Big( \frac{1}{e_k \beta_k} - \frac{\alpha_d + \|A_t\|_2^2}{2} \Big) \|\boldsymbol{\xi}^{k+1} - \boldsymbol{\xi}^k\|^2 \\
&+ \Big( \frac{1}{e_k \beta_k} - \frac{1}{2} \Big) \Big( \|\boldsymbol{\rho}^{k+1} - \boldsymbol{\rho}^k\|^2 + \|\boldsymbol{\lambda}^{k+1} - \boldsymbol{\lambda}^k\|^2 + \|\mathbf{r}^{k+1} - \mathbf{r}^k\|^2 \Big) \Big] \\
\leq\ & V_0.
\end{aligned}
\tag{56}
$$

By compressing (55) from $k = 0$ to $\infty$ and combining with the inequality (56), we obtain that

$$\sum_{k=0}^{\infty} \frac{1}{\beta_k^2} \phi_{res}^k(\mathbf{z}^{k+1})^2 \leq 2 C_{res} V_0.$$

Given $\beta_k = \underline{\beta}(1 + k)^p$ and $0 < p < \frac{1}{2}$, we conclude that

$$\min_{0 \leq k \leq K} \phi_{res}^k(\mathbf{z}^{k+1}) = \mathcal{O}(\frac{1}{K^{1/2-p}}).$$

From the definition of $\phi_{fea}$ in (17), we know that

$$0 \leq \beta_k \phi_{fea}(\mathbf{z}^k) \leq 2(F_k(\mathbf{z}^k) - L(\mathbf{z}^k)).$$

If the sequence $\{F_k(\mathbf{z}^k)\}$ is bounded, we know that there exists $M > 0$ such that $F_k(\mathbf{z}^k) \leq M$ for each $k$. Meanwhile, $L(\mathbf{x}^k) \geq 0$ holds from Assumption 3.1. Then we have

$$\beta_k \phi_{fea}(\mathbf{z}^k) \leq 2M,$$

which implies that $\phi_{fea}(\mathbf{z}^k) = \mathcal{O}(\frac{1}{K^p})$. $\qquad\square$

## C.4   Proof of Theorem 3.7

*Proof.* From the update rule for $\mathbf{u}$ in (23), we have

$$\mathcal{L}_\gamma^k(\mathbf{z}^{k+1}, \mathbf{u}^{k+1}, \boldsymbol{\mu}^k) \leq \mathcal{L}_\gamma^k(\mathbf{z}^{k+1}, \mathbf{u}^k, \boldsymbol{\mu}^k). \tag{57}$$

Additionally, the update rule for $\boldsymbol{\mu}$ in (24) implies

$$\mathcal{L}_\gamma^k(\mathbf{z}^{k+1}, \mathbf{u}^{k+1}, \boldsymbol{\mu}^{k+1}) - \mathcal{L}_\gamma^k(\mathbf{z}^{k+1}, \mathbf{u}^{k+1}, \boldsymbol{\mu}^k) = -\frac{1}{\gamma} \|\boldsymbol{\mu}^{k+1} - \boldsymbol{\mu}^k\|^2. \tag{58}$$

According to Assumptions 3.1 and 3.2, we know that $\mathcal{L}_\gamma^k(\mathbf{z}, \mathbf{u}, \boldsymbol{\mu})$ is $M_k$-smooth with respect to $\mathbf{z}$, where $M_k = \max\{\frac{\alpha_L + \beta_k \|A_t\|_2^2 \alpha_p}{\beta_k}, \alpha_d + \|A_t\|_2^2, 1\}$. According to [10, Lemma 5.7], we have

$$\mathcal{L}_\gamma^k(\mathbf{z}^{k+1}, \mathbf{u}^k, \boldsymbol{\mu}^k) \leq \mathcal{L}_\gamma^k(\mathbf{z}^k, \mathbf{u}^k, \boldsymbol{\mu}^k) + \langle \nabla_{\mathbf{z}} \mathcal{L}_\gamma^k(\mathbf{z}^k, \mathbf{u}^k, \boldsymbol{\mu}^k), \mathbf{z}^{k+1} - \mathbf{z}^k \rangle + \frac{M_k}{2} \|\mathbf{z}^{k+1} - \mathbf{z}^k\|^2.$$

Given the update rule $\mathbf{z}^{k+1} = \mathbf{z}^k - e_k \nabla_{\mathbf{z}} \mathcal{L}_\gamma^k(\mathbf{z}^k, \mathbf{u}^k, \boldsymbol{\mu}^k)$, the inequality becomes

$$\mathcal{L}_\gamma^k(\mathbf{z}^{k+1}, \mathbf{u}^k, \boldsymbol{\mu}^k) \leq \mathcal{L}_\gamma^k(\mathbf{z}^k, \mathbf{u}^k, \boldsymbol{\mu}^k) + \Big( \frac{M_k}{2} - \frac{1}{e_k} \Big) \|\mathbf{z}^{k+1} - \mathbf{z}^k\|^2. \tag{59}$$

Combining (57), (58) and (59) and dividing both sides by $\beta_k$, we conclude

$$\mathcal{L}_\gamma^k(\mathbf{z}^{k+1}, \mathbf{u}^{k+1}, \boldsymbol{\mu}^{k+1}) - \mathcal{L}_\gamma^k(\mathbf{z}^k, \mathbf{u}^k, \boldsymbol{\mu}^k) \leq \Big( \frac{M_k}{2} - \frac{1}{e_k} \Big) \|\mathbf{z}^{k+1} - \mathbf{z}^k\|^2 - \frac{1}{\gamma} \|\boldsymbol{\mu}^{k+1} - \boldsymbol{\mu}^k\|^2.$$

812 According to $\beta_k = \underline{\beta}(1+k)^p$, we obtain that $\frac{1}{\beta_{k+1}} \leq \frac{1}{\beta_k}$. With the non-negativity of $L$, it holds that

813 $\mathcal{L}_\gamma^{k+1}(\mathbf{z}^{k+1}, \mathbf{u}^{k+1}, \boldsymbol{\mu}^{k+1}) \leq \mathcal{L}_\gamma^k(\mathbf{z}^{k+1}, \mathbf{u}^{k+1}, \boldsymbol{\mu}^{k+1})$, which implies that

$$\mathcal{L}_\gamma^{k+1}(\mathbf{z}^{k+1}, \mathbf{u}^{k+1}, \boldsymbol{\mu}^{k+1}) - \mathcal{L}_\gamma^k(\mathbf{z}^k, \mathbf{u}^k, \boldsymbol{\mu}^k) \leq \left(\frac{M_k}{2} - \frac{1}{e_k}\right) \|\mathbf{z}^{k+1} - \mathbf{z}^k\|^2 - \frac{1}{\gamma}\|\boldsymbol{\mu}^{k+1} - \boldsymbol{\mu}^k\|^2. \quad (60)$$

814 Now we define $U_k = \mathcal{L}_\gamma^k(\mathbf{z}^k, \mathbf{u}^k, \boldsymbol{\mu}^k)$. Given that $0 < e_k \leq \min\{\frac{\beta_k}{\alpha_L + \beta_k\|A_t\|_2^2\alpha_p}, \frac{1}{\alpha_d + \|A_t\|_2^2}, 1\} \leq$

815 $\frac{1}{M_k}$, we can deduce from (60) that

$$U_{k+1} - U_k \leq -\frac{1}{2e_k}\|\mathbf{z}^{k+1} - \mathbf{z}^k\|^2 - \frac{1}{\gamma}\|\boldsymbol{\mu}^{k+1} - \boldsymbol{\mu}^k\|^2. \quad (61)$$

816 From the expression for $\mathcal{L}_\gamma^k$, we can deduce the following,

$$\begin{aligned}
\mathcal{L}_\gamma^k(\mathbf{z}, \mathbf{u}, \boldsymbol{\mu}) &= F_k(\mathbf{z}) + \sum_{i=1}^{M+1} g_i(\mathbf{u}_i) + \sum_{i=1}^{M+1} \langle \boldsymbol{\mu}_i, \mathbf{u}_i - \mathbf{z}\rangle + \frac{\gamma}{2}\sum_{i=1}^{M+1}\|\mathbf{u}_i - \mathbf{z}\|^2 \\
&= F_k(\mathbf{z}) + \sum_{i=1}^{M+1} g_i(\mathbf{u}_i) + \frac{\gamma}{2}\sum_{i=1}^{M+1}\|\mathbf{u}_i - \mathbf{z} + \frac{\boldsymbol{\mu}_i}{\gamma}\|^2 - \sum_{i=1}^{M+1}\frac{\|\boldsymbol{\mu}_i\|^2}{2\gamma}.
\end{aligned}$$

817 According to Assumption 3.6, we know that there exists some $M_{\boldsymbol{\mu}}$ such that $\|\boldsymbol{\mu}^k\|^2 \leq M_{\boldsymbol{\mu}}$ for all
818 $k \in \mathbb{N}$. Additionally, the functions $L$ and $p$ are non-negative. This implies that

$$U_k \geq -\sum_{i=1}^{M+1}\frac{\|\boldsymbol{\mu}_i^k\|^2}{2\gamma\beta_k} \geq -\frac{(M+1)M_{\boldsymbol{\mu}}}{2\gamma\underline{\beta}} \triangleq \mathcal{L}_b, \forall k \in \mathbb{N}, \quad (62)$$

819 indicating that $U_k$ is lower bounded. By telescoping the inequality (61) for $k = 0$ to $\infty$, we get

$$\sum_{k=0}^{\infty}\frac{1}{2e_k}\|\mathbf{z}^{k+1} - \mathbf{z}^k\|^2 + \frac{1}{\gamma}\sum_{k=0}^{\infty}\|\boldsymbol{\mu}^{k+1} - \boldsymbol{\mu}^k\|^2 \leq U_0 - \mathcal{L}_b. \quad (63)$$

820 The sufficient decrease property (60) ensures that the $U_0 - \mathcal{L}_b \geq U_0 - U_k \geq 0$ for any $k \in \mathbb{N}$.
821 Combining with the fact that $0 < \frac{1}{e_k} \leq \frac{1}{\underline{e}}$ are bounded, and both $e_k$ and $\gamma$ is positive, we obtain from
822 (63) that

$$\lim_{k\to\infty}\frac{1}{e_k}\|\mathbf{z}^{k+1} - \mathbf{z}^k\|^2 = 0, \quad \lim_{k\to\infty}\|\boldsymbol{\mu}^{k+1} - \boldsymbol{\mu}^k\| = 0. \quad (64)$$

823 Additionally, the step size $e_k$ satisfies $0 < \underline{e} < e_k \leq \min\{\frac{\beta_k}{\alpha_L + \beta_k\|A_t\|_2^2\alpha_p}, \frac{1}{\alpha_d + \|A_t\|_2^2}, 1\} \leq \frac{1}{M_k}$. This
824 implies that $\max\{\|A_t\|_2^2\alpha_p, \alpha_d + \|A_t\|_2^2, 1\} \leq \lim_{k\to\infty}\frac{1}{e_k} \leq \frac{1}{\underline{e}}$. Therefore, (64) ensures that

$$\lim_{k\to\infty}\|\mathbf{z}^{k+1} - \mathbf{z}^k\| = 0 \quad (65)$$

825 From the update of $\boldsymbol{\mu}_i$, we further derive that

$$\lim_{k\to\infty}\|\mathbf{u}_i^k - \mathbf{z}^k\| = 0. \quad (66)$$

826 Meanwhile, from the form (22) for updating $\mathbf{u}_i$, we derive

$$\begin{aligned}
\mathbf{0} &\in \partial g_i(\mathbf{u}_i^{k+1}) + \gamma(\mathbf{u}_i^{k+1} - \mathbf{z}^{k+1} + \frac{\boldsymbol{\mu}_i^k}{\gamma}) \\
&\overset{(a)}{=} \mathcal{N}_{\mathcal{K}_i \times \mathcal{K}_*^d}(\mathbf{u}_i^{k+1}) + \gamma(\mathbf{u}_i^{k+1} - \mathbf{z}^{k+1}) + \boldsymbol{\mu}_i^k \\
&\overset{(b)}{=} \mathcal{N}_{\mathcal{K}_i \times \mathcal{K}_*^d}(\mathbf{u}_i^{k+1}) + \boldsymbol{\mu}_i^{k+1}, \quad i = 1, ..., M+1,
\end{aligned} \quad (67)$$

827 where $(a)$ utilizes the fact that the normal cone is equivalent to the subdifferential of indicator
828 functions and $(b)$ follows from the update of $\boldsymbol{\mu}_i^{k+1}$. In (67), we use Moreau-Rockafellar theorem [75,
829 Theorem 23.8] to calculate the sum rule of subdifferentials. (67) implies that

$$-\boldsymbol{\mu}_i^{k+1} \in \mathcal{N}_{\mathcal{K}_i \times \mathcal{K}_*^d}(\mathbf{u}_i^{k+1}).$$

830 Combining the outer semi-continuity of the normal cone and (66), we can obtain that

$$\lim_{k\to\infty}\text{dist}(-\boldsymbol{\mu}_i^k, \mathcal{N}_{\mathcal{K}_i \times \mathcal{K}_*^d}(\mathbf{z}^k)) = 0. \quad (68)$$

831 Furthermore, according to the definition $\mathcal{K} = (\mathcal{K}_1 \cap \cdots \cap \mathcal{K}_{M+1}) \times \mathcal{K}_*^d$, we know that $\mathcal{K} =$
832 $(\mathcal{K}_1 \times \mathcal{K}_*^d) \cap \cdots \cap (\mathcal{K}_{M+1} \times \mathcal{K}_*^d)$. It implies that

$$\mathcal{N}_\mathcal{K} = \mathcal{N}_{\mathcal{K}_1 \times \mathcal{K}_*^d} + \cdots + \mathcal{N}_{\mathcal{K}_{M+1} \times \mathcal{K}_*^d}.$$

833 From (68), we know

$$\lim_{k \to \infty} \operatorname{dist}(-\sum_{i=1}^{M+1} \boldsymbol{\mu}_i^k, \mathcal{N}_\mathcal{K}(\mathbf{z}^k)) = 0. \tag{69}$$

834 From the update of $\mathbf{z}$, we have

$$\mathbf{z}^{k+1} = \mathbf{z}^k - e_k \nabla_\mathbf{z} \mathcal{L}_\gamma^k(\mathbf{z}^k, \mathbf{u}^k, \boldsymbol{\mu}^k).$$

835 Combining with the definition of $F_k$ in (8), the above equality can be further expressed as

$$\begin{aligned}
\mathbf{0} &= -\tfrac{1}{e_k}(\mathbf{z}^{k+1} - \mathbf{z}^k) + \tfrac{1}{\beta_k}\nabla_\mathbf{z} F_k(\mathbf{z}^k) - \sum_{i=1}^{M+1} \boldsymbol{\mu}_i^k - \gamma \sum_{i=1}^{M+1} (\mathbf{u}_i^k - \mathbf{z}^k) \\
&= -\tfrac{1}{e_k}(\mathbf{z}^{k+1} - \mathbf{z}^k) + \tfrac{1}{\beta_k}\nabla_\mathbf{z} F_k(\mathbf{z}^k) - \sum_{i=1}^{M+1} \boldsymbol{\mu}_i^{k+1} + \sum_{i=1}^{M+1} (\boldsymbol{\mu}_i^{k+1} - \boldsymbol{\mu}_i^k) - \gamma \sum_{i=1}^{M+1} (\mathbf{u}_i^k - \mathbf{z}^k).
\end{aligned} \tag{70}$$

836 Now we define

$$M_\mathbf{z}^k = \nabla_\mathbf{z} F_k(\mathbf{z}^{k+1}) - \beta_k \sum_{i=1}^{M+1} \boldsymbol{\mu}_i^{k+1}.$$

837 From (69), we know that

$$\lim_{k \to \infty} \operatorname{dist}(M_\mathbf{z}^k, \nabla_\mathbf{z} F_k(\mathbf{z}^{k+1}) + \mathcal{N}_\mathcal{K}(\mathbf{z}^{k+1})) = 0.$$

838 Therefore, we evaluate $\|M_\mathbf{z}^k\|$ as follows. According to (70), we know that

$$M_\mathbf{z}^k = \frac{\beta_k}{e_k}(\mathbf{z}^{k+1} - \mathbf{z}^k) + (\nabla_\mathbf{z} F_k(\mathbf{z}^{k+1}) - \nabla_\mathbf{z} F_k(\mathbf{z}^k)) + \sum_{i=1}^{M+1} \beta_k(\boldsymbol{\mu}_i^k - \boldsymbol{\mu}_i^{k+1}) + \gamma \beta_k \sum_{i=1}^{M+1} (\mathbf{u}_i^k - \mathbf{z}^k).$$

839 With the notation $M_k$, we know that $F_k(\mathbf{z})$ is $(\beta_k M_k)$-smooth wit respect to $\mathbf{z}$. Then we have

$$\begin{aligned}
\|M_\mathbf{z}^k\| &\leq \quad \tfrac{\beta_k}{e_k}\|\mathbf{z}^{k+1} - \mathbf{z}^k\| + \beta_k M_k \|\mathbf{z}^{k+1} - \mathbf{z}^k\| + \beta_k \|\boldsymbol{\mu}^{k+1} - \boldsymbol{\mu}^k\| + \gamma \beta_k \sum_{i=1}^{M+1} \|\mathbf{u}_i^k - \mathbf{z}^k\| \\
&\overset{(a)}{\leq} \quad \tfrac{2\beta_k}{e_k}\|\mathbf{z}^{k+1} - \mathbf{z}^k\| + \beta_k \|\boldsymbol{\mu}^{k+1} - \boldsymbol{\mu}^k\| + \gamma \beta_k \sum_{i=1}^{M+1} \|\mathbf{u}_i^k - \mathbf{z}^k\|,
\end{aligned}$$

840 where $(a)$ use the fact that $e_k \leq \frac{1}{M_k}$. Combining the definition of $\phi_{res}^k$ in (25), we obtain

$$\begin{aligned}
\phi_{res}^k(\mathbf{z}^{k+1}) &\leq \|M_\mathbf{z}^k\| + \operatorname{dist}(M_\mathbf{z}^k, \nabla_\mathbf{z} F_k(\mathbf{z}^{k+1}) + \mathcal{N}_\mathcal{K}(\mathbf{z}^{k+1})) \\
&\leq \tfrac{2\beta_k}{e_k}\|\mathbf{z}^{k+1} - \mathbf{z}^k\| + \beta_k \|\boldsymbol{\mu}^{k+1} - \boldsymbol{\mu}^k\| + \gamma \beta_k \sum_{i=1}^{M+1} \|\mathbf{u}_i^k - \mathbf{z}^k\| \\
&\quad + \operatorname{dist}(M_\mathbf{z}^k, \nabla_\mathbf{z} F_k(\mathbf{z}^{k+1}) + \mathcal{N}_\mathcal{K}(\mathbf{z}^{k+1})).
\end{aligned}$$

841 (63) and (65) imply that $\|\mathbf{z}^{k+1} - \mathbf{z}^k\| \leq \mathcal{O}(1/\sqrt{k})$, $\|\boldsymbol{\mu}^{k+1} - \boldsymbol{\mu}^k\| \leq \mathcal{O}(1/\sqrt{k})$ and $\|\mathbf{u}_i^k - \mathbf{z}^k\| \leq$
842 $\mathcal{O}(1/\sqrt{k})$. Combining with the fact that $0 < \frac{1}{e_k} \leq \frac{1}{\underline{e}}$ and $0 < p < 1/2$, we take the limit as $k \to \infty$
843 in the above inequality and obtain that

$$\lim_{k \to \infty} \phi_{res}^k(\mathbf{z}^k) = 0.$$

844 If the sequence $\{F_k(\mathbf{z}^k)\}$ is bounded, we know that there exists a constant $M$ such that $F_k(\mathbf{z}^k) \leq M$
845 for all $k$. From the formulation $\phi_{fea}$ in (26), we observe that

$$0 \leq \beta_k \phi_{fea}(\mathbf{z}^k) \leq 2(F_k(\mathbf{z}^k) - L(\mathbf{z}^k)) \overset{(a)}{\leq} 2M,$$

846 where $(a)$ holds from the non-negativity of $L$ from Assumption 3.1. With the non-negativity of $\phi_{fea}$,
847 we take the limit $k \to \infty$ in the above inequality and obtain that

$$\lim_{k \to \infty} \phi_{fea}(\mathbf{z}^k) = 0.$$

848 $\qquad\qquad\qquad\qquad\qquad\qquad\qquad\qquad\qquad\qquad\qquad\qquad\qquad\qquad\qquad\qquad\qquad\qquad\qquad\qquad\quad \square$

 # D    Experiments

All experiments are implemented using Python 3.9 on a computer equipped with an Apple M2 chip (8-core architecture: 4 performance cores and 4 efficiency cores), running the macOS operating system with 8 GB memory. The competing methods are implemented using the code provided by [36, 22, 95].

## D.1    Introduction for Competitors

We now introduce the competing methods evaluated in our experiments:

- **Grid Search**: We perform a $10 \times 10$ uniformly-spaced grid search over the hyperparameter space.
- **Random Search**: We uniformly sample 100 configurations for each hyperparameter direction.
- **Implicit Differentiation**: This category includes IGJO [31] and IFDM [14, 15], both of which rely on implicit differentiation techniques.
- **TPE**: We adopt the Tree-structured Parzen Estimator approach [13], a widely used Bayesian optimization method.
- **VF-iDCA**: [36] formulates the lower-level problem as a value function and approximately solves the bilevel problem via DC programming.
- **LDMMA**: Based on lower-level duality, [22] reformulates the original problem (3) into a more tractable form.
- **BiC-GAFFA**: [94] solves the bilevel optimization problem using a gap function-based framework.

We apply IFDM only to the elastic net and logistic regression problems, as its available implementation supports only these two among our tested tasks. LDMMA is used exclusively for Lasso-type regression and the smoothed support vector machine, as its reformulation is not compatible with logistic regression. Furthermore, [36] does not provide experimental results for logistic regression, and therefore we do not include it in the comparison for that task.

## D.2    Experimental on Synthetic Data

For experiments on synthetic data, we consider hyperparameter optimization for elastic net, group Lasso, and sparse group Lasso. These models are equipped with a least squares loss and different regularization terms. We outline the specific mathematical form of (3) for each problem below.

Elastic net [100] is a linear combination of the Lasso and ridge penalties. Its formulation in (3) is given by:

$$\begin{aligned} \min_{\mathbf{x}} \quad & \tfrac{1}{2}\|A_{val}\mathbf{x} - \mathbf{b}_{val}\|^2 \\ \text{s.t.} \quad & \mathbf{x} \in \arg\min_{\hat{\mathbf{x}}} \tfrac{1}{2}\|A_{tr}\hat{\mathbf{x}} - \mathbf{b}_{tr}\|^2 + \lambda_1\|\hat{\mathbf{x}}\|_1 + \tfrac{\lambda_2}{2}\|\hat{\mathbf{x}}\|_2^2, \end{aligned} \tag{71}$$

Group Lasso [99] is an extension of the Lasso with penalty to predefined groups of coefficients. This problem is captured in (3) as:

$$\begin{aligned} \min_{\mathbf{x}} \quad & \tfrac{1}{2}\|A_{val}\mathbf{x} - \mathbf{b}_{val}\|^2 \\ \text{s.t.} \quad & \mathbf{x} \in \arg\min_{\hat{\mathbf{x}}} \tfrac{1}{2}\|A_{tr}\hat{\mathbf{x}} - \mathbf{b}_{tr}\|^2 + \sum_{i=1}^{M} \lambda_i\|\hat{\mathbf{x}}^{(i)}\|_2, \end{aligned} \tag{72}$$

where $\mathbf{x}^{(i)}$ is a sub-vector of $\mathbf{x}$ and $\mathbf{x} = (\mathbf{x}^{(1)}, ..., \mathbf{x}^{(M)})$.

Sparse group Lasso [83] combines the group Lasso and Lasso penalties, which are designed to encourage sparsity and grouping of predictors [31]. Its formulation in (3) is represented as:

$$\begin{aligned} \min_{\mathbf{x}} \quad & \tfrac{1}{2}\|A_{val}\mathbf{x} - \mathbf{b}_{val}\|^2 \\ \text{s.t.} \quad & \mathbf{x} \in \arg\min_{\hat{\mathbf{x}}} \tfrac{1}{2}\|A_{tr}\hat{\mathbf{x}} - \mathbf{b}_{tr}\|^2 + \lambda_{M+1}\|\hat{\mathbf{x}}\|_1 + \sum_{i=1}^{M} \lambda_i\|\hat{\mathbf{x}}^{(i)}\|_2, \end{aligned} \tag{73}$$

where $\mathbf{x}^{(i)}$ is a sub-vector of $\mathbf{x}$ and $\mathbf{x} = (\mathbf{x}^{(1)}, ..., \mathbf{x}^{(M)})$.

Based on the different cases discussed in Section 3.1 and Section 3.2, we naturally employ Algorithm 1 to solve (72), and Algorithm 2 to address (71) and (73). To evaluate the performance of each method, we calculate validation and test error with obtained LL minimizers in each experiment. We provide detailed experimental settings and report the results for elastic net and group lasso below.

### D.2.1 Elastic Net

The synthetic data is generated following the methodology described by [31], as outlined below. Feature vectors $\mathbf{a}_i \in \mathbb{R}^p$ are sampled from a multivariate normal distribution with a mean of 0 and covariance structure $\mathrm{cor}(a_{ij}, a_{ik}) = 0.5^{|j-k|}$. The response vector $\mathbf{b}$ is computed as $b_i = \boldsymbol{\beta}^\top \mathbf{a}_i + \sigma \epsilon_i$, where $\beta_i \in \mathbb{R}^p$ is generated such that each element takes a value of either 0 or 1, with exactly 15 nonzero elements. The noise $\boldsymbol{\epsilon}$ is sampled from a standard normal distribution, and the value of $\sigma$ is determined to ensure that the signal-to-noise ratio satisfies $\mathrm{SNR} \triangleq \|A\boldsymbol{\beta}\|/\|\mathbf{b} - A\boldsymbol{\beta}\| = 2$. Since [95] does not provide experiments or code for the elastic net problem, we compare only with search-based methods, IGJO, IFDM, VF-iDCA and LDMMA in this experiment. We implement the algorithms we compared with the same settings according to the description in [36, 22]. For LDPM with Algorithm 2, we set $\beta_k = (1+k)^{0.3}$, $e_k = 0.1$ and $\gamma = 10$. For elastic net problem, the stopping criterion is set as $\|\mathbf{z}^{k+1} - \mathbf{z}^k\|/\|\mathbf{z}^{k+1}\| \le 0.1$.

We conduct repeated experiments with 10 randomly generated synthetic data, and calculate the mean and variance. The numerical results on elastic net are reported in Table 3. Overall, LDPM achieves the lowest test error while maintaining a significantly reduced time cost, especially for large-scale datasets. In contrast, the search methods incur a high computational cost and exhibit poor performance on the test dataset. The gradient-based method IGJO demonstrates slightly better accuracy and efficiency but converges very slowly.

As discussed in [36, 22], both VF-iDCA and LDMMA achieve consistently low validation errors across various experiments, indicating strong learning performance on training and validation sets. However, they tend to suffer from overfitting, as reflected in increasing test errors over iterations and poor generalization to unseen data. This phenomenon occurs across experiments with several machine learning models.

Table 3: Elastic net problems on synthetic data, where $|I_{tr}|$, $|I_{val}|$, $|I_{te}|$ and $p$ represent the number of training observations, validation observations, predictors and features, respectively.

| Settings | Methods | Time(s) | Val. Err. | Test Err. | Settings | Time(s) | Val. Err. | Test Err. |
|---|---|---|---|---|---|---|---|---|
| $|I_{tr}|=100$ $|I_{val}|=20$ $|I_{te}|=250$ $p=250$ | Grid | $5.76 \pm 0.33$ | $7.05 \pm 2.02$ | $6.98 \pm 1.14$ | $|I_{tr}|=100$ $|I_{val}|=100$ $|I_{te}|=250$ $p=450$ | $11.72 \pm 1.32$ | $6.05 \pm 1.47$ | $6.49 \pm 0.82$ |
| | Random | $5.74 \pm 0.26$ | $7.01 \pm 2.01$ | $7.01 \pm 1.11$ | | $12.85 \pm 2.11$ | $6.04 \pm 1.45$ | $6.49 \pm 0.83$ |
| | IGJO | $1.54 \pm 0.84$ | $4.99 \pm 1.69$ | $5.42 \pm 1.21$ | | $3.37 \pm 1.85$ | $5.22 \pm 1.50$ | $5.72 \pm 0.91$ |
| | IFDM | $1.20 \pm 0.50$ | $4.19 \pm 0.91$ | $4.81 \pm 1.39$ | | $1.44 \pm 2.85$ | $4.89 \pm 0.12$ | $4.98 \pm 0.17$ |
| | VF-iDCA | $3.16 \pm 0.63$ | $2.72 \pm 1.57$ | $5.18 \pm 1.40$ | | $6.08 \pm 2.24$ | $3.13 \pm 0.78$ | $5.39 \pm 0.92$ |
| | LDMMA | $1.64 \pm 0.07$ | $0.00 \pm 0.00$ | $6.97 \pm 0.79$ | | $3.95 \pm 0.22$ | $0.00 \pm 0.00$ | $6.56 \pm 0.70$ |
| | LDPM | $\mathbf{0.60 \pm 0.02}$ | $2.56 \pm 0.80$ | $\mathbf{4.92 \pm 0.51}$ | | $\mathbf{1.02 \pm 0.03}$ | $3.42 \pm 0.39$ | $\mathbf{4.23 \pm 0.37}$ |
| $|I_{tr}|=100$ $|I_{val}|=100$ $|I_{te}|=250$ $p=250$ | Grid | $6.09 \pm 0.60$ | $6.39 \pm 1.09$ | $6.27 \pm 1.02$ | $|I_{tr}|=100$ $|I_{val}|=100$ $|I_{te}|=100$ $p=2500$ | $32.99 \pm 3.81$ | $7.81 \pm 1.53$ | $8.82 \pm 0.92$ |
| | Random | $6.44 \pm 1.28$ | $4.39 \pm 1.10$ | $6.27 \pm 1.05$ | | $33.82 \pm 2.66$ | $6.44 \pm 1.53$ | $8.67 \pm 0.94$ |
| | IGJO | $3.86 \pm 2.09$ | $4.41 \pm 0.98$ | $4.31 \pm 0.95$ | | $31.30 \pm 6.41$ | $7.78 \pm 1.12$ | $8.61 \pm 0.82$ |
| | IFDM | $1.17 \pm 0.38$ | $4.54 \pm 1.06$ | $4.38 \pm 1.06$ | | $3.94 \pm 2.28$ | $7.57 \pm 0.79$ | $8.10 \pm 1.45$ |
| | VF-iDCA | $4.74 \pm 1.77$ | $2.35 \pm 1.56$ | $4.47 \pm 1.11$ | | $23.21 \pm 4.96$ | $0.00 \pm 0.00$ | $4.61 \pm 0.77$ |
| | LDMMA | $0.98 \pm 0.09$ | $0.00 \pm 0.00$ | $5.61 \pm 0.77$ | | $16.26 \pm 1.44$ | $0.00 \pm 0.00$ | $5.67 \pm 1.21$ |
| | LDPM | $\mathbf{0.73 \pm 0.08}$ | $3.41 \pm 0.48$ | $\mathbf{3.51 \pm 0.40}$ | | $\mathbf{4.83 \pm 0.08}$ | $1.65 \pm 0.14$ | $\mathbf{4.37 \pm 0.65}$ |

In our experiments, we report the numerical results of VF-iDCA and LDMMA based on the final iteration output when the algorithm terminates. In contrast, [36, 22] reports the best results observed across all iterations. As a result, the test errors reported for VF-iDCA and LDMMA in Table 3 appear slightly worse in our study. Additionally, our test error is slightly worse than that reported in [22] only under the first data setting in Table 3. [22] implements LDMMA with employing off-the-shelf solver MOSEK in MATLAB to solve the subproblems. Therefore, LDMMA yields highly favorable results for small-scale problems, while its efficiency deteriorates significantly as the data size increases, making it less effective for large-scale problem instances.

We observe that the running time performance of IFDM is highly competitive and significantly fast in large scale. This is because the IFDM algorithm leverages the sparsity of the Jacobian of the hyper-objective in bilevel optimization, which is also stated in [15].

 **D.2.2    Sparse Group Lasso**

We generate the synthetic data with the method in [31], including 100 training, validation and test samples, respectively. The feature vector $\mathbf{a}_i \in \mathbb{R}^p$ is drawn from a standard normal distribution. The response vector $\mathbf{b}$ is computed as $b_i = \boldsymbol{\beta}^\top \mathbf{a}_i + \sigma \epsilon_i$, where $\boldsymbol{\beta} = \left[ \boldsymbol{\beta^{(1)}}, \boldsymbol{\beta^{(2)}}, \boldsymbol{\beta^{(3)}} \right]$, $\boldsymbol{\beta^{(i)}} = (1, 2, 3, 4, 5, 0, \ldots, 0)$, for $i = 1, 2, 3$. The noise vector $\boldsymbol{\epsilon}$ follows a standard normal distribution, and $\sigma$ is set such that the signal-to-noise ratio (SNR) is 2. For different dimensions in Table 2, we set the group size to 30 for $p = 600$ and $p = 1200$, and to 300 for $p = 2400$ and $p = 4800$. Notably, compared to [36, 22], our feature vector dimensions are larger, while the number of samples is evidently smaller.

We compare our method with search methods, IGJO, VF-iDCA, LDMMA and BiC-GAFFA in this experiment. For the compared method BiC-GAFFA, we follow the recommended procedure outlined in [95]. For the other comparison methods, we adopt the exact settings from [36, 22]. For LDPM with Algorithm 2, we set $\beta_k = (1+k)^{0.3}$, $\gamma = 10$ and the step size $e_k = 0.001$. For sparse group Lasso problem, the stopping criterion is set as $\|\mathbf{z}^{k+1} - \mathbf{z}^k\|/\|\mathbf{z}^{k+1}\| \le 0.2$.

From Table 2, we observe that LDPM achieves lowest test error and outperforms other algorithms in terms of time cost. As the scale of data increases, LDPM consistently finds the best hyperparameters and model solutions. In comparison, search methods become extremely unstable when facing dozens of hyperparameters. IGJO converges slowly and requires huge amount of computation. Similar to the experiments on the elastic net problem, LDMMA and VF-iDCA still exhibit a certain degree of overfitting. Both LDPM and BiC-GAFFA belong to the class of single-loop Hessian-free algorithms. Since LDPM employs projection to handle nonsmooth constraints, it achieves slightly better performance and efficiency compared to BiC-GAFFA.

**D.2.3    Group Lasso**

Compared to the sparse group Lasso problem, this experiment removes the $\ell_1$-norm regularization term, leading to a reduction in the complexity of the LL problem. However, this omission also results in weaker control over the sparsity of $\mathbf{x}$, potentially affecting the structure and interpretability of the solution. While the lower computational complexity may improve efficiency, the trade-off is a less strictly enforced sparsity constraint, which could affect the ability to capture key features in high-dimensional settings.

The synthetic data is generated following the same procedure as described in Appendix D.2.2. For this experiment, we adopt the same settings for other compared algorithms as those used in the experiment for the sparse group Lasso problem in Appendix D.2.2. For LDPM, we conduct Algorithm 1 with $\beta_k = (1+k)^{0.3}$ and $e_k = 0.01$.

We conduct experiments with different data scales and report numerical results over 10 repetitions in Table 4. The overall comparison results in Table 4 are similar to those in Table 2. In this case, LDPM only requires projected gradient descent, leading to a significant improvement in efficiency.

Table 4: Group Lasso problems on the synthetic data, where $p$ represents the number of features.

| Settings | $p = 600$ | | | $p = 1200$ | | |
|---|---|---|---|---|---|---|
| | Time(s) | Val. Err. | Test Err. | Time(s) | Val. Err. | Test Err. |
| Grid | $5.72 \pm 1.69$ | $93.20 \pm 5.82$ | $96.07 \pm 17.50$ | $12.31 \pm 2.24$ | $93.15 \pm 4.74$ | $94.60 \pm 20.27$ |
| Random | $5.42 \pm 1.81$ | $148.69 \pm 6.55$ | $162.17 \pm 28.09$ | $11.38 \pm 2.56$ | $151.66 \pm 15.63$ | $160.88 \pm 17.07$ |
| IGJO | $1.42 \pm 0.25$ | $112.12 \pm 4.48$ | $105.99 \pm 15.09$ | $6.62 \pm 1.31$ | $143.62 \pm 15.42$ | $117.37 \pm 4.41$ |
| VF-iDCA | $0.50 \pm 0.14$ | $62.66 \pm 6.14$ | $84.52 \pm 12.46$ | $7.77 \pm 2.62$ | $95.02 \pm 7.04$ | $96.34 \pm 9.79$ |
| LDMMA | $0.51 \pm 0.12$ | $90.97 \pm 5.53$ | $79.68 \pm 16.19$ | $4.25 \pm 1.94$ | $92.32 \pm 8.05$ | $92.43 \pm 9.99$ |
| BiC-GAFFA | $0.35 \pm 0.02$ | $74.16 \pm 6.91$ | $78.60 \pm 11.81$ | $2.27 \pm 0.26$ | $90.43 \pm 5.53$ | $87.79 \pm 8.43$ |
| LDPM | $\mathbf{0.32 \pm 0.03}$ | $71.62 \pm 7.28$ | $\mathbf{76.43 \pm 10.34}$ | $\mathbf{1.94 \pm 0.13}$ | $89.53 \pm 7.16$ | $\mathbf{85.92 \pm 6.99}$ |

| Settings | $p = 2400$ | | | $p = 4800$ | | |
|---|---|---|---|---|---|---|
| | Time(s) | Val. Err. | Test Err. | Time(s) | Val. Err. | Test Err. |
| Grid | $21.81 \pm 3.65$ | $105.19 \pm 15.54$ | $93.35 \pm 16.60$ | $42.38 \pm 5.71$ | $141.83 \pm 26.52$ | $126.95 \pm 19.38$ |
| Random | $19.95 \pm 6.17$ | $132.04 \pm 16.90$ | $161.45 \pm 18.37$ | $41.67 \pm 5.01$ | $109.35 \pm 18.21$ | $134.74 \pm 21.41$ |
| IGJO | $10.03 \pm 6.69$ | $100.75 \pm 16.47$ | $127.58 \pm 16.43$ | $26.78 \pm 8.50$ | $109.73 \pm 16.66$ | $117.14 \pm 8.23$ |
| VF-iDCA | $12.88 \pm 1.31$ | $69.53 \pm 5.90$ | $90.11 \pm 11.59$ | $40.61 \pm 2.79$ | $81.03 \pm 11.58$ | $105.70 \pm 10.05$ |
| LDMMA | $6.75 \pm 0.19$ | $72.85 \pm 8.22$ | $87.00 \pm 15.13$ | $32.53 \pm 3.29$ | $86.47 \pm 13.55$ | $105.39 \pm 10.37$ |
| BiC-GAFFA | $4.60 \pm 0.09$ | $95.51 \pm 14.88$ | $84.02 \pm 9.46$ | $4.53 \pm 0.57$ | $103.77 \pm 9.01$ | $101.26 \pm 7.84$ |
| LDPM | $\mathbf{4.38 \pm 0.05}$ | $101.55 \pm 7.28$ | $\mathbf{81.55 \pm 3.07}$ | $\mathbf{4.12 \pm 0.15}$ | $100.49 \pm 6.64$ | $\mathbf{99.23 \pm 6.31}$ |

### D.2.4 Low-rank Matrix Completion

We consider low-rank matrix completion problem on synthetic data. The formulation in (3) of the low-rank matrix completion is given as:

$$
\begin{aligned}
\min_{\boldsymbol{\theta},\boldsymbol{\beta},\Gamma} \quad & \sum_{(i,j)\in\Omega_{val}} |M_{ij} - \mathbf{x}_i\boldsymbol{\theta} - \mathbf{z}_j\boldsymbol{\beta} - \Gamma_{ij}|^2 \\
\text{s.t.} \quad & (\boldsymbol{\theta},\boldsymbol{\beta},\Gamma) \in \arg\min_{\boldsymbol{\theta},\boldsymbol{\beta},\Gamma} \left\{ \sum_{(i,j)\in\Omega_{tr}} |M_{ij} - \mathbf{x}_i\boldsymbol{\theta} - \mathbf{z}_j\boldsymbol{\beta} - \Gamma_{ij}|^2 \right. \\
& \left. + \lambda_0\|\Gamma\|_* + \sum_{g=1}^{G} \lambda_g\|\boldsymbol{\theta}^{(g)}\|_2 + \sum_{g=1}^{G} \lambda_{g+G}\|\boldsymbol{\beta}^{(g)}\|_2 \right\}
\end{aligned}
\tag{74}
$$

The data generation procedure follows the approach in [31, 36]. Specifically, two entries per row and column are selected as the training set $\Omega_{\text{tr}}$, and one entry per row and column is selected as the validation set $\Omega_{\text{val}}$. The remaining entries form the test set $\Omega_{\text{test}}$. The row and column features are each grouped into 12 groups, with 3 covariates per group, resulting in $p = 36$ and $G = 12$.

The true coefficients are set as $\boldsymbol{\alpha}^{(g)} = g\mathbf{1}_3$ for $g = 1, \ldots, 4$ and $\boldsymbol{\beta}^{(g)} = g\mathbf{1}_3$ for $g = 1, 2$, with all other group coefficients set to zero. The low-rank effect matrix $\Gamma$ is generated as a rank-one matrix $\Gamma = \mathbf{u}\mathbf{v}^\top$, where $\mathbf{u}$ and $\mathbf{v}$ are sampled from the standard normal distribution.

The row features $X$ and column features $Z$ are also sampled from a standard normal distribution and then scaled so that the Frobenius norm of $X\boldsymbol{\alpha}\mathbf{1}^\top + (Z\boldsymbol{\beta}\mathbf{1}^\top)^\top$ matches that of $\Gamma$. Finally, the matrix observations are generated as

$$
M_{ij} = \mathbf{x}_i^\top \boldsymbol{\alpha} + \mathbf{z}_j^\top \boldsymbol{\beta} + \Gamma_{ij} + \sigma\epsilon_{ij},
$$

where $\epsilon_{ij}$ is standard Gaussian noise, and the noise level $\sigma$ is chosen such that the signal-to-noise ratio (SNR) equals 2.

In this experiment, we compare LDPM with grid serach, random search, TPE, IGJO, VF-iDCA. For grid search, we explore two hyperparameters $\mu_1$ and $\mu_2$ with the regularization parameters defined as $\lambda_0 = 10^{\mu_1}$ and $\lambda_g = 10^{\mu_2}$ for each $g = 1, \ldots, 2G$. A $10 \times 10$ grid uniformly spaced over the range $[-3.5, -1] \times [-3.5, -1]$ is employed, consistent with the approach of [31]. For both the random search and TPE methods, the optimization is conducted over transformed variables $u_g = \log_{10}(\lambda_m)$ for $m = 0, 1, 2, \ldots, 2G$, where each $u_g$ is drawn from a uniform distribution on the interval $[-3.5, -1]$. For IGJO, the initial values for the regularization vector $\boldsymbol{\lambda}$ are set to $[0.005, 0.005, \ldots, 0.005]$. For VF-iDCA, the initial guess for the auxiliary parameter $\mathbf{r}$ is chosen as $[1, 0.1, 0.1, \ldots, 0.1]$. The algorithm is terminated when the stopping criterion $(\|\mathbf{z}^{k+1} - \mathbf{z}^k\|)/\|\mathbf{z}^k\| \leq 0.1$ is satisfied. For LDPM with Algorithm 2, we set $\beta_k = (1 + k)^{0.3}$, $\gamma = 10$ and the step size $e_k = 0.025$.

Throughout all experiments, feature grouping is performed sequentially as follows, every three consecutive features are assigned to the same group, starting from the first feature onward.

We present the statistical results in repeated experiments in Table 5. Both VF-iDCA and LDPM incur longer runtimes than search methods because they perform more intensive iterative updates—VF-iDCA leverages inexact DC-programming steps to more faithfully enforce the low-rank and group-sparsity penalties. This additional computational effort yields tighter approximation of the underlying low-rank factors, resulting in substantially lower validation and test errors. LDPM repeatedly perform costly matrix projections as discussed in Appendix B.2 to enforce the rank constraints accurately. These intensive projection steps allow them to recover the underlying low-rank structure more precisely, which translates into substantially lower validation and test errors.

Table 5: Low-rank matrix completion problems on synthetic data

| Methods | Time(s) | Val. Acc. | Test Acc. |
|---|---|---|---|
| Grid | $21.02 \pm 0.95$ | $0.71 \pm 0.21$ | $0.76 \pm 0.20$ |
| Random | $33.12 \pm 2.10$ | $0.72 \pm 0.22$ | $0.79 \pm 0.19$ |
| TPE | $36.80 \pm 9.45$ | $0.69 \pm 0.20$ | $0.75 \pm 0.18$ |
| IGJO | $1205.0 \pm 312.5$ | $0.67 \pm 0.20$ | $0.71 \pm 0.17$ |
| VF-iDCA | $55.20 \pm 12.05$ | $0.65 \pm 0.18$ | $0.69 \pm 0.15$ |
| LDPM | $62.10 \pm 15.31$ | $\mathbf{0.58 \pm 0.14}$ | $\mathbf{0.66 \pm 0.13}$ |

## D.3 Sensitivity of Parameters

In this part, we conduct experiments to analyze the sensitivity of our methods to different parameter combinations. We evaluate both Algorithm 1 and Algorithm 2. To investigate the parameter sensitivity of Algorithm 1, we carry out supplementary experiments on the group Lasso problem with a problem dimension of 1200. In each trial, we vary one parameter while keeping the others fixed. The corresponding convergence times and projected gradient descent (PGD) iteration counts are summarized in Table 6a. A similar analysis is also performed for Algorithm 2 on the sparse group Lasso instance, also with a dimension of 1200. The convergence performance, including time and steps, is likewise reported in Table 6b.

| Strategy | $e_k$ | $\underline{\beta}$ | $p$ | Steps | Time(s) |
|---|---|---|---|---|---|
| Original | 0.01 | 1 | 0.3 | 29 | 2.04 |
| $e_k$ | 0.005 | 1 | 0.3 | 42 | 3.75 |
| | 0.05 | 1 | 0.3 | 18 | 1.67 |
| | 0.08 | 1 | 0.3 | 14 | 1.42 |
| $\underline{\beta}$ | 0.01 | 2 | 0.3 | 40 | 3.60 |
| | 0.01 | 10 | 0.3 | 44 | 3.89 |
| | 0.01 | 40 | 0.3 | 38 | 3.95 |
| $p$ | 0.01 | 1 | 0.05 | 95 | 11.72 |
| | 0.01 | 10 | 0.15 | 56 | 4.85 |
| | 0.01 | 40 | 0.5 | 31 | 2.93 |

(a) Parameter Sensitivity for Algorithm 1

| Strategy | $e_k$ | $\underline{\beta}$ | $p$ | $\gamma$ | Steps | Time(s) |
|---|---|---|---|---|---|---|
| Original | 0.01 | 1 | 0.3 | 10 | 36 | 2.30 |
| $e_k$ | 0.005 | 1 | 0.3 | 10 | 49 | 4.97 |
| | 0.05 | 1 | 0.3 | 10 | 21 | 1.89 |
| | 0.08 | 1 | 0.3 | 10 | 17 | 1.54 |
| $\underline{\beta}$ | 0.01 | 2 | 0.3 | 10 | 48 | 4.16 |
| | 0.01 | 10 | 0.3 | 10 | 56 | 4.35 |
| | 0.01 | 40 | 0.3 | 10 | 52 | 5.15 |
| $p$ | 0.01 | 1 | 0.05 | 10 | 129 | 16.57 |
| | 0.01 | 10 | 0.15 | 10 | 58 | 6.12 |
| | 0.01 | 40 | 0.5 | 10 | 72 | 8.83 |
| $\gamma$ | 0.01 | 1 | 0.3 | 5 | 62 | 5.12 |
| | 0.01 | 1 | 0.3 | 20 | 39 | 2.48 |

(b) Parameter Sensitivity for Algorithm 2

Table 6: Parameter Sensitivity Analysis for LDPM

In Algorithm 2, larger $\gamma$ enforces the constraint more aggressively, so the primal residual in $\mathbf{z}$-subproblem drops quickly. Smaller $\gamma$ makes $\mathbf{z}$-update more flexible, but the residual decays more slowly, so it end up needing more iterations and longer overall runtime. As presented in Table 6, the algorithm consistently achieves convergence and exhibits strong robustness across a broad spectrum of parameter configurations, highlighting its stability and reliability under varying conditions.

## D.4 Experimental on Real-world Datasets

This section of the experiments aims to demonstrate the numerical performance of our method on real-world datasets.

### D.4.1 Elastic Net

We consider elastic net problem on high dimendional datasets gisette and sensit. The mathmatical formulation follows (71). The datasets have a large number of features, which are suitable for evaluating the performance of regularization techniques like the elastic net. Following the approach in [36], we partition the datasets as follows: 50 and 25 examples are extracted as the training set, respectively; 50 and 25 examples are used as the validation set, respectively; and the remaining data was reserved for testing. For the same reasons as in Appendix D.2.1, we also compare LDPM with search method, IGJO, IFDM, VF-iDCA and LDMMA in this experiment. We conduct compared algorithms with the same settings as [36, 22]. For LDPM with Algorithm 2, we set $\beta_k = (1+k)^{0.3}$, $e_k = 0.01$ and $\gamma = 5$. The stopping criterion in this experiment is also set as $\|\mathbf{z}^{k+1} - \mathbf{z}^k\|/\|\mathbf{z}^{k+1}\| \leq 0.1$. We report the experimental results in Figure 1 and summarize them in Table 7 as auxiliary experimental results. These demonstrate that LDPM consistently achieves competitive performance while maintaining fast computational speeds on real-world datasets for elastic net problems.

As described in [36, 22], the implementation of VF-iDCA and LDMMA relies heavily on optimization solvers. In particular, the subproblems of LDMMA are entirely dependent on the commercial solver MOSEK, while the subproblems of VF-iDCA also rely on the CVXPY package, utilizing ECOS or CSC as solvers. For large-scale datasets, frequent solver calls can become a major computational bottleneck, limiting the scalability of these methods in high-dimensional or complex problem settings. Furthermore, the conic programming reformulation proposed in [22] introduces second-order cone

Table 7: Elastic net problem on datasets gisette and sensit, where $|I_{tr}|$, $|I_{val}|$, $|I_{te}|$ and $p$ represent the number of training samples, validation samples, test samples and features, respectively.

| Dataset | Methods | Time(s) | Val. Err. | Test Err. | Dataset | Time(s) | Val. Err. | Test Err. |
|---|---|---|---|---|---|---|---|---|
| | Grid | $37.21 \pm 4.80$ | $0.24 \pm 0.02$ | $0.24 \pm 0.02$ | | $1.62 \pm 0.19$ | $1.41 \pm 0.75$ | $1.33 \pm 0.47$ |
| | Random | $56.67 \pm 9.55$ | $0.22 \pm 0.05$ | $0.26 \pm 0.02$ | | $1.46 \pm 0.12$ | $1.52 \pm 0.58$ | $1.48 \pm 0.43$ |
| | IGJO | $18.24 \pm 3.17$ | $0.24 \pm 0.02$ | $0.23 \pm 0.03$ | | $0.57 \pm 0.14$ | $0.52 \pm 0.18$ | $0.61 \pm 0.14$ |
| gisette | IFDM | $35.40 \pm 0.74$ | $0.22 \pm 0.02$ | $0.23 \pm 0.03$ | sensit | $6.35 \pm 0.04$ | $0.37 \pm 0.10$ | $0.41 \pm 0.23$ |
| | VF-iDCA | $10.75 \pm 2.72$ | $0.01 \pm 0.00$ | $0.22 \pm 0.01$ | | $0.47 \pm 0.06$ | $0.27 \pm 0.03$ | $0.52 \pm 0.06$ |
| | LDMMA | $9.45 \pm 2.98$ | $0.01 \pm 0.00$ | $0.21 \pm 0.01$ | | $0.41 \pm 0.05$ | $0.25 \pm 0.04$ | $0.50 \pm 0.04$ |
| | LDPM | $4.85 \pm 0.23$ | $0.09 \pm 0.05$ | $0.14 \pm 0.03$ | | $0.28 \pm 0.02$ | $0.08 \pm 0.01$ | $0.34 \pm 0.05$ |

constraints, making LDMMA inherently a second-order algorithm. Consequently, its efficiency deteriorates significantly when applied to large-scale problems.

In this experiment, we omit the validation/test error-vs-time curves in Figure 1 for both the grid/random search methods and IFDM because their numerical instability leads to highly erratic traces. As discussed in [31, 14], implicit differentiation methods can suffer from numerical instability when applied to problems with sparse regularization like elastic net. In such cases, the inner optimization problems often have poor conditioning, causing oscillatory behavior during convergence.

### D.4.2 Smoothed Support Vector Machine

The smoothed support vector machine incorporates smoothed hinge loss function and squared $\ell_2$-norm regularization. The formulation in (3) of the smoothed support vector machine is given as:

$$
\begin{aligned}
\min_{\mathbf{x},\lambda} \quad & \sum_{i \in I_{val}} l_h(-b_i \mathbf{a}_i^T \mathbf{x}) \\
\text{s.t.} \quad & \mathbf{x} \in \arg\min_{\hat{\mathbf{x}}} \sum_{i \in I_{tr}} l_h(b_i \mathbf{a}_i^T \hat{\mathbf{x}}) + \tfrac{\lambda}{2}\|\hat{\mathbf{x}}\|_2^2,
\end{aligned}
\tag{75}
$$

where $l_h$ denotes the smoothed hinge loss function detailed in Table 1. Since there is only one regularization term in (75), we conduct LDPM using Algorithm 1 according to the discussion in Section 3.

We use the LIBSVM toolbox[4] to load the datasets and extract the corresponding observation matrix and label vector for each dataset. Each dataset is divided into two separate parts: a cross-validation training set $\Omega$ consisting of $3\lfloor N/6 \rfloor$ samples, and a test set $\Omega_{\text{test}}$ containing the remaining samples. Within this division, the training set is further partitioned into multiple equal parts, and we iteratively use one part as the validation set while utilizing the remaining parts as the training set to solve the SVM problem. For the experiments, we conducted 6-fold cross-validation on the training and validation sets across all three datasets to optimize the hyperparameters.

During the process of solving the smoothed support vector machine problem with $K$-fold cross-validation, the loss function on the validation set is defined as follows:

$$
\Theta_{val}(\mathbf{w}^1, \mathbf{w}^2, \ldots, \mathbf{w}^K, \mathbf{c}) := \frac{1}{K} \sum_{k=1}^{K} \frac{1}{|\Omega_{val}^k|} \sum_{j \in \Omega_{val}^k} l_h(b_j \mathbf{a}_j^T \mathbf{w}^k),
\tag{76}
$$

Following the approach used for support vector machine [48], we reformulate the primal problem into the following bilevel optimization model for the smoothed support vector machine:

$$
\begin{aligned}
\min_{\mathbf{w},c} \quad & \Theta_{val}(\mathbf{w}^1, \mathbf{w}^2, \ldots, \mathbf{w}^K, \mathbf{c}) \\
\text{s.t.} \quad & \lambda > 0, \bar{\mathbf{w}}_{lb} \leq \bar{\mathbf{w}} \leq \bar{\mathbf{w}}_{ub} \\
& \mathbf{w}^k \in \arg\min_{-\overline{\mathbf{w}} \leq \mathbf{w} \leq \overline{\mathbf{w}}} \left\{ \sum_{j \in \Omega_{tr}^k} l_h(b_j \mathbf{a}_j^T \mathbf{w}) + \tfrac{\lambda}{2}\|\mathbf{w}\|_2^2 \right\}, k = 1, 2, \ldots, K,
\end{aligned}
\tag{77}
$$

where $\mathbf{w}^1, \mathbf{w}^2, \ldots, \mathbf{w}^K$ are $K$ parallel copies of $c$ and $\mathbf{w}$. $\bar{\mathbf{w}}_{ub}$ and $\bar{\mathbf{w}}_{lb}$ are the upper and lower bounds of $\bar{\mathbf{w}}$. Similarly, we define the loss function on the training set in a manner analogous to (76):

$$
\Theta_{tr}(\mathbf{w}^1, \mathbf{w}^2, \ldots, \mathbf{w}^K, \mathbf{c}) := \frac{1}{K} \sum_{k=1}^{K} \frac{1}{|\Omega_{tr}^k|} \sum_{j \in \Omega_{tr}^k} l_h(b_j \mathbf{a}_j^T \mathbf{w}^k).
\tag{78}
$$

---

[4]https://www.csie.ntu.edu.tw/ cjlin/libsvmtools/datasets/

We also implement other competitive methods following the effective practice in [36, 22]. For LDPM with Algorithm 1, the penalty parameter is configured as $\beta_k = (1 + k)^{0.3}$ and the step size in each iteration is fixed at $e_k = 0.1$. We plot the convergence curves of each algorithm for validation and test error in Figure 2.

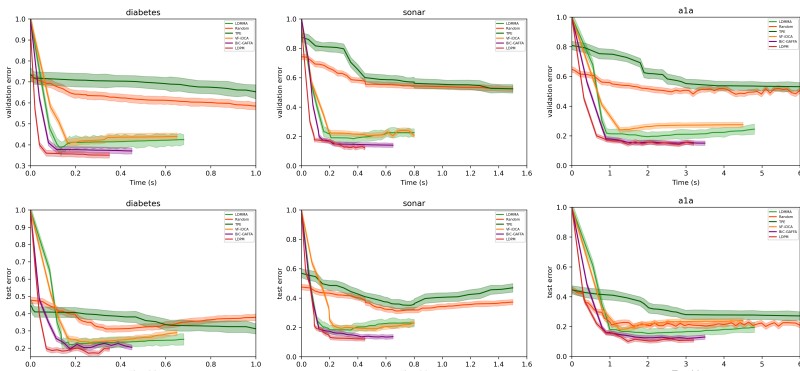

Figure 2: Comparison of the algorithms for SSVM problem on real-world datasets.

### D.4.3 Sparse Logistic Regression

The sparse logistic regression [46] is equipped with logistic loss function and $\ell_1$-norm regularization. Its formulation in (3) is

$$
\begin{aligned}
\min_{\mathbf{x}, \lambda} \quad & \sum_{i \in I_{val}} \log(1 + e^{-b_i \mathbf{a}_i^T \mathbf{x}}) \\
\text{s.t.} \quad & \mathbf{x} \in \arg\min_{\hat{\mathbf{x}}} \sum_{i \in I_{tr}} \log(1 + e^{-b_i \mathbf{a}_i^T \hat{\mathbf{x}}}) + \lambda \|\hat{\mathbf{x}}\|_1.
\end{aligned}
\tag{79}
$$

Similar to Appendix D.4.2, we also apply LDPM with Algorithm 1 in this experiment. Following the experimental setup in [15], we conduct our evaluations on large-scale real-world datasets. Specifically, we use the same datasets as [15], namely news20, rcv1 and real-sim, all of which can be downloaded from LIBSVM website[5]. Table 8 provides a brief introduction to the basic characteristics of these three datasets.

Table 8: Dataset Overview

| Datasets | Samples | Features | Sparsity | Ratio |
|---|---|---|---|---|
| news20.binary | $19,996$ | $1,355,191$ | $0.034\%$ | $0.5236$ |
| rcv1.binary | $20,242$ | $47,236$ | $0.155\%$ | $0.46948$ |
| real-sim | $72,309$ | $20,958$ | $0.245\%$ | $0.33113$ |

This experiment is initially conducted in [15]. Since VF-iDCA and LDMMA are not suitable for solving large-scale problems, and the reformulation of LDMMA is not applicable to the logistic loss function, we do not compare these algorithms in this experiment. We compare our method with search methods, IFDM, and BiC-GAFFA. Random search uniformly samples 50 hyperparameter values in the interval $[\lambda_{\max} - 4\log(10), \lambda_{\max}]$. The algorithm settings for IFDM follow the configurations in [15] for each real dataset without modification. For BiC-GAFFA, we use $\gamma_1 = 10, \gamma_2 = 0.01, \eta_k = 0.01, r = 5, \alpha_k = 0.01, \rho = 0.3$, with a maximum iteration limit of $1000$. For LDPM with Algorithm 1, we set $\beta_k = (1 + k)^{0.3}, e_k = 0.05$.

In this experiment, we implement the code provided in [15]. Each experiment is repeated 10 times to compute the average and variance of runtime, validation error, validation accuracy, test error, and test accuracy. The convergence curves of each algorithm with respect to validation and test error are illustrated in Figure 3. Additionally, we calculate the corresponding accuracy and report them in Table 9.

---

[5]https://www.csie.ntu.edu.tw/ cjlin/libsvmtools/datasets/

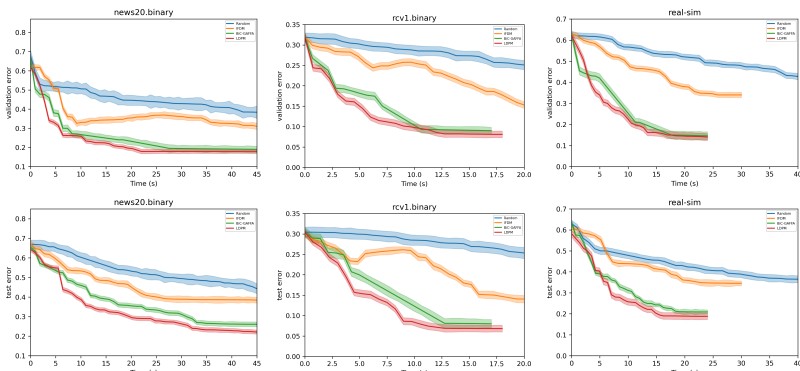

Figure 3: Comparison of the algorithms for sparse logistic regression on real-world datasets.

Table 9: Accuracy of sparse logistic regression problem on real-world datasets.

| Dataset | Methods | Time(s) | Val. Acc. | Test Acc. |
|---|---|---|---|---|
| news20.binary | Random | $654.63 \pm 33.26$ | $81.49 \pm 1.10$ | $80.89 \pm 1.24$ |
| | IFDM | $41.16 \pm 6.81$ | $86.87 \pm 1.14$ | $84.07 \pm 1.09$ |
| | BiC-GAFFA | $32.64 \pm 4.48$ | $90.98 \pm 1.03$ | $90.17 \pm 0.81$ |
| | LDPM | $30.85 \pm 3.29$ | $90.59 \pm 1.15$ | $92.94 \pm 0.73$ |
| rcv1.binary | Random | $214.46 \pm 67.15$ | $96.51 \pm 1.19$ | $94.24 \pm 2.39$ |
| | IFDM | $21.08 \pm 5.47$ | $97.95 \pm 0.26$ | $96.12 \pm 1.29$ |
| | BiC-GAFFA | $15.92 \pm 0.94$ | $98.72 \pm 0.25$ | $96.50 \pm 1.21$ |
| | LDPM | $14.13 \pm 1.43$ | $98.70 \pm 0.33$ | $97.92 \pm 1.29$ |
| real-sim | Random | $624.45 \pm 38.03$ | $68.30 \pm 1.10$ | $67.65 \pm 1.23$ |
| | IFDM | $25.86 \pm 1.57$ | $91.23 \pm 2.18$ | $91.10 \pm 1.31$ |
| | BiC-GAFFA | $18.08 \pm 0.71$ | $93.28 \pm 1.48$ | $91.68 \pm 2.42$ |
| | LDPM | $17.93 \pm 0.68$ | $95.10 \pm 1.13$ | $94.19 \pm 1.57$ |

Overall, we observe from Figure 3 and Table 9 that LDPM achieves the lowest time cost and test error in the experiment on sparse logistic regression.

The comprehensive experimental results provide strong evidence of the efficiency and practicality of our algorithm in addressing bilevel hyperparameter optimization. These results highlight its effectiveness in real-world applications, demonstrating its ability to achieve superior performance while maintaining computational efficiency.

## E    Further Discussions

LDPM effectively solves bilevel optimization problems of the form (3), as demonstrated by strong empirical results. However, the core of LDPM relies on a projected gradient descent, which currently cannot handle nonsmooth loss functions without dedicated solvers, such as the hinge loss in SVMs. In contrast, [36, 22] circumvent this issue by leveraging existing solvers to deal with such nonsmooth components.

