# OpenReview forum: "Lower-level Duality Based Penalty Methods for Bilevel Hyperparameter Optimization"
_NeurIPS.cc/2025/Conference — Submitted to NeurIPS 2025_

### Official Review · Reviewer_BK21 · 2025-06-10

**Clarity:** 2
**Significance:** 2
**Originality:** 2
**Rating:** 2
**Confidence:** 5

**Summary:**

The manuscript introduces a novel penalty method based on lower-level duality for solving bilevel hyperparameter optimization problems. The core idea is a penalty framework that leverages the strong duality between the original lower-level problem and its dual. The authors also provide convergence guarantees and empirical results.

**Questions:**

Please refer to the Weakness above.

**Ethical Concerns:**

["NO or VERY MINOR ethics concerns only"]

**Final Justification:**

# 0808

I've spent nearly a week carefully going through the proof again and provided detailed feedback on some incorrect steps in their proof. The authors have acknowledged that some key steps are wrong, which unfortunately undermines their main conclusions in Section 3. I've also carefully reviewed their responses and the details of the paper. Yet, their response to the new corrected proof sketch may lack proof details and inequalities, and I'm incapable of checking them out. And some steps are not illustrated with convincing evidence, except for some theoretical works under different assumptions and conditions.

**Given these issues on learning theory, I recommend rejection**. These factual issues in proof must be checked and corrected before acceptance.

It is important to note that *publishing a paper with flawed theoretical foundations in a widely disseminated conference like NeurIPS is highly inappropriate and could potentially mislead the research community*. The empirical validations in the submissions seem good and could be a pretty good optimization algorithm, while I have not reproduced the results yet.

**Limitations:**

yes

**Quality:**

3

**Strengths And Weaknesses:**

Strengths:

1. This work presents an interesting penalty method for nonsmooth lower-level problems in bilevel optimization. The use of lower-level duality is common in handling complex constraints like non-smoothness and non-convexity.

2. The authors provide a thorough theoretical analysis on the convergence properties of the proposed algorithms.

3. This work includes comprehensive numerical experiments that validate the efficiency and superiority of the proposed methods in several tasks, e.g., sparse regression and matrix completion.

Weaknesses:

1. Limitation in linear case. The current framework focuses on linear cases of the form \(Ax - b\). It would be beneficial to explore nonlinear scenarios like networks or sparse/generalized linear/additive models, which realizes nonlinear approximation through sparse linear modeling. Current analysis and experiments are limited in some simple scenarios.
2. Lacking clarity. The introduction and explanation of various symbols and concepts, particularly in Lemma 2.1, are not sufficiently detailed. Terms such as "convex semi-continuous functions" are introduced without adequate context on definitions or properties, which may cause confusion. A more detailed and intuitive explanation of these concepts would improve the readability and accessibility of the manuscript.

3. Lacking highlights. Upon careful examination of the literature, it appears that the current work builds incrementally on existing research [1]. The core contributions, proof challenges and differences are suggested to be highlighted.

[1] Lower-level duality based reformulation and majorization minimization algorithm for hyperparameter optimization. AISTAT 2024.

4. Weak Convergence Rates. I’ve checked part of the proof. The work seems lacking clear convergence rates for Algorithm 1 and Algorithm 2. For Algorithm 1, the analysis only discusses the boundedness for the minimizer of the function \(\phi\) over iterations \(k = 0 \ldots K\). For Algorithm 2, the convergence analysis is limited to \(\phi \to 0\) as \(K \to \infty\). Additionally, the proof of convergence in the appendix for Algorithm 2 does not clearly demonstrate the update sequence \(z_t, u_t, \mu_t \to z_{t+1}, u_t, \mu_t \to z_{t+1}, u_{t+1}, \mu_t \to z_{t+1}, u_{t+1}, \mu_{t+1}\). Clarification or correction of this aspect is necessary.

5. Comparison with Existing Work. The manuscript does not discuss or compare its methods with other first-order single-loop algorithms for bilevel optimization, such as those presented in [2,3,4,5]. A detailed comparison, especially in terms of algorithmic complexity under the \(\epsilon\)-assumption, would strengthen the manuscript's contributions and provide a clearer picture of its advantages over existing techniques.

[2] A fully first-order method for stochastic bilevel optimization

[3] Bome! bilevel optimization made easy: A simple first-order approach

[4] First-order penalty methods for bilevel optimization

[5] Fine-grained analysis of stability and generalization for stochastic bilevel Optimization

6. Suggestions on Titles. The primary focus of this work appears to be linear Lasso-type regularized problems through the transformation of the original single-level problem into a bilevel optimization framework. The current title “Lower-level Duality Based Penalty Methods for Bilevel Hyperparameter Optimization” may not be entirely appropriate, due to its limitations w.r.t. general Hyperparameter Optimization tasks with complex hyper-parameters and nonlinear scenario.

7. Question on dataset setting and primal-dual gap. The used dataset in upper level problem, named validation/meta set, is sometimes required to be drawn from the (ideally clean) test distribution for better generalization. And I wonder how the authors design the data splitting. Moreover, I would be grateful if the authors could present some analysis on the gaps between the optimizers of primal problem and its Fenchel-dual form, which would strengthen the reliability of convergence guarantee of dual-based solvers.

---

> ### Author Rebuttal · Authors · 2025-07-26
>
> Thanks for your insightful and constructive review comments! We have carefully considered all of your questions and concerns and provide point-by-point responses below. **All references are listed at the end of the rebuttal.**
>
> # Weakness 1
>
> Thank you for pointing this out! Our reformulation is based on lower-level duality. Specifically, strong duality holds when the constraints in Eq. (27) are affine, i.e., in linear cases of the form $Ax-b$. However, strong duality in the LL problem may no longer hold in **nonlinear scenarios**, which makes our framework **inapplicable** in such cases. This limitation is common in bilevel optimization settings outside of neural network applications. On the other hand, it is generally unrealistic to solve nonconvex LL problems (e.g., involving neural networks),  which further makes BLO unsolvable. We will consider this question and limitation in our future work.
>
> # Weakness 2
>
> Thanks for your reminder! The definitions of convex and lower semi-continuous functions are given as follows.
>
> **Definition (Convex Function):** Let $C \subseteq R^n$ be a convex set. A function $f: C \to R$ is called **convex** if for all $x, y \in C$ and for all $\theta \in [0, 1]$, the following inequality holds:
> $$
> f(\theta x + (1-\theta) y) \leq\theta f(x)+(1-\theta) f(y).
> $$
> **Definition (Lower semi-continuous Function):**
> A function $f:R \to R \cup \\{+\infty\\}$ is said to be **lower semi-continuous** at a point $x_0 \in R^n$ if
> $$
> \liminf_{x \to x_0} f(x) \geq f(x_0).
> $$
> Equivalently, for all $\alpha \in R$, the **sublevel set** $\\{x \in R^n : f(x) \leq \alpha\\}$ is a closed set. If $f$ is lower semi-continuous at every point in its domain, we say that $f$ is a lower semi-continuous function.
>
> These two properties are essential in our framework: convexity ensures the validity of lower-level duality (as detailed in Appendix A.1), while the lower semi-continuity guarantees the existence of a minimizer in the LL problem.
>
> # Weakness 3
>
> LDMMA differs significantly from LDPM. In the reformulation, we utilize the inner structure of $l(x)$. The methods for solving the subproblem also differ substantially: we primarily combine gradient descent and projection algorithms, whereas LDMMA directly uses off-the-shelf solver to address the subproblem. We outline the challenges faced by LDMMA in real-world applications as follows.
> 1. LDMMA cannot deal with general convex losses, e.g., the logistic loss.
> 2. In numerical experiments, each iteration of LDMMA invokes a solver based on interior point methods, which are not suitable for large-scale problems.
> 3. An important aspect of the LDMMA algorithm is the value function and an additional parameter $\epsilon$, which yields an approximation of the original BLO. LDPM directly solves the original BLO.
>
>
> As we outline in Lines 60–67, existing approaches for nonsmooth BLO employ smoothing or reformulation techniques. However, smoothing methods lead to additional computational cost. Reformulation-based strategies convert the nonsmooth BLO problem into tractable forms, but the resulting subproblems are highly complex and rely on off-the-shelf solvers in the inner loop.
>
> **In contrast, our LDPM framework overcomes both challenges.** It avoids reliance on smoothing schemes by directly handling nonsmooth components through efficient projections. Moreover, it requires only first-order differentiation of loss functions and simple projections in each iteration, avoiding the use of solvers and thereby significantly improving computational efficiency.
>
> # Weakness 4
> Thanks for your comment!
> - **Item 1:** In Theorem 3.5, the notation $\min_{0\leq k\leq K}\phi^k$ is adopted as the **standard** non-asymptotic convergence criterion . This criterion and the merit function $\phi$ is **widely used** in the convergence analysis of penalty-based single-loop first-order methods for BLO [6,7,8]. Moreover, we explain that $\phi$ is associated with the optimality condition of the penalty problem (8) and reformulation (5) in lines 190-194. The corresponding proofs is given in Appendix C.1.
>
> - **Item 2:** We appreciate the reviewer’s interest in the convergence behavior of $\phi_{res}^k$. In Theorem 3.7, we can only provide an asymptotic convergence and are unable to characterize the non-asymptotic convergence of $\phi_{\text{res}}^k$. **This limitation arises from the proof of Theorem 3.7 in Appendix C.4, detailed as follows:**
>
>     Combining the definition of $\phi_{\text{res}}^k$ with (70), we need to analyze the convergence properties of four terms: $\\|z^{k+1} - z^k\\|$, $\\|\mu^{k+1} - \mu^k\\|$, $\\|u_i^{k} - z^k\\|$, and $dist(-\sum\limits_{i=1}^{M+1}\mu_i^k, \mathcal{N}_{\mathcal{K}}(z^k))$.
>
>     Equation (63) implies that $\\|z^{k+1} - z^k\\|$, $\\|\mu^{k+1} - \mu^k\\|$, and $\\|u^k_i - z^k\\|$ exhibit a non-asymptotic convergence rate of $\mathcal{O}(1/\sqrt{k})$. For the last term, we obtain $-\mu_i^{k+1}\in\mathcal{N}_{\mathcal{K}_i\times\mathcal{K}_d^*}(u_i^{k+1})$ in line 829. However, based on the outer semi-continuity of normal cone, we can only establish the convergence property
>
>     $$\lim_{k\rightarrow\infty}dist(-\mu_i^{k},\mathcal{N}_{\mathcal{K}_i\times\mathcal{K}_d^*}(z^k))=0$$
>     in (69), without yielding an explicit non-asymptotic rate with respect to $k$.
>
> - **Item 3:** We wish to clarify that the updates of $(x,u,\mu)$ in Algorithm 2 are clearly specified in Equations (21–24), following an alternating scheme with updating each variable by fixing the other two. The detailed procedure is as follows:
>
> 1. **Update $z^{k+1}$** with Equation (21):
>   $(z^k, u^k, \mu^k) \rightarrow (z^{k+1}, u^k, \mu^k)$, after which Equation (59) holds.
>
> 2. **Update $u^{k+1}$** with Equation (23):
>   $(z^{k+1}, u^k, \mu^k) \rightarrow (z^{k+1}, u^{k+1}, \mu^k)$, after which Equation (57) holds.
>
> 3. **Update $\mu^{k+1}$** with Equation (24):
>   $(z^{k+1}, u^{k+1}, \mu^k) \rightarrow (z^{k+1}, u^{k+1}, \mu^{k+1})$, after which Equation (58) holds.
>
> We will make these more explicit in the revised version.
>
> # Weakness 5
>
> The assumptions in [2,3,5] do not hold in nonsmooth settings, as all of them require the LL problem to be smooth. This is explicitly stated in Assumption 2 of [2], Assumption 2 of [3] and Assumption 2 of [5].
> [4] proposes first-order penalty methods for two types of bilevel problems: unconstrained (Section 2) and constrained (Section 3). Only the unconstrained setting aligns with our problem. However, the assumptions in Section 2 of [4] still require the LL objective to be smooth.
>
> We will cite these works and show why they do not apply to our setting.
>
> # Weakness 6
>
> Lasso-type problems represent only a subset of the problem classes we consider. As summarized in Table 1, our study also encompasses low-rank matrix completion, smoothed SVM, and logistic regression, with corresponding experiments presented in Section 4. Therefore, we believe the current title is appropriate for our paper.
>
> # Weakness 7
>
> - **Item 1 Dataset Splitting:** Due to space constraints, the details of dataset splitting procedure are provided in the Appendix D. We summarize them as follows.
> 1. For elastic net, we follow the approach in [1,2] and claim it in lines 1017-1019. Gisette and SensIT datasets are split into training (50 and 25 samples), validation (50 and 25 samples), and test sets (remaining data).
> 2. For smoothed SVM, we follow the approach in [1,2] and claim it in lines 1046-1050. Each dataset is split into a cross-validation training set $\Omega$ with $3\lfloor N/6\rfloor$ samples and a test set $\Omega_{test}$ with the remaining samples. The training set is further divided into equal parts for iterative validation.
> 3.  For sparse logistic, We follow the strategy in [3]: 20\% of the entire dataset is randomly selected as the test set. The remaining 80\% is further randomly split into training and validation sets in an 80/20 ratio. This information is omitted and we will include it in the revised version.
>
> - **Item 2: gaps between the optimizers of primal problem and its Fenchel-dual form**
>
>     We appreciate the reviewer’s comments.   In our approach, we in fact solve an exact reformulation of the bilevel problem by replacing the lower‐level subproblem with its Fenchel conjugate (dual) representation. In Lemma 2.1, strong duality guarantees a zero duality gap between the primal and dual formulations. Consequently, the unique minimizers coincide.
>
>     Moreover, by leveraging this exact reformulation, our algorithm becomes a single‑loop, first‑order method: at each iteration we perform only gradient updates and efficient epigraphic projections, without any inner‑outer loops or nested solves. This structure not only preserves the exactness of the reformulation but also yields a provably lower per‐iteration cost compared to multi‑loop dual solvers that require solving subproblems to high accuracy.
>
> # Reference
>
> [1] Chen et al. Lower-level duality based reformulation and majorization minimization algorithm for hyperparameter optimization.
>
> [2] Kwon et al. A fully first-order method for stochastic bilevel optimization.
>
> [3] Liu et al. Bome! bilevel optimization made easy: A simple first-order approach.
>
> [4] Lu et al. First-order penalty methods for bilevel optimization.
>
> [5] Zhang et al. Fine-grained analysis of stability and generalization for stochastic bilevel optimization.
>
> [6] Yao et al., Overcoming lower-level constraints in bilevel optimization: A novel approach with regularized gap functions.
>
> [7] Liu et al., Moreau envelope for nonconvex bi-level optimization: A single-loop and hessian-free solution strategy.
>
> [8] Yao et al. Constrained bi-level optimization: Proximal lagrangian value function approach and hessian-free algorithm.
>
> [9] Gao et al. Value function based difference-of-convex algorithm for bilevel hyperparameter selection problems.
>
> [10] Bertrand et al. Implicit differentiation for fast hyperparameter selection in non-smooth convex learning.

---

> > ### Comment · Reviewer_BK21 · 2025-08-03
> > **Response to Authors**
> >
> > I have read the rebuttal and appreciate the discussions. I am also curious to see the discussions of the Assumption 3.2 and comparisons on  smooth baselines concerned by another reviewer. So I am not yet ready to finalize my review. I will continue to checkproof the paper over the next few days and let the authors know if I have any more specific questions.

---

> ### Author Response · Authors · 2025-08-03
>
> Thank you for your thoughtful consideration and for engaging with our rebuttal and discussions. We appreciate your continued efforts in checking our submission. If any further questions arise during your continued review, we would be happy to clarify and discuss. We look forward to further discussions.

---

> ### Comment · Reviewer_BK21 · 2025-08-06
> **More Concerns on Proof**
>
> I found the theoretical aspects of this paper particularly fascinating! While reviewing the proofs during my spare time, I noticed a few points that might need clarification. These observations came to me at different times, so they're not strictly in the order of the proofs themselves.
>
> ## 1. In Theorem 2.4's Proof
> I was following the logic where p(z̄) + A(z̄) ≤ 0 is derived from inequality (31), but this step seems to make a couple of implicit assumptions. First, it appears to take for granted that z* is feasible for (6), though this isn't explicitly shown. Also, the convergence of L(x^{k+1}) to L(x*) is used without much explanation. Wouldn't the argument need to establish these points more clearly to be fully convincing?
>
> ## 2. Issue in Lemma C.2's Inequality
> Looking at inequality (43), it suggests V_{k+1} - V_k is always negative, but I'm not entirely sure about those coefficients. For example, take (\alpha_L + \beta_k‖A_t‖² \alpha_p)/2\beta_k - 1/e_k\beta_k - is this negative for every k? If e_k isn't chosen carefully, some terms might flip positive. Maybe the proof could benefit from being more explicit about the constraints on e_k?
>
> ## 3. Unjustified Constant in Theorem 3.5's Proof
> When the proof jumps from (54) to (55), it introduces the C_res constant out of nowhere. The relationship claims (1/\beta_k^2) \Phi^k_res(z^{k+1})² ≤ (C_res/e_k \beta_k)‖z^{k+1} - z^k‖², but how exactly does C_res behave with respect to \beta_k and e_k? If it depends on them in certain ways, the bound might not hold up. Would it be possible to define C_res more precisely?
>
> ## 4. Assumption in Theorem 3.7's Proof
> Inequality (60) seems to assume M_k ≤ 1/e_k, but M_k is defined as a maximum of several terms. What ensures this inequality always holds? If M_k happens to be larger than 1/e_k, wouldn't that reverse the disparity? Perhaps the argument needs to either tighten the conditions on e_k or adjust this part of the reasoning.
>
> ## 5. Unverified Non-Negativity in Eq.(48)
> The proof states V_K ≥ 0 based on L(x^k) ≥ 0 and p(z^k) ≥ 0, but I didn't see where p(z^k)'s non-negativity is actually established. If p(z^k) could be negative, then V_0 - V_K ≤ V_0 might not hold. Would it be possible to either prove p(z^k) ≥ 0 or modify the inequality accordingly?
>
> ## 6. Dubious Lower Bound in Eq.(63)
> The lower bound L_b = -((M+1)M_μ)/2\gamma \beta relies on ‖μ^k‖ being bounded by M_μ, but is this bound always satisfied? If ‖μ^k‖ grows indefinitely, L_b becomes meaningless, and telescoping (60) to (63) fails. Maybe the proof could either demonstrate ‖μ^k‖'s boundedness or find another way to structure this part?
>
> **In short, the proofs may contain multiple gaps where inequalities might be either unjustified or rely on unverified assumptions. These issues may not be comprehensive and are only those I have identified. Please correct me if I'm wrong. Otherwise, these issues are required to be solved before acceptance.**

---

> > ### Author Response · Authors · 2025-08-07
> >
> > Once again, we sincerely appreciate your continued engagement with our work, as well as your careful examination of our proof and insightful questions. We have addressed all your questions and concerns point by point in the above block. Your efforts have significantly improved the presentation and correctness of our work. We will incorporate the corresponding revisions based on your comments and our responses in the revised version. If you have any further questions or comments, we would greatly appreciate the opportunity to continue the discussion.

---

> > ### Author Response · Authors · 2025-08-09
> >
> > Dear Reviewer BK21:
> >
> > We hope this message finds you well.
> >
> > As the author–reviewer discussion period ends in less than 12 hours, we would be most grateful if you could let us know whether our replies have adequately addressed the concerns regarding the proof raised in your official comment. If any concerns remain, we would appreciate it if you could share them at your convenience so that we have sufficient time to provide a detailed response before the discussion window closes.
> >
> > We are very grateful for your time and constructive feedback, and for your continued effort in reviewing our work.
> >
> > Best regards,
> > Authors

---

> ### Author Response · Authors · 2025-08-07
>
> We sincerely thank you for your careful review of our proofs and the insightful further discussion. Your detailed feedback has substantially enriched and strengthened our theoretical development and has helped us improve the clarity of our presentation.
> # Q1
> **Item 1:** As stated in Line 584, $z^*$ is defined as a solution of reformulation (6), and is therefore necessarily feasible for (6).
>
> **Item 2:** We would like to clarify that the inequality
> $p(\bar z) + A(\bar z) \le 0$ derived from (31) **follows from the lower semi-continuity property of $p(z)$, the continuity of $A(z)$ together with the fact that** $\beta_k \to \infty$, **rather than** from the convergence of $L(x^{k+1})$. Specifically:
> 1. Since $p$ is lower semi-continuous and $A$ is continuous in $z$, we have
> $$p(\bar z)+A(\bar z)\le \lim_{k\rightarrow\infty}p(z^{k+1})+A(z^{k+1})$$
> 2. From $\beta_k\rightarrow\infty$, we have $\frac{1}{\beta_k}(L(x^\*)-L(x^{k+1}))\rightarrow0$ as $k\rightarrow\infty$
>
> Combining these facts, (31) gives
> $$p(\bar z)+A(\bar z)\le \lim_{k\rightarrow\infty}p(z^{k+1})+A(z^{k+1})\le\lim_{k\rightarrow\infty}\frac{1}{\beta_k}(L(x^*)-L(x^{k+1}))=0.$$
> # Q2
> The condition on $e_k$ ensuring the nonnegativity of the coefficients in (43) is explicitly stated in Line 763 of Lemma C.2. When $e_k\le\min\\{\frac{2}{\alpha_L+\beta_k\\|A_t\\|_2^2\alpha_p},\frac{2}{\beta_k(\alpha_d+\\|A_t\\|_2^2)},\frac{2}{\beta_k}\\}$, the nonnegativity follows directly from a straightforward calculation.
> The choice of $e_k$ in Theorem 3.5 also clearly satisfies the above inclusion.
> # Q3
> Thank you very much for highlighting this detail. Consistent with Theorem 3.7, the admissible range of $e_k$ in Theorem 3.5 **should be revised to**: $0<\frac{\underline{e}}{\underline{\beta}}\le e_k\le\min\\{\frac{1}{\alpha_L+\beta_k\\|A_t\\|_2^2\alpha_p},\frac{1}{\beta_k},\frac{1}{\beta_k(\alpha_d+\\|A_t\\|_2^2)}\\}$. The lower bound $\underline{e}$ is analogous to Theorem 3.7 and is **commonly utilized in single-loop Hessian-free algorithms for BLO, as discussed in Lines 247–248**.
>
> Next, we clarify the step from (54) to (55). From (54), we have $\phi_{res}^k(z^{k+1})^2\le(\beta_k L_z+1/e_k)^2\\|z^{k+1}-z^k\\|^2.$ Since $\beta_k\le 1/e_k$, it follows that $\phi_{res}^k(z^{k+1})^2\le(L_z+1)^2\frac{1}{e_k^2}\\|z^{k+1}-z^k\\|^2.$
> Therefore,
> $$\frac{1}{\beta_k^2}\phi_{res}^k(z^{k+1})^2\le(L_z+1)^2\frac{1}{\beta_k^2e_k^2}\\|z^{k+1}-z^k\\|^2.$$
> 1. Invoking the revised admissible range for $e_k$, we deduce that $\frac{1}{e_k\beta_k}=\frac{1}{e_k\underline{\beta}(1+k)^p}\le\frac{1}{e_k\underline{\beta}}\le\frac{1}{\underline{e}}$.
> 2. Furthermore, the definition of $L_z$ in Line 792 implies that $$L_z\le\max\\{\frac{\alpha_L}{\underline{\beta}}+\\|A_t\\|^2_2\alpha_p,\\|A_t\\|^2_2+\alpha_d,1\\}\overset{\triangle}{=}R_z.$$
>
> Based on the above inclusions, we may set $C_{res}=\frac{(R_z+1)^2}{\underline{e}}$. **This $C_{res}$ is a constant independent of $e_k$ and $\beta_k$**. Hence, we conclude that
> $$\frac{1}{\beta_k^2}\phi^k_{res}(z^{k+1})\le\frac{C_{res}}{e_k\beta_k}\\|z^{k+1}-z^k\\|^2$$
> as stated in (55).
>
> # Q4
> In Theorem 3.7, the step size $e_k$ satisfies
> $$0<e_k<\min\\{\frac{\beta_k}{\alpha_L+\beta_k\\|A_t\\|_2^2\alpha_p},\frac{1}{\alpha_d+\\|A_t\\|_2^2},1\\},$$
> and $M_k$ is defined in Line 809 as
> $$M_k=\max\\{\frac{\alpha_L+\beta_k\\|A_t\\|_2^2\alpha_p}{\beta_k},\alpha_d+\\|A_t\\|_2^2,1\\}>0.$$
> Since each entry of $M_k$ is **exactly the reciprocal** of the corresponding upper bound on $e_k$, the inequality $M_k<1/e_k$ holds automatically **without any additional assumptions**.
> # Q5
> We wish to clarify that the non-negativity of $p(z)$ **is established in Line 592**. Combining the definition of $p(z)$ in (7) with the equivalence in (29-30), we have
> $$p(z)=l(x)+\sum_{i=1}^{M+1}\lambda_iR_i(x)-\min_x\\{l(x)+\sum_{i=1}^{M+1}\lambda_iR_i(x)\\},$$
> which directly implies that $p(z)\geq0$. This property follows directly from the construction of the reformulation. **Consequently, $p(z)$ remains non-negative throughout the paper. We will make this point more explicit in the revised version.**
> # Q6
> The boundedness of $\mu_k$ is directly ensured by Assumption 3.6. As discussed in [1,2], the convergence of nonconvex nonsmooth ADMM is highly challenging without imposing assumptions like Assumption 3.6, which is an **open question. This assumption is widely employed in ADMM approaches** [3,4,5,6].
>
> [1] Wang et al. Global convergence of ADMM in nonconvex nonsmooth optimization
>
> [2] Lin et al. Alternating direction method of multipliers for machine learning
>
> [3] Bai et al. An augmented lagrangian decomposition method for chance-constrained optimization problems
>
> [4] Cui et al. Decision making undercumulative prospect theory: An alternating direction method of multipliers
>
> [5] Shen et al. Augmented lagrangian alternating direction method for matrix separation based on low-rank factorization
>
> [6] Xu et al. An alternating direction algorithm for matrix completion with nonnegative factors

---

> ### Comment · Reviewer_BK21 · 2025-08-09
>
> Thanks for further clarification, which alleviates partial concerns regarding the theory. I've updated the rate with the Final Justification based on the responses.

---

> > ### Author Response · Authors · 2025-08-09
> >
> > We sincerely appreciate your continued effort and careful attention in reviewing our work. Your constructive feedback has materially improved the quality of the manuscript. We are also grateful that you updated your score. We will re-check our theoretical derivations and results thoroughly. We will incorporate your suggestions and revise our work accordingly.

---

### Official Review · Reviewer_NsoD · 2025-06-11

**Clarity:** 3
**Significance:** 2
**Originality:** 3
**Rating:** 4
**Confidence:** 4

**Summary:**

This paper focuses on the hyperparameter optimization (HO) problem and adopts a bilevel optimization (BLO) framework to address it. Specifically, it considers a setting where the training loss includes both empirical loss and regularization terms, and the hyperparameters are the coefficients associated with the regularizers. The lower-level (LL) objective is defined as the combination of the loss and regularization, while the upper-level objective is a validation loss, with practical applications summarized in Table 1.

Assuming that the LL objective is strongly convex (as implicitly specified in Assumption 3.2), the paper reformulates the BLO problem into a constrained optimization problem (Equation 6), treating the LL optimality condition and non-negativity of hyperparameters as constraints. To avoid direct projection or inner optimization, the authors then penalize the non-projection-friendly parts of this reformulated problem to construct a new objective F_k(z), whose minimizers asymptotically approximate the solution to the constrained formulation.

The paper proposes a novel penalty-based formulation that avoids the need for inner minimization or maximization, unlike many existing methods such as value-function-based approaches. Two common HO scenarios are considered: (1) separable regularizers and (2) non-separable regularizers. For the separable case, the authors propose a fully single-loop algorithm with complexity O(\epsilon^{-4}) when choosing p = 1/4; for the non-separable case, they also propose a fully single-loop algorithm and prove its asymptotic convergence. Experiments compare the proposed LDPM algorithm with existing HO methods and demonstrate its superior efficiency.

**Questions:**

1.	Clarity of Assumptions and Theoretical Foundations:

• Assumption 3.2 is unnecessarily complicated via the use of duality-based arguments. Given that the assumption essentially requires the strong convexity of the lower-level objective, why not state this directly? This would significantly improve transparency and reader comprehension. Is there a specific reason for avoiding a direct assumption of strong convexity?

• More broadly, what is the practical scope of this assumption? Would the proposed method still apply under general convexity?

2.	Which algorithm is referred to as LDPM in the main theorems—Algorithm 1 or Algorithm 2? This should be clearly linked in the theoretical analysis.

3.	Terminology and Technical Presentation

•	What assumptions are required for Lemma 2.1 and Proposition 2.3? These are not explicitly stated.

•	In Lemma 2.1 (line 99), the term “equivalent form” is used. Does this refer to equivalence in the set of minimizers only, or also in function values? Please clarify.

•	What is the parameter p in Theorem 3.5, and how does it relate to convergence? This should be explained before being used in the complexity result.

4.	Related Work and Motivation:

•	The related work section should be more concise and focused. It currently reviews some less relevant works in excessive detail without a clear narrative.

Suggested structure:

• Introduce HO problems and the need for BLO formulations.

• Categorize BLO methods into: implicit gradient descent, explicit gradient tracking, and penalty-based approaches.

• For penalty methods, distinguish between value-function-based [2], Moreau-envelope-based [3], and the current method, etc. Discuss computational trade-offs.

• Include some more relevant work [1], [2], and [3]. And comment that if they either struggle with constraints, require projections, or incur inner-loop costs. Then highlight how this paper overcomes these challenges.

5.	Experimental Evaluation:

•	Are the experiments in Section 5 conducted under separable or non-separable regularizer settings? Please specify.

•	How does the performance of LDPM compare in terms of complexity and runtime with other BLO methods like [2] and [3]?

•	Is it possible to adopt these existing BLO methods to solve the same HO problem? If so, please discuss why LDPM is preferable.

•	Please also clarify how this work differs from [4], particularly in terms of formulation, algorithmic complexity, and applicability.



[1] Xu, Siyuan, and Minghui Zhu. "Efficient gradient approximation method for constrained bilevel optimization." In Proceedings of the AAAI Conference on Artificial Intelligence, vol. 37, no. 10, pp. 12509-12517. 2023.

[2] Jiang, Liuyuan, Quan Xiao, Victor M. Tenorio, Fernando Real-Rojas, Antonio G. Marques, and Tianyi Chen. "A primal-dual-assisted penalty approach to bilevel optimization with coupled constraints." arXiv preprint arXiv:2406.10148 (2024).

[3] Shen, Han, and Tianyi Chen. "On penalty-based bilevel gradient descent method." In International Conference on Machine Learning, pp. 30992-31015. PMLR, 2023.

[4] Chen, H., Xu, H., Jiang, R. and So, A.M.C., 2024, April. Lower-level duality based reformulation and majorization minimization algorithm for hyperparameter optimization. In International Conference on Artificial Intelligence and Statistics (pp. 784-792). PMLR.

**Ethical Concerns:**

["NO or VERY MINOR ethics concerns only"]

**Final Justification:**

Most of my questions are resolved. I am keeping my positive score.

**Limitations:**

Yes.

**Quality:**

2

**Strengths And Weaknesses:**

Strengths:

• Presents a novel penalty-based formulation for hyperparameter optimization that avoids explicit inner-loop optimization.

• Proposes a fully single-loop method (LDPM) with convergence guarantees under separable regularization.

• Includes a theoretical analysis and experiments that support the proposed approach.

Weaknesses:

• The strongly convexity assumption, which is critical for the theoretical analysis, is not clearly presented. Instead, it is obscured through a less direct argument involving double duality in Assumption 3.2. This significantly hinders readability and makes it difficult to assess the generality and implications of the results.

• The related work section lacks focus and does not clearly motivate the methodological gap this paper aims to address.

• Some assumptions and definitions (e.g., DC methods, LDPM naming, Lemma 2.1 assumptions) are insufficiently explained or buried in notation.

• The claimed “equivalence” between reformulated problems is not rigorously distinguished between pointwise optimality and function value equality.

---

> ### Author Rebuttal · Authors · 2025-07-26
>
> Thanks for your insightful and constructive review comments! We have carefully considered all of your questions and concerns, and provide our point-by-point responses below.
>
> # Question 1 & Weakness 1: Assumption 3.2
>
> We appreciate the reviewer for this valuable comment. We agree that Assumption 3.2 should be clarified to specify as follows.
> >The function $\varphi$ and its conjugate $\varphi^*$ are $\alpha_p$- and $\alpha_d$-Lipschitz smooth on any **compact subset** in their domains, respectively.
>
> - This condition rather implies the strong convexity of $\varphi$ and $\varphi^\*$ on some compact subset in their domains. Specifically, if $\varphi^\*$ is Lipschitz smooth on a compact set $K$, then $\varphi$ is strongly convex on the compact set $\nabla\varphi^\*(K)\subset dom(\varphi)$. Likewise, if $\varphi$ is Lipschitz smooth on a compact set, then $\varphi^\*$ is strongly convex on the corresponding image set.
>
>     The inclusion $\nabla\varphi^\*(K)\subset dom(\varphi)$ follows from the relation $\nabla\varphi^*=(\nabla\varphi)^{-1}$.
> - We would like to emphasize that **the corrected assumption is also more general than the strong convexity of LL objective**, and in fact **does not** force the LL objective to be strongly convex. For example, the function $l(x)=\varphi(A_tx-b_t)$ is convex but not strongly convex when $A_t^TA_t$ is singular.
>
>     Therefore, we do not require any additional assumption of strong convexity on the lower-level loss $l$ or the regularizers $R_i$. This is consistent with explicit clarification in **Remark 2.2** and **Remark 3.4**. We will make this point clearer in the revised version.
>
> - We illustrate with the examples in Table 1 that the above assumption holds in every case. Specifically, referring to the explicit forms of $\varphi$ in Lines 92–94 and its conjugate $\varphi^*$ in Lines 600–602, we verify the following:
>
>     - **Least Squares Loss:** Both $\varphi$ and $\varphi^*$ are $\tfrac12$–strongly convex and 1–Lipschitz smooth.
>
>     - **Smoothed SVM:** $\varphi$ is $\tfrac12$–Lipschitz smooth and only $\tfrac12$–strongly convex on the interval $[0,1]$. $\varphi^*$ is 1–Lipschitz smooth and $\tfrac12$–strongly convex.
>
>     - **Logistic Loss:** $\varphi$ is $\tfrac14$–Lipschitz smooth and strongly convex on any bounded interval in $R$. $\varphi^*$ is Lipschitz smooth and strongly convex on the interval $[\epsilon, 1-\epsilon]$ for any $\epsilon>0$.
>
> - According to the examples and explanations above, we observe that, in the smoothed SVM case, $\varphi$ is strongly convex only on a single interval $[0,1]$ rather than on every compact subset of its domain. Therefore, the direct assumption of strong convexity for $\varphi$ and its conjugate $\varphi^*$ on any compact subset of their domains is **not appropriate** for our settings and examples. More importantly, since our reformulation framework and theoretical analysis do not directly rely on the strong convexity of $\varphi$, we have not adopted it in our analysis.
>
> # Question 2: the name LDPM
>
> Thank you for this helpful suggestion. In our paper, LDPM is the overarching name we give to the penalty‐based reformulation procedure introduced in Section 2. Algorithms 1 and 2 are the concrete solvers for subproblem (8) under this penalty framework, each tailored to different regularizers. In our experimental descriptions, we typically refer to the proposed method as “LDPM with Algorithm 1 or 2”, as detailed in lines 900, 937, 985, 1021, 1059 and 1065.
>
> To improve clarity and coherence, we will consistently emphasize the name LDPM in Sections 2 and 3 in the revised version.
>
> # Question 3 & Weakness 2/3
>
> -  The assumptions of Lemma 2.1 are given in the first part of this lemma. We assume that $l$ and $R_i$ in LL problem are convex, proper lower‑semicontinuous functions, and that Slater’s condition holds for LL problem. Proposition 2.3 builds directly on Lemma 2.1 and relies on exactly the same assumptions, which were missing; we will add these assumptions in a revision.
>
> -  In Lemma 2.1, the phrase **equivalent form** denotes equivalence **both** in the set of minimizers and in the optimal objective value. Under Slater’s condition, strong duality holds between the LL problem in (3) and its Fenchel dual, guaranteeing that their optimal values coincide. Concretely, our equivalence is from relation
> $$l(x)+\sum\lambda_iR_i(x)\leq\min_x\\{l(x)+\sum\lambda_iR_i(x)\\}.$$
> We reformulate it by replacing the right part with its Fenchel dual in Equation (29-30). Therefore, the constraint enforces recovery of the LL solution and its dual multipliers introduced for (27). As a result, if $(x,\lambda,\rho,\xi)$ is the minimizer of (5), then $(x,\lambda)$ is the minimizer of (3). Moreover, since the objective $L(x)$ remains unchanged, the optimal value is also preserved. We will clarify this in the revision
>
> - We would like to clarify the parameter $p$ is explicitly defined in Lines 178–179 when introducing Algorithm 1. The parameter $p$ serves as the exponent in the penalty schedule. Specifically, the penalty parameter is updated as $\beta_k = \underline{\beta}(1+k)^p$, where $0<p<1/2$.
> We thank the reviewer for the suggestion and will revise the text around Theorem 3.5 to explicitly restate the definition of $p$ for improved readability.
>
> - The DC method (in line 62) refers to the Difference-of-Convex algorithm, which solves nonconvex problems by decomposing the objective into the difference of two convex functions and iteratively approximating it via convex subproblems. We will include the full term "Difference-of-Convex" (DC) in our revision.
>
> # Question 4
>
> Thanks for your constructive comments! We will revise our manuscript following your suggestion!
>
> # Question 5
>
> - **Item 1:** We thank the reviewer for this question. Due to space constraints, we have placed the relevant details in the Appendix D. Specifically, the separability of the regularizer and the corresponding choice of algorithm are explained as follows:
>
> 1. Elastic Net: The problem formulation is given in Equation (71) (Line 880), which involves non-separable regularizers. We use Algorithm 2, as noted in Line 901.
>
> 2. Sparse Group Lasso: The formulation appears in Equation (72) (Line 882), also with non-separable regularizers. Accordingly, we apply Algorithm 2, as stated in Line 937.
>
> 3. Group Lasso: Presented in Equation (73) (Line 885), this regularizer is separable, and we employ Algorithm 1. This is clarified in Line 956.
>
> 4. Low-Rank Matrix Completion: Described in Equation (74) (Line 963), which involves non-separable regularizers. We use Algorithm 2, as explained in Line 985.
>
> 5. Smoothed SVM: The problem is defined in Equation (75) (Line 1041), where the regularizer is separable. Thus, Algorithm 1 is used, as indicated in Line 1059.
>
> 6. Sparse Logistic Regression: Shown in Equation (79) (Line 1064), where the regularizer is separable. We apply Algorithm 1, as clarified in Line 1076.
>
> - **Item 2 & 3:** Thanks for your question! The method proposed in [2] focuses on bilevel optimization problems with coupled constraints and explicitly assumes that both the UL and LL objectives are **smooth**, as specified in Assumption 1 of [2]. Consequently, it is not applicable to bilevel hyperparameter optimization involving **nonsmooth regularization**. In addition, the experiments in [2] are limited to fully smooth settings, which makes a direct comparison with our method not meaningful.
>
>     Similarly, the method in [3] assumes that the UL objective is **Lipschitz continuous**, and that the LL problem is **differentiable** and satisfies the **Polyak–Lojasiewicz (PL) condition**. However, these assumptions do not hold for the bilevel hyperparameter optimization problems considered in our work. Therefore, the method in [3] is not directly applicable to our setting and a direct comparison may not be meaningful.
>
>     [2] Jiang et al. A primal-dual-assisted penalty approach to bilevel optimization with coupled constraints. arXiv preprint arXiv:2406.10148 (2024).
>
>     [3] Shen et al. On penalty-based bilevel gradient descent method. In International Conference on Machine Learning, pp. 30992-31015. PMLR, 2023.
>
> - **Item 4:**  LDMMA in [4] differs significantly from our proposed LDPM. In the reformulation, we utilize the inner structure of $l(x)$, which is different from LDMMA. The methods for solving the subproblem also differ substantially: we primarily combine gradient descent and projection algorithms, whereas LDMMA directly uses an off-the-shelf solver to address the subproblem. The following outlines the key challenges faced by LDMMA in real-world applications.
>
> 1. A key requirement for applying LDMMA is conic programming reformulation of the original problem, which necessitates expressing all objective functions and constraints in conic or linear forms. LDMMA cannot deal with general convex losses, e.g., the logistic loss.
>
> 2. In numerical experiments, each iteration of LDMMA invokes a solver based on interior point methods, which are not suitable for large-scale problems. Consequently, the efficiency of LDMMA decreases significantly when handling large-scale problems.
>
> 3. An important aspect of the LDMMA algorithm is the value function and an additional parameter $\epsilon$, which yields an approximation of the original BLO. LDPM directly solves the original BLO.
>
>     Indeed, LDPM **addresses the above challenges** and demonstrates improved numerical performance.
>
>     [4] Chen et al. Lower-level duality based reformulation and majorization minimization algorithm for hyperparameter optimization. In International Conference on Artificial Intelligence and Statistics (pp. 784-792). PMLR.

---

> > ### Comment · Reviewer_NsoD · 2025-08-01
> >
> > **1. Assumption 3.2**
> >
> > This updated assumption, however, is still nearly as restrictive as strongly convexity. In my point of view, assuming strong convexity (or SC in effective domain) is fine but I think it is very difficult to read if you are hiding the strong convexity under tedious uncommon assumptions such as smooth conjugate.
> >
> > **2. Experiment using [2],[3]**
> >
> > Although the analysis of other BLO alogrithm, e.g. [2], [3] usually requires smoothness, they can empirically be implemented via subgradient. So it is unclear of the motivation of inventing a method specially designed for "non-smooth" questions.

---

> > > ### Author Response · Authors · 2025-08-02
> > >
> > > Thanks for your questions!
> > >
> > > # Assumption 3.2
> > >
> > > 1. As discussed in our response to **Question 1 \& Weakness 1**, we are **unable to identify a unified characterization** of the regions in which the strong convexity holds for $\varphi$ and $\varphi^*$, which makes it **difficult to accurately describe** their strong convexity. In contrast, we **can uniformly describe** the regions in which the Lipschitz-smoothness holds for $\varphi$ and $\varphi^\*$ (any compact subset of their domains).
> > >
> > > 2. In our reformulation and penalty framework, as well as in the design of our algorithm and its convergence analysis, we **do not invoke** the strong-convexity of $\varphi$ or $\varphi^*$. We **rely solely on** their Lipschitz-smooth properties.
> > >
> > > In summary, we believe that assuming Lipschitz smoothness is better suited to our problem and settings.
> > >
> > > **To improve clarity, we will fully incorporate your suggestions and include in the revised manuscript a detailed remark on this assumption, explaining its implication of strong convexity as well as the related discussions provided in our rebuttal.**
> > >
> > > # Experiment using [2],[3]
> > >
> > > Subgradient methods are a classical tool in nonsmooth optimization. However, simply replacing gradients with subgradient in smooth bilevel algorithms [2,3] is generally not applicable to nonsmooth bilevel hyperparameter optimization, and faces several critical limitations.
> > >
> > > - **Reduced computational efficiency.** Simply replacing the gradient descent updates in [2,3] with subgradient descent leads to a significant drop in computational efficiency, primarily because subgradient methods require far more iterations and involve complex direction selection and step‐size strategies.
> > > - **Broken convergence guarantees.** In [2], convergence relies on the Lipschitz‐smooth assumption (see Assumption 1 in [2]), which no longer holds for nonsmooth functions and subgradient methods. In [3], the convergence analysis invokes the nonconvex PL condition, yet there is no established PL analogue for fully nonsmooth functions.
> > >
> > > In contrast, our approach handles the nonsmooth term via epigraphic projection, which incurs lower computational cost. In practice, it runs significantly faster and more reliably than repeated subgradient steps. Moreover, our method comes with rigorous theoretical convergence guarantees.

---

> > > > ### Comment · Reviewer_NsoD · 2025-08-04
> > > >
> > > > We thank the reviewer for response. I will keep my positive score

---

> ### Author Response · Authors · 2025-08-05
>
> We sincerely appreciate your thoughtful comments and your effort in reviewing our work. We are grateful for your positive feedback. Your insightful and constructive comments have been invaluable in improving the rigor and clarity of our work. We will incorporate your suggestions and revise the manuscript accordingly.

---

### Official Review · Reviewer_nUmk · 2025-06-30

**Clarity:** 2
**Significance:** 2
**Originality:** 2
**Rating:** 4
**Confidence:** 3

**Summary:**

This paper proposes a novel penalty-based method for bilevel hyperparameter optimization (HO), focusing on lower-level (LL) problems with nonsmooth regularization. The approach builds on the LL duality framework and introduces a penalty formulation to eliminate nested loops. Two single-loop first-order algorithms are developed to handle separable and nonseparable regularizers using epigraphical projections. The paper provides convergence guarantees without requiring strong convexity in the LL objective and demonstrates empirical performance on several synthetic and real datasets.

**Questions:**

1.Can you provide practical examples from deep learning where nonsmooth LL problems (e.g., using L1 or nuclear norm regularization) are essential?

2.How does the computational cost of epigraphical projections scale with dimensionality, compared to unrolled-gradient methods?

3.What exactly is the novelty compared to BiC-GAFFA or LDMMA? Both are single-loop and handle nonsmoothness efficiently.

4.The phrase “only first-order differentiability” is potentially misleading — which parts of the objective are assumed differentiable, and which are not?

**Ethical Concerns:**

["NO or VERY MINOR ethics concerns only"]

**Final Justification:**

My concerns have been largely addressed, and I will therefore cautiously raise my score to a Borderline accept.

**Limitations:**

No discussion is provided on how this method integrates with modern DL frameworks in very high-dimensional settings with complex constraints.

**Quality:**

2

**Strengths And Weaknesses:**

Strengths

1.The reformulation is mathematically rigorous and extends prior work ([22]) using duality in bilevel optimization.

2.The proposed algorithms are single-loop, Hessian-free, and efficiently handle nonsmooth terms through projection, which improves practicality.

3.The paper proves non-asymptotic convergence under mild conditions, including nonsmoothness in the LL problem.

4.Empirical Results: The proposed LDPM method performs competitively or better in several benchmark tasks in terms of test error and runtime.

Weaknesses

1.The claimed novelty of “single-loop first-order method” lacks strong justification, as prior works like BiC-GAFFA and BOME also fit this category.

2.In several datasets (e.g., Table 2), LDPM only slightly outperforms competitors. Significance tests or robustness analysis are missing.

3.The method relies on epigraphical projections, but their cost (especially for group or nuclear norms) is not analyzed or compared with existing methods.

---

> ### Author Rebuttal · Authors · 2025-07-26
>
> Thanks for your insightful and constructive review comments! We have carefully considered all of your questions and concerns, and provide our point-by-point responses below.
>
> # Weakness 1
>
> Thank you for kindly pointing this out. BiC-GAFFA is a single-loop first-order method for bilevel optimization with coupled lower-level constraints, but it relies on differentiability and Lipschitz smoothness of the lower-level functions, as outlined in Assumption 4.2 of [1]. Likewise, BOME assumes smoothness for both the upper and lower levels, as stated in Assumption 2 of [2]. MEHA [3] considers nonsmooth lower-level problems but addresses them through a Moreau envelope smoothing approach.
>
> In contrast, our method LDPM does not rely on any smoothing approximations and instead handles nonsmooth objectives directly via efficient epigraphic projections with less computation cost. We will emphasize the distinction from other single-loop first-order methods in the revised version.
>
> [1] Wei Yao, Haian Yin, Shangzhi Zeng, and Jin Zhang. Overcoming lower-level constraints in bilevel optimization: A novel approach with regularized gap functions. arXiv preprint arXiv:2406.01992, 2024.
>
> [2] Bo Liu, Mao Ye, Stephen Wright, Peter Stone, and Qiang Liu. Bome! bilevel optimization made easy: A simple first-order approach. Advances in neural information processing systems,35:17248–17262, 2022.
>
> [3] Risheng Liu, Zhu Liu, Wei Yao, Shangzhi Zeng, and Jin Zhang. Moreau envelope for nonconvex bi-level optimization: A single-loop and hessian-free solution strategy. arXiv preprint arXiv:2405.09927, 2024.
>
> # Weakness 2
>
> We wish to clarify that we conduct the sensitivity analysis in our experiments, which is introduced in lines 276-278 and detailed in Appendix D.3.
>
> We respectfully note that LDPM not only achieves lower ultimate error than competitors but also exhibits **faster convergence**. This improvement stems from our algorithm’s reduced computational overhead: instead of relying on smoothing schemes, we directly address the nonsmooth components via efficient projection operations detailed in Appendix B. Consequently, each iteration in our method requires only gradient evaluations and projections, which results in lower per‑iteration cost and faster overall convergence.
>
> By contrast, competing methods such as LDMMA and VF-iDCA rely on off-the-shelf solvers in each iteration, which can lead to significantly reduced efficiency when applied to large-scale problems. BiC-GAFFA depends on smoothing-based reformulation that converts the nonsmooth lower-level problem into a constrained lower-level problem, which also introduces additional computational burden.
>
> # Weakness 3 & Question 2: computation cost of epigraphical projections
>
> Thank you for this insightful question! In Appendix B, we show that the projections onto the $l_2$ norm cones and rotated second-order cones have closed-form solutions, whose cost is $O(n)$. For the $l_1$ norm, squared $l_2$ norm, nuclear norm and spectral norm, which do not admit explicit epigraphic projection formulas, the projection can be computed by finding the root of a nonincreasing **scalar** function $\psi(\lambda)$ or by solving a **one-dimensional** optimization problem, **as detailed in lines 628, 635, 663 and 676**. These procedures leverage efficient quick-select routines (as in [4]) to ensure fast computation.
>
> - For epigraphic projection for vector inputs, the overall runtime is $\tilde{\mathcal{O}}(n)$, where the tilde hides logarithmic factors.
>
> - For epigraphic projection for matrix inputs, the dominant cost arises from computing the SVD, which takes $\tilde{\mathcal{O}}(mn\min\\{m,n\\})$, followed by a root-finding step of complexity $\mathcal{O}(r)$, where $r = \text{rank}(X)$.
>
> - For epigraphic projection of group norms, we project each group independently and then assemble the full vector, resulting in a total cost of $\mathcal{O}(n)$.
>
> Since each projection uses the same low-dimensional routine, the overall computation remains efficient.
> We will add the associated runtime in our revised manuscript.
>
> [4] Po-Wei Wang, Matt Wytock, and Zico Kolter. Epigraph projections for fast general convex programming. In International Conference on Machine Learning, pages 2868–2877. PMLR, 2016.
>
> # Question 1: practical examples in deep learning
>
> We refer to several examples from deep learning concerning nonsmooth LL problems. References [5,6,7,8] discuss the bilevel framework concerning $\ell_1$ or nuclear norm in deep neural network (DNN) and image reconstruction. References [9,10] discuss the nonsmooth regularization in convolutional neural networks (CNN)
>
> [5] Xu X, Zhang L, Kong Q. Learning Bilevel Sparse Regularized Neural Network[C]//CAAI International Conference on Artificial Intelligence. Cham: Springer International Publishing, 2021: 188-199.
>
> [6] Ghosh A, Mccann M T, Ravishankar S. Bilevel Learning of l1 Regularizers with Closed-Form Gradients (BLORC)[C]//ICASSP 2022-2022 IEEE International Conference on Acoustics, Speech and Signal Processing (ICASSP). IEEE, 2022: 1491-1495.
>
> [7] Barlaud M, Perez G, Marmorat J P. A new Linear Time Bi-level $l_{1,\infty}$ projection; Application to the sparsification of auto-encoders neural networks[J]. CoRR, 2024.
>
> [8] Yang C, Zhao P, Li Y, et al. Pruning parameterization with bi-level optimization for efficient semantic segmentation on the edge[C]//Proceedings of the IEEE/CVF Conference on Computer Vision and Pattern Recognition. 2023: 15402-15412.
>
> [9] Lee D, Lee E, Hwang Y. Pruning from scratch via shared pruning module and nuclear norm-based regularization[C]//Proceedings of the IEEE/CVF Winter Conference on Applications of Computer Vision. 2024: 1393-1402.
>
> [10] Shridhar K, Laumann F, Liwicki M. A comprehensive guide to bayesian convolutional neural network with variational inference[J]. arXiv preprint arXiv:1901.02731, 2019.
>
> # Question 3: our novelty
>
> **Comparison to BiC-GAFFA:** Our framework LDPM does not rely on smoothing schemes. We directly address the nonsmooth components via efficient projection operations. Therefore, each iteration in our method requires only gradient evaluations and projections. By contrast, BiC-GAFFA depends on smoothing-based reformulation that converts the nonsmooth lower-level problem into a constrained lower-level problem, which also introduces additional computational burden.
>
> **Comparison to LDMMA:** LDMMA differs significantly from our proposed LDPM. In the reformulation, we utilize the inner structure of $l(x)$, which is different from LDMMA. The methods for solving the subproblem also differ substantially: we primarily combine gradient descent and projection algorithms, whereas LDMMA directly uses an off-the-shelf solver to address the subproblem. The following outlines the key challenges faced by LDMMA in real-world applications.
>
> 1. A key requirement for applying LDMMA is conic programming reformulation of the original problem, which necessitates expressing all objective functions and constraints in conic or linear forms. LDMMA cannot deal with general convex losses, e.g., the logistic loss.
>
> 2. In numerical experiments, each iteration of LDMMA invokes a solver based on interior point methods, which are not suitable for large-scale problems. Consequently, the efficiency of LDMMA decreases significantly when handling large-scale problems.
>
> 3. An important aspect of the LDMMA algorithm is the value function and an additional parameter $\epsilon$, which yields an approximation of the original BLO. LDPM directly solves the original BLO.
>
>  Indeed, LDPM addresses the above challenges and demonstrates improved numerical performance.
>
> # Question 4
>
> Thank you for pointing out this imprecise phrasing! Here, we meant that our framework relies only on the first-order differentiability of the loss functions $L$ and $l$ in UL and LL problems. We will correct this in a revised version.
>
> # Limitations
>
> We thank the reviewer for raising this important point. Indeed, our current work is focused exclusively on nonsmooth bilevel optimization problems and does not target deep learning applications. Extending our framework to handle high‑dimensional neural network settings with complex lower‑level constraints is an exciting direction that we plan to pursue in future work. In particular, we will investigate how to integrate our framework into the large parameter spaces typical of contemporary neural network models.

---

> > ### Comment · Reviewer_nUmk · 2025-08-06
> >
> > Thank the authors for their thoughtful response. Their clarifications regarding my concerns and those raised by other reviewers have partially alleviated my reservations. I will therefore cautiously increase my score.

---

> ### Author Response · Authors · 2025-08-05
>
> Dear Reviewer nUmk,
>
> We hope this message finds you well.
>
> As the author-reviewer discussion period is coming to an end, we would like to kindly ask whether we have addressed your concerns. If our responses have adequately addressed your concerns, we would be grateful if you could consider updating your score.
>
> If there are any remaining concerns, we would greatly appreciate it if you could share them with us, so we may have enough time to provide a detailed response. Thank you once again for your time and valuable feedback!
>
> Best regards,
> Authors

---

> ### Author Response · Authors · 2025-08-07
>
> We sincerely appreciate your thoughtful comments and your effort in reviewing our work. We are grateful for your raised rating. Your insightful and constructive comments have been invaluable in improving the rigor and clarity of our work. We will incorporate your suggestions and revise the manuscript accordingly.

---

### Official Review · Reviewer_Zrpb · 2025-06-30

**Clarity:** 3
**Significance:** 3
**Originality:** 3
**Rating:** 5
**Confidence:** 5

**Summary:**

The paper addresses a bi-level optimization problem featuring a non-smooth lower-level problem, specifically of the form: $ \min_xl(x)+\Sigma_i\lambda_iR_i(x)$, where $R_i$ defines as norms. The authors reformulate the lower-level problem and combine it with a penalty method to obtain a projection-friendly formulation, subsequently employing an ADMM-inspired approach for its solution.

**Questions:**

1. Clarification on Assumption 3.6:
Could you please elaborate on how the finiteness of the sum in Assumption 3.6 is guaranteed in most cases?

2. Generalizability of the Lower-Level Problem:
   The current formulation of the lower-level problem is restricted to a sum of norms. Is it possible to extend this framework to more general cases?&#x20;

3. Real-World Applicability in Experiments:
   The numerical experiments primarily focus on synthetic or idealized settings. Could additional experiments be conducted on real-world application scenarios to better demonstrate the method's practical utility?

4. Advantages of LDPM over BiC-GAFFA:
   In Figures 2 and 3, the performance of LDPM appears very close to that of BiC-GAFFA. What are the key advantages of LDPM that justify its use over existing methods?

**Ethical Concerns:**

["NO or VERY MINOR ethics concerns only"]

**Final Justification:**

My concerns have been fully addressed.

**Paper Formatting Concerns:**

None.

**Quality:**

3

**Strengths And Weaknesses:**

**Strengths:**

1. The approach used to handle the lower-level problem by considering duality and penalty is novel.

2. The utilization of projection-friendly norms to solve the reformulated lower-level problem is intriguing.

3. The mathematical derivation, explanation, and proofs are clear and correct.

**Weaknesses:**

1. Assumption 3.1 does not hold for the KL divergence. Although Remark 3.3 states that it is valid for the loss functions listed in Table 1, this limitation may affect real-world applications.

2. Due to the specific form of the lower-level problem (a sum of norms), its applicability to real-world problems is limited.

3. These limitations also reduce the persuasiveness of the numerical experiments.

4. While Assumption 3.6 is necessary for Theorem 3.7, it cannot be verified or satisfied a priori through hyperparameter settings before conducting experiments. Therefore, its practical significance may be limited.

---

> ### Author Rebuttal · Authors · 2025-07-27
>
> Thanks for your insightful and constructive review comments! We have carefully considered all of your questions and concerns, and provide our point-by-point responses below.
>
> # Weakness 1: Assumption 3.1 and KL divergence
>
> We thank the reviewer for noting that Assumption 3.1 excludes the KL divergence. While our current analysis indeed requires Lipschitz smoothness of the loss function, our framework and algorithm remain valid for other common loss functions in machine learning (e.g., least squares and logistic loss). We will consider in future work how to extend it to the KL divergence.
>
> # Weakness 2/3 & Question 2: the specific form of the lower-level problem (a sum of norms)
>
> - We wish to clarify that our work is squarely focused on bilevel hyperparameter optimization, where incorporating a regularization term of the “sum of norms” form is not an artificial constraint but rather a practical necessity. In real‑world hyperparameter tuning, such composite regularizers are ubiquitous. By designing LDPM around this specific structure, we directly address the core challenges practitioners face when tuning these essential hyperparameters, rather than treating the regularizer as a mere mathematical convenience.
>
> - We also wish to emphasize that our reformulation strategy extends beyond sums of norms. Whenever the lower‑level problem takes the form
> $$y \in \arg\min_y \\{g(x,y) + \sum_i g_i(x,y)\\},$$
> it can be reformulated via lower‑level duality (cf.[1, Lemma 2.1]), and our reformulation and penalty framework are both applicable. As long as each $g_i$ is prox‑friendly with respect to the LL variable $y$, both Algorithms 1 and 2 remain applicable. In that case, the epigraphical projection onto $\\{(y,t)\mid g_i(x,y)\leq t\\}$ can be performed uniformly via Theorem B.1, in which the resulting scalar equation can be solved efficiently using the well‑established quick-select algorithm analogous to those projections provided in Appendix B.
>
> [1] Chen et al. Lower-level duality based reformulation and majorization minimization algorithm for hyperparameter optimization.
>
> # Weakness 4 & Question 1: Assumption 3.6
>
> We appreciate the reviewer’s observation that Assumption 3.6 is rather strong. As discussed in [2,3], the convergence of nonconvex nonsmooth ADMM is highly challenging without imposing assumptions like Assumption 3.6, which is an open question. This assumption is widely employed in ADMM approaches [4,5,6,7].
> In our numerical experiments, Assumption 3.6 is empirically satisfied automatically.
>
> [2] Wang et al. Global convergence of ADMM in nonconvex nonsmooth optimization.
>
> [3] Lin et al. Alternating direction method of multipliers for machine learning.
>
> [4] Bai et al. An augmented lagrangian decomposition method for chance-constrained optimization problems.
>
> [5] Cui et al. Decision making undercumulative prospect theory: An alternating direction method of multipliers.
>
> [6] Shen et al. Augmented lagrangian alternating direction method for matrix separation based on low-rank factorization.
>
> [7] Xu et al. An alternating direction algorithm for matrix completion with nonnegative factors.
>
> # Question 3: real-world experiments
>
> Thank you for raising this important point. We have indeed conducted extensive experiments on real‑world datasets. Due to space constraints, we only provide brief summary in **Section 4.2**, while the dataset descriptions, experimental settings and results are detailed in **Appendix D.4**. Specifically, our real-world experiments are evaluated on three representative problems: Elastic Net, smoothed SVM, and sparse logistic regression, each tested across multiple real‑world datasets.
>
> # Question 4: advantages of LDPM over BiC-GAFFA
>
> We respectfully note that, in Figures 2 and 3, LDPM not only achieves lower ultimate error than BiC‑GAFFA but also exhibits **faster convergence**. This improvement stems from our algorithm’s reduced computational overhead: instead of relying on smoothing schemes, we directly address the nonsmooth components via efficient projection operations. Consequently, each iteration in our method requires only gradient evaluations and projections, which results in lower per‑iteration cost and faster overall convergence. By contrast, other existing methods for nonsmooth bilevel optimization depend on smoothing strategies or off‑the‑shelf solvers, introducing additional computational burden.

---

> > ### Comment · Reviewer_Zrpb · 2025-08-05
> >
> > Thanks for the reply. My concerns have been addressed, I am glad to raise my rating.

---

> ### Author Response · Authors · 2025-08-05
>
> We sincerely appreciate your thoughtful comments and the time you dedicated to reviewing our work. We are grateful for your raised rating. Your insightful and constructive suggestions have been invaluable in enhancing the rigor and clarity of our work. We will carefully incorporate your suggestions in the revised version of the manuscript.

---

### Note · Authors · 2025-08-13

We thank all reviewers for their careful reviews, constructive discussions and valuable time. Your feedback has improved clarity and presentation of our work. Reviewers recognize the novelty of our work, the fascinating and rigorous theory, and the comprehensive experimental valuation.

No major errors are identified. Comments mainly concern clarifying assumptions, minor textual edits, suggesting future extensions and discussing our method’s advantages, all of which are addressed in our rebuttal.

Reviewers Zrpb, nUmk, NsoD express support with questions on assumptions, applicability and advantages. In the rebuttal,
1. We show that our assumptions are reasonable, broadly applicable to machine learning scenarios, and aligned with standard conditions widely adopted in ADMM-based methods.
2. We also demonstrate applicability to BLO with a finite-sum lower-level structure. We will explore deep learning extensions as suggested by Reviewer nUmk.
3. We highlight computational gains: our method avoids smoothing strategies and off-the-shelf solvers, tackling nonsmoothness directly via efficient projections, which greatly reduces per-iteration cost and runtime, making it computationally **superior to existing single-loop first-order BLO algorithms**.

After we address the concerns, we appreciate that you **raise the ratings or maintain the positive score**.

Reviewer BK21 also support our rebuttal and all concerns in the official review are resolved. After that, you also find our theory fascinating. Regarding the comment “More Concerns on Proof”, we would emphasize that Questions 1, 2, 4, 5, 6 **do not indicate any logical gap. The proofs are correct as clarified in our reply.** Question 3 points to a **minor resolvable issue, which we have revised and explained**. We also carry out a thorough re-check and can confirm that the derivations are correct. We appreciate that you **update the score following our clarifications.**

In summary, we present novel projection-based single-loop first-order algorithms for nonsmooth bilevel hyperparameter optimization, which achieve superior efficiency with rigorous theoretical guarantees. All concerns have been addressed and the positive scores reflect novelty, correctness and applicability of our work. In the revision, we will further highlight our advantages, add explanations and justifications of the assumptions.

We sincerely thank you for handling our work. We would be glad to clarify any further questions you might have.

---

### Decision · Program_Chairs · 2025-09-17

**Decision:**

Reject

**Comment:**

This paper is rather borderline. During the final discussion, reviewers and myself have checked the issue pointed out by Reviewer BK21 in the proof of Theorem 3.5. We felt this gap may need a new round of revision to be fully fixed. Unfortunately, we may have to reject this time.